# Rank Overspecified Robust Matrix Recovery: Subgradient Method and Exact Recovery

**Lijun Ding**
Department of Mathematics
University of Washington
ljding@uw.edu

**Liwei Jiang** *
School of ORIE
Cornell University
lj282@cornell.edu

**Yudong Chen**
Department of Computer Sciences
University of Wisconsin-Madison
yudongchen@wisc.edu

**Qing Qu**
Department of EECS
University of Michigan-Ann Arbor
qingqu@umich.edu

**Zhihui Zhu**
Department of ECE
University of Denver
zhihui.zhu@du.edu

## Abstract

We study the robust recovery of a low-rank matrix from sparsely and grossly corrupted Gaussian measurements, with no prior knowledge on the intrinsic rank. We consider the robust matrix factorization approach. We employ a robust $\ell_1$ loss function and deal with the challenge of the unknown rank by using an overspecified factored representation of the matrix variable. We then solve the associated non-convex nonsmooth problem using a subgradient method with diminishing stepsizes. We show that under a regularity condition on the sensing matrices and corruption, which we call restricted direction preserving property (RDPP), even with rank overspecified, the subgradient method converges to the exact low-rank solution at a sublinear rate. Moreover, our result is more general in the sense that it automatically speeds up to a linear rate once the factor rank matches the unknown rank. On the other hand, we show that the RDPP condition holds under generic settings, such as Gaussian measurements under independent or adversarial sparse corruptions, where the result could be of independent interest. Both the exact recovery and the convergence rate of the proposed subgradient method are numerically verified in the overspecified regime. Moreover, our experiment further shows that our particular design of diminishing stepsize effectively prevents overfitting for robust recovery under overparameterized models, such as robust matrix sensing and learning robust deep image prior. This regularization effect is worth further investigation.

## 1 Introduction

Robust low rank matrix recovery problems are ubiquitous in computer vision [1], signal processing [2], quantum state tomography [3], etc. In the vanilla setting, the task is to recover a rank $r$ positive semidefinite (PSD) matrix $X_\natural \in \mathbb{R}^{d \times d}$ ($r \ll d$) from a few corrupted linear measurements $\{(y_i, A_i)\}_{i=1}^n$ of the form

$$y_i = \langle A_i, X_\natural \rangle + s_i, \quad i = 1, \ldots, m, \tag{1.1}$$

where $\{s_i\}_{i=1}^m$ are the corruptions. This model can be written succinctly as $y = \mathcal{A}(X_\natural) + s$, where $\mathcal{A} : \mathbb{R}^{d \times d} \to \mathbb{R}^m$ is the measurement operator. Recently, tremendous progress has been made on this problem, leveraging advances on convex relaxation [1,2], nonconvex optimization [4–7], and high dimension probability [8–11]. However, several practical challenges remain:

---

*The first two authors contributed to this work equally.

- **Nonconvex and nonsmooth optimization for sparse corruptions.** For computational efficiency, we often exploit the low-rank structure via the factorization $X = FF^\top$ with $F \in \mathbb{R}^{d \times r}$, and solve the problem with respect to $F$ via nonconvex optimization. This problem is challenging since the corruption can be arbitrary. Fortunately, the corruptions $s_i$ are *sparse* [1]. This motivates us to consider the $\ell_1$ loss function and lead to a nonsmooth nonconvex problem.

- **Unknown rank and overspecification.** Another major challenge in practice is that the *exact* true rank $r$ is usually *unknown* apriori.[2] To deal with this issue, researchers often solve the problem in *rank overspecified regime*, using a factor variable $F \in \mathbb{R}^{d \times k}$ with a larger inner dimension $k > r$. Most existing analysis relies on the $k$-th eigenvalue of $X_\natural$ being nonzero and is inapplicable in this setting. Recent work studied the problem either in some special case (e.g., rank one) [13] or under unrealistic communitability conditions [14].

**Our approach and contributions.** To deal with these challenges, we consider the following nonconvex and nonsmooth formulation with overspecified rank $k \geq r$,

$$\min_{F:\, F \in \mathbb{R}^{d \times k}} f(F) := \frac{1}{2m} \sum_{i=1}^{m} \left| \left\langle A_i, FF^\top \right\rangle - y_i \right| = \frac{1}{2m} \left\| \mathcal{A}(FF^\top) - y \right\|_1, \tag{1.2}$$

and apply a subgradient method with *adaptive* stepsize. We summarize our contributions below.

- **Exact recovery with rank overspecification.** In Theorem 3.2, we show that under Gaussian measurements and sparse corruptions, with $m = \tilde{\mathcal{O}}(dk^3)$ samples [3], even with overspecified rank $k > r$, our subgradient method with spectral initialization converges to the ground truth $X_\natural$ *exactly*. When $k > r$, the convergence rate is $\mathcal{O}(\frac{1}{t})$ in the operator norm error. $\left\| F_t F_t^\top - X_\natural \right\|$. When $k = r$, the convergence rate boosts to linear (geometric) rate.

- **Convergence under a regularity condition.** Our convergence result is established under a regularity condition termed *restricted direction preserving property* (RDPP); see Definition 2.1. RDPP ensures that the *direction* of the finite-sample subgradient stays close to its population version along the algorithm's iteration path. The key to our proof is to use this property to leverage a recent result on rank overspecification under *smooth* formulations [15]. Moreover, we show that RDPP holds with high probability under arbitrary sparse corruptions and Gaussian measurement matrices.

- **Adaptive stepsize and implicit regularization** As shown in [13, 14], for overparametrized robust matrix recovery problems, subgradient methods with constant stepsizes suffer from overfitting, even when initialized near the origin. In contrast, numerical experiments demonstrate that our theory-inspired diminishing stepsize rule has an implicit regularization effect that prevents overfitting. Our stepsize rule is easy to implement and almost tuning-free, and performs better than other non-adaptive stepsize rules (e.g., sublinear or geometric decaying stepsizes). We demonstrate similar good performance of our method (with a fixed decaying stepsize) in image recovery with a robust deep image prior (DIP) [16].

**Related work.** The nonconvex, factorized formulation (1.2) is computationally more efficient than convex relaxation approaches [1, 17–19], as it involves a smaller size optimization variable and only requires one partial singular value decomposition (SVD) in the initialization phase. In the corruption-free ($s_i \equiv 0$) and exact rank ($k = r$) setting, provably efficient methods have been studied for the *smooth*, least squares version of (1.2) [20–23]; in this case, the loss function has no spurious local minimum and is locally strongly convex [24, 25]. For smooth formulations with rank overspecification ($k \geq r$ or even $k = d$), gradient descent initialized near the origin is shown to *approximately* recover the ground truth matrix [26, 27].

When the rank $r$ is known and measurement corruption is present, the work [4] considers the smooth formulation and proposes a gradient descent method with a robust gradient estimator using median truncation. The nonsmooth formulation (1.2) is considered in a recent line of work [5–7], again in the known rank setting ($k = r$), where they show that the loss function satisfies certain sharpness [28] properties and hence subgradient methods converge locally at a linear rate. The number of samples required in these works for exact recovery of $X_\natural$ is $\mathcal{O}(dr^2)$ when the corruption rate is sufficiently small. The $\mathcal{O}(r^2)$ dependence is due to their initialization procedure. We emphasize that the above results crucially rely on knowing and using the exact true rank. When the rank is unknown, the work [14] considers a doubly over-parameterized formulation but requires a stringent, commutability

---

[2]There are very few exceptions, such as the case of robust phase retrieval with $r = 1$ [12].

[3]Here $\tilde{\mathcal{O}}$ hides the condition number of $X_\natural$ and other log factors.

|  | [4–7] | [13] | Ours |
|---|---|---|---|
| Rank specification | $k = r$ | $k \geq r = 1$ | $k \geq r$ |
| Exact Recovery | ✓ | ✗ | ✓ |
| Sample complexity | $\tilde{\mathcal{O}}(dr^2)$ | $\tilde{\mathcal{O}}\left(\frac{dk}{\delta^4}\right)$ | $\tilde{\mathcal{O}}(dk^3)$ |

Table 1: **Comparison of results on robust matrix sensing.**

assumptions on the measurement matrices; moreover, their analysis concerns continuous dynamic of gradient descent and is hard to generalize to discrete case. Our results only impose standard measurement assumptions and apply to the usual discrete dynamic of subgradient methods.

The work most related to ours is [13], which studies subgradient methods for the nonsmooth formulation (1.2) in the rank-one case ($r = 1$). Even in this special case, only approximate recovery is guaranteed (i.e., finding an $F$ with $\|FF^\top - X_\natural\|_F \lesssim \delta \log(\frac{1}{\delta})$ for a fixed $\delta > 0$). Additionally, the number of samples required in [13] for this approximate recovery is $\tilde{\mathcal{O}}\left(\frac{dk}{\delta^4}\right)$ when the corruption rate is sufficiently small. We note that the lack of exact recovery is inherent to their choice of stepsizes, and spectral initialization already guarantees $\|FF^\top - X_\natural\|_F \lesssim \delta$ [13, Equation (85)]. In contrast, our result applies beyond rank-1 case and guarantees exact recovery thanks to the proposed adaptive diminishing stepsize rule. We summarize the above comparisons in Table 1.

## 2 Models, regularity, and identifiability

In this section, we formally describe our model and assumptions. We then introduce the restricted direction preserving property and establish a basic identifiability result under rank overspecification, both key to the convergence analysis of our algorithm given in Section 3 to follow.

### 2.1 Observation and corruption models

In this work, we study subgradient methods for the robust low-rank matrix recovery problem (1.2) under the observation model (1.1). Throughout this paper, we assume that the sensing matrices $A_i \in \mathbb{R}^{d \times d}$, $i = 1, \ldots, m$ are generated *i.i.d.* from the Gaussian Orthogonal Ensemble (GOE).[4] That is, each $A_i$ is symmetric, with the diagonal entries being *i.i.d.* standard Gaussian $N(0, 1)$, and the upper triangular entries are *i.i.d.* $N(0, \frac{1}{2})$ independent of the diagonal. We consider two models for the corruption $s = (s_1, \ldots, s_m)^\top$: (*i*) arbitrary corruption (AC) and (*ii*) random corruption (RC).

**Model 1** (AC model). *Under the model AC($X_\natural, \mathcal{A}, p, m$), the index set $S \subset \{1, \ldots, m\}$ of nonzero entries of $s$ is uniformly random with size $\lfloor pm \rfloor$, independently of $\mathcal{A}$, and the values of the nonzero entries are arbitrary.*

**Model 2** (RC model). *Under the model RC($X_\natural, \mathcal{A}, p, m$), each $s_i$, $i = 1, \ldots, m$, is a random variable, independent of $\mathcal{A}$ and each other, with probability $p \in (0, 1)$ being nonzero.*

In the AC model, the values of the nonzero entries of $s$ can be chosen in a coordinated and potentially adversarial manner dependent on $\{A_i\}$ and $X_\natural$, whereas their locations are independent. Therefore, this model is more general than those in [1, 29], which assume random signs.[5] The RC model puts no restriction on the random variables $\{s_i\}$ except for independence, and hence is more general than the model in [13]. In particular, the distributions of $\{s_i\}$ can be heavy-tailed, non-identical and have non-zero means (where the means cannot depend on $\{A_i\}$). Under the RC model, we can allow the corruption rate $p$ to be close to $\frac{1}{2}$ and still achieve exact recovery (see Theorem 3.2).

### 2.2 Restricted direction preserving property (RDPP)

When the true rank $r$ is known, existing work [5, 6] on subgradient methods typically makes use of the so-called $\ell_1/\ell_2$-restricted isometry property (RIP), which ensures that the function $g(X) = \frac{1}{m}\|\mathcal{A}(X) - y\|_1$ concentrates uniformly around its expectation. Note that the factorized objective $f$ in (1.2) is related to $g$ by $f(F) = \frac{1}{2}g(FF^\top)$. With $\ell_1/\ell_2$-RIP, existing work establishes the sharpness of $g$ when $k = r$, i.e., linear growth around the minimum [5, Proposition 2]. In the

---

[4]Our results can be easily extended sensing matrices with i.i.d. standard normal entries since $X_\natural$ is symmetric.

[5]Our results can be extended to the even more general setting with adversarial locations [4–7], with an extra poly$(k, \kappa)$ factor in the requirement of the corruption rate $p$ compared to the one we present in Section 3.

| $d$ | $d = 50$ | $d = 100$ | $d = 200$ | $d = 300$ |
|---|---|---|---|---|
| RDPP: $\left\| D(X_\natural) - \sqrt{\frac{2}{\pi}} \frac{X_\natural}{\|X_\natural\|_{\mathrm{F}}} \right\|$ | 0.62/0.27 | 0.62/0.28 | 0.63/0.28 | 0.63/0.28 |
| sign-RIP [13]: $\left\| D(X_\natural) - \sqrt{\frac{2}{\pi}} \frac{X_\natural}{\|X_\natural\|_{\mathrm{F}}} \right\|_{\mathrm{F}}$ | 2.26/1.01 | 3.17/1.42 | 4.48/2.00 | 5.49/2.45 |

Table 2: **Comparison of RDPP and sign-RIP** for $d \in \{50, 100, 200, 300\}$ and $r \in \{1, 5\}$. For each entry with two numbers, the left corresponds to $r = 1$ and the right corresponds to $r = 5$. We set $p = 0$ and $m = 5dr$. Note sign-RIP is never satisfied as the above entries for sign-RIP has value larger than or equal to 1.

rank-overspecification setting, however, $g$ is *not* sharp, as it grows more slowly than linear in the direction of superfluous eigenvectors of $F$. We address this challenge by directly analyzing the subgradient dynamics, detailed in Section 3, and establishing concentration of the *subdifferential* of $g$ rather than $g$ itself.

Recall the regular subdifferential of $g(X)$ [30] is

$$\mathcal{D}(X) := \frac{1}{m} \sum_{i=1}^{m} \overline{\operatorname{sign}}(\langle A_i, X \rangle - s_i) A_i, \quad \text{and} \quad \overline{\operatorname{sign}}(x) := \begin{cases} \{-1\}, & x < 0 \\ [-1, 1], & x = 0 \\ \{1\}, & x > 0 \end{cases} \quad (2.1)$$

Since our goal is to study the convergence of the subgradient method, which utilizes only one member $D(X)$ and will be illustrated in Section 3.1, it is enough to consider this member $D(X)$ by setting $\operatorname{sign}(x) = \begin{cases} -1, & x < 0 \\ 1, & x \geq 0 \end{cases}$. We are now ready to state the RDPP condition.

**Definition 2.1** (RDPP). *The measurements and corruption $\{(A_i, s_i)\}_{i=1}^{m}$ are said to satisfy restricted direction preserving property with parameters $(k', \delta)$ and a scaling function $\psi : \mathbb{R}^{d \times d} \to \mathbb{R}$ if for every symmetric matrix $X \in \mathbb{R}^{d \times d}$ with rank no more than $k'$, we have*

$$D(X) := \frac{1}{m} \sum_{i=1}^{m} \operatorname{sign}(\langle A_i, X \rangle - s_i) A_i, \quad \text{and} \quad \left\| D(X) - \psi(X) \frac{X}{\|X\|_{\mathrm{F}}} \right\| \leq \delta. \quad (2.2)$$

Here, the scaling function accounts for the fact that the expectation is not exactly $\frac{X}{\|X\|_{\mathrm{F}}}$, e.g., when no corruption, $\nabla \mathbb{E}(g(X)) = \sqrt{\frac{2}{\pi}} \frac{X}{\|X\|_{\mathrm{F}}}$. How does RDPP relate to the $\ell_1/\ell_2$-RIP? In the absence of corruption, as shown in Appendix H, RDPP implies $\ell_1/\ell_2$-RIP for $\delta \lesssim \frac{1}{\sqrt{r}}$. Conversely, we know if $\ell_1/\ell_2$-RIP condition holds for all rank $d$ matrices, then RDPP holds when $p = 0$ [13, Proposition 9].

Our RDPP is inspired by the condition *sign-RIP* introduced in the recent work [13, Definition 1]. The two conditions are similar, with the difference that RDPP is based on the operator norm (see the inequality in equation (2.2)), whereas sign-RIP uses the Frobenius norm, hence a more stringent condition. In fact, the numerical results in Table 2 suggest that such a Frobenius norm bound *cannot* hold if $m$ is on the order $\mathcal{O}(dr)$ even for a fixed rank-1 $X_\natural$—in our experiments $\left\| D(X_\natural) - \sqrt{\frac{2}{\pi}} \frac{X_\natural}{\|X_\natural\|_{\mathrm{F}}} \right\|_{\mathrm{F}}$ is always larger than 1.[6] [7] In contrast, standard Gaussian/GOE sensing matrices satisfy RDPP for any rank $k' \geq 1$, and it suffices for our analysis. Indeed, we can show the following.

**Proposition 2.2.** *Let $m \gtrsim \frac{dk' \left( \log(\frac{1}{\delta}) \vee 1 \right)}{\delta^4}$. For Model 1, $(k', \delta + 3\sqrt{\frac{dp}{m}} + 3p)$-RDPP holds with a scaling function $\psi(X) = \sqrt{\frac{2}{\pi}}$ with probability at least $1 - \exp(-(pm+d)) - \exp(-c'm\delta^4)$. For Model 2,*

---

[6] To the best of our knowlegde, the proof of [13, Proposition 1] actually establishes a bound on the $\ell_2$ norm of top $r$ singular values, called partial Frobenius norm, rather than on Frobenius norm (i.e., the $\ell_2$ norm of all singular values). This is consistent with our empirical observations in Table 2.

[7] After the first version of this paper was posted, the work [31] became available, which is an updated version of [13]. In this updated paper, the definition and associated results for sign-RIP (see [31, Definition 2 and Theorem 1]) use the rank-$r$ partial Frobenius norm, instead of the Frobenius norm as used in the original [13, Definition 1].

$(k', \delta)$-*RDPP holds with a scaling function* $\psi(X) = \frac{1}{m} \sum_{i=1}^{m} \sqrt{\frac{2}{\pi}} \left( 1 - p + p\mathbb{E}_{s_i} \left[ \exp(-\frac{s_i^2}{2\|X\|_{\mathrm{F}}^2}) \right] \right)$ *with probability at least* $1 - Ce^{-cm\delta^4}$. *Here* $c, c', C$ *are universal constants.*

Our proof of Proposition 2.2 follows that of [13, Proposition 5] with minor modifications accounting for our corruption models and the definition of RDPP. For completeness we provide the proof in Appendix C. The operator norm bound in the definition (2.2) of RDPP turns out to be sufficient for our purpose in approximating the population subgradient of $f$ for our factor $F$.

### 2.3 Identifiability of the ground truth $X_\natural$

If the exact rank is given (i.e., $(k = r)$), it is known that our objective function (1.2) indeed identifies $X_\natural$, in the sense that any global optimal solution $F_\star$ coincides with $X_\natural = F_\star F_\star^\top$ [5]. When $k > r$, one may suspect that there might be an idenitfiability or overfitting issue due to rank oversepcification and the presence of corruptions, i.e., there might be a factor $F$ that has better objective value than $F_\star$, where $F_\star F_\star^\top = X_\natural$. However, so long as the number of samples are sufficiently large $m = \Omega(dk)$, we show that the identifiability of $X_\natural$ is ensured. We defer the proof to Appendix F.

**Theorem 2.3.** *Fix any* $p < \frac{1}{2}$. *Under either Model 1 or Model 2 with corruption rate* $p$, *if* $m \geq cdk$, *then with probability at least* $1 - e^{-c_1 m}$, *any global optimizer* $F_\star$ *of (1.2) satisfies* $F_\star F_\star^\top = X_\natural$. *Here, the constants* $c, c_1 > 0$ *only depend on* $p$.

Here, overfitting does not occur since sample size $m$ matches the degree of freedom of our formulation $\mathcal{O}(dk)$, making it possible to guarantee success of the algorithm described in Section 3. On the other hand, if we are in the heavily overparameterized regime with $m = \mathcal{O}(dr) \ll dk$ (e.g., $k = d$),[8] the above identifiability result should not be expected to hold. Perhaps surprisingly, in Section 4 we empirically show that the proposed method works even in this heavy overparametrization regime with $k = d$. This is in contrast to the overfitting behavior of subgradient methods with constant stepsizes observed in [14], and it indicates an implicit regularization effect of our adaptive stepsize rule.

## 3 Main results and analysis

In this section, we first introduce the subgradient method with implementation details. We then show our main convergence results, that subgradient method with diminishing stepsizes converge to the exact solutions even under the setting of overspecified rank. Finally, we sketch the high-level ideas and intuitions of our analysis. All the technical details are postponed to the appendices

### 3.1 Subgradient method with diminishing stepsizes

A natural idea of optimizing the problem (1.2) is to deploy the subgradient method starting from some initialization $F_0 \in \mathbb{R}^{d \times k}$, and iterate[9]

$$F_{t+1} = F_t - \eta_t g_t, \quad g_t = D(FF^\top - X_\natural)F, \quad , t = 0, 1, 2, \ldots. \tag{3.1}$$

with certain stepsize rule $\eta_t \geq 0$. The vector $g_t$ indeed belongs to the regular subdifferential of $\partial f(F_t) = \mathcal{D}(F_t F_t^\top - X_\natural)F_t$ by chain rule (recall $\mathcal{D}$ in (2.1)).

For optimizing a nonconvex and nonsmooth problem with subgradient methods, it should be noted that the choice of initialization and stepsize is critical for algorithmic success. As observed in Section 4.1, a fixed stepsize rule, i.e., stepsize determined before seeing iterates, such as geometric or sublinear decaying stepsize, cannot perform uniformly well in various rank specification settings, e.g., $k = r, r \leq k = \mathcal{O}(r), k = d$. Hence it is important to have an adaptive stepsize rule.

**Initialization**   We initialize $F_0$ by a spectral method similar to [13] yet we do not require $F_0 F_0^\top$ to be close to $X_\natural$ even in operator norm. We discuss the initialization in more detail in Section 3.4.

---

[8]A few authors [13, 15] refer to the setting with $k > r$ and $m \asymp dk$ as "overparametrization" since $k$ is larger than necessary. Others reserve the term for the setting $m < dk$, i.e., the number of parameters exceeds the sample size. We generally refer to the former setting as "(rank-)overspecifcation" and the latter as "overparametrization".

[9]Recall the definition of the subgradient $D(X) = \frac{1}{m} \sum_{i=1}^{m} \mathrm{sign}(\langle A_i, X \rangle - s_i) A_i$ in (2.2) for the objective, $g(X) = \frac{1}{2m} \|\mathcal{A}(X) - y\|_1$ in the original space $X \in \mathbb{R}^{d \times d}$.

**Stepsize choice** To obtain exact recovery, we deploy a simple and adaptive diminishing stepsize rule for $\eta_t$, which is inspired by the analysis and is easy to tune. As we shall see in Theorem 3.1 and its proof intuition in Section 3.3, the main requirement for stepsize is that it scales with $\|FF^\top - X_\natural\|_F$ for an iterate $F$, and hence diminishing and adaptive as $FF^\top \to X_\natural$. To estimate $\|FF^\top - X_\natural\|_F$ when corruption exists, we define the operator $\tau_{\mathcal{A},y}(F) : \mathbb{R}^{d \times k} \to \mathbb{R}$ and the corresponding stepsize:

$$\tau_{\mathcal{A},y}(F) = \xi_{\frac{1}{2}}\left(\{|\langle A_i, FF^\top\rangle - y_i|\}_{i=1}^m\right), \quad \text{and} \quad \eta_t = C_\eta \tau_{\mathcal{A},y}(F_t), \tag{3.2}$$

where $\xi_{\frac{1}{2}}(\cdot)$ is the median operator, and $C_\eta > 0$ is a free parameter to be chosen. As $\mathbb{E}|\langle A_1, FF^\top\rangle - y_1| = \sqrt{\frac{2}{\pi}}\|FF^\top - X_\natural\|_F$ without corruption, we expect $\tau_{\mathcal{A},y}(F)$ to estimate $\|FF^\top - X_\natural\|_F$ well up to some scalar when corruption exists since median is robust to corruptions.

## 3.2 Main theoretical guarantees

First, we present our main convergence result, that ensures that the iterate $F_t F_t^\top$ in subgradient method (3.1) converges to the rank $r$ ground-truth $X_\natural$, based on conditions on initialization, stepsize, and RDPP. Let the condition number of $X_\natural$ be $\kappa := \frac{\sigma_1(X_\natural)}{\sigma_r(X_\natural)}$, the result can be stated as follows.

**Theorem 3.1.** *Suppose the following conditions hold:*

(i) *Let $0 < \epsilon < 1$ be a fixed parameter. Suppose $F_0$ satisfies*

$$\|F_0 F_0^\top - c^* X_\natural\| \leq c_0 \sigma_r / \kappa. \tag{3.3}$$

*with some constant $c^* \in [\epsilon, \frac{1}{\epsilon}]$ and $c_0 = \tilde{c}_0 \epsilon$ for some sufficiently small $\tilde{c}_0$, which only depends on $c_1$ below. Moreover, all the constants $c_i, i \geq 3$ in the following depend only on $\epsilon$ and $c_1$.*

(ii) *The stepsize satisfies $0 < c_1 \frac{\sigma_r}{\sigma_1^2} \leq \frac{\eta_t}{\|F_t F_t^\top - X_\natural\|_F} \leq c_2 \frac{\sigma_r}{\sigma_1^2}$ for some small numerical constants $c_1 < c_2 \leq 0.01$ and all $t \geq 0$.*

(iii) *$(r+k, \delta)$-RDPP holds for $\{A_i, s_i\}_{i=1}^m$ with $\delta \leq \frac{c_3}{\kappa^3 \sqrt{k}}$ and a scaling function $\psi \in \left[\sqrt{\frac{1}{2\pi}}, \sqrt{\frac{2}{\pi}}\right]$.*

*Define $\mathcal{T} = c_4 \kappa^2 \log \kappa$. Then, we have a sublinear convergence in the sense that for any $t \geq 0$,*

$$\left\|F_{t+\mathcal{T}} F_{t+\mathcal{T}}^\top - X_\natural\right\| \leq c_5 \sigma_1 \frac{\kappa}{\kappa^3 + t}.$$

*Moreover, if $k = r$, then under the same set of condition, we have convergence at a linear rate*

$$\left\|F_{t+\mathcal{T}} F_{t+\mathcal{T}}^\top - X_\natural\right\| \leq \frac{c_6 \sigma_r}{\kappa}\left(1 - \frac{c_7}{\kappa^2}\right)^t, \quad t \geq 0.$$

We sketch the high-level ideas of the proof in Section 3.3, and postpone all the details to Appendix A. In the last, we show that the conditions of Theorem 3.1 can be met with our model assumptions in Section 2, our initialization scheme in Section 3.4, and our stepsize choice (3.2), with sufficient number of samples $m$ and proper sparsity level $p$ of the corruption $s$. Define $\theta_\epsilon$ to be the left quantile of the folded normal distribution $|Z|, Z \sim N(0,1)$, i.e., $\mathbb{P}(Z \leq \theta_\epsilon) = \epsilon$. The quantile function is used for the stepsize condition. For both RC and AC models, we now state our results as follows.

**Theorem 3.2.** *Suppose under Model 1, we have $m \geq c_1 dk^3 \kappa^{12}(\log \kappa + \log k) \log d$ and $p \leq \frac{c_2}{\kappa^3 \sqrt{k}}$ for some universal constants $c_1, c_2$. And under Model 2, we have fixed $p < \frac{1}{2}$ and $m \geq c_3 dk^3 \kappa^{12}(\log \kappa + \log k) \log d$ for some $c_3 > 0$ depending only on $p$. Then under both models, with probability at least $1 - c_4 \exp(-c_5 \frac{m}{\kappa^{12} k^2}) - \exp(-(pm+d))$ for some universal constants $c_4, c_5$, our subgradient method (3.1) with the initialization in Algorithm 1, and the adaptive stepsize choice (3.2) with $C_\eta \in \left[\frac{c_6}{\theta_{1-\frac{0.5-p}{3}}}\frac{\sigma_r}{\sigma_1^2}, \frac{c_7}{\theta_{1-\frac{0.5-p}{3}}}\frac{\sigma_r}{\sigma_1^2}, \right]$ with some universal $c_6, c_7 \leq 0.001$, converges as stated in Theorem 3.1.*

The proof follows by verifying the three conditions in Theorem 3.1. The initialization scheme is verified in Proposition 3.3 in Section 3.4. The RDPP is verified in Proposition 2.2. We defer the verification of the validity of our stepsize rule (3.2) to Appendix D.

Note that the initialization condition in Theorem 3.1 is weaker than those in the literature [4, 5], which requires $F_0 F_0^\top$ to be close to $X_\natural$. Instead, we allow a constant multiple $c^*$ in the initialization condition (3.3). This means we only require $F_0 F_0^\top$ to be accurate in the direction of $X_\natural$ and allows a

crude estimate of the scale $\|X_\natural\|_F$. This in turn allows us to have $p$ close to $\frac{1}{2}$ in Theorem 3.2 for the RC Model.[10] If we require $p \lesssim \frac{1}{\kappa\sqrt{r}}$ in both models, which is assumed in [4,5], then the sample complexity can be improved to $O(dk^3\kappa^4(\log\kappa + \log k)\log d)$ and the convergence rate can also be improved. See Appendix G for more details.

### 3.3 A sketch of analysis for Theorem 3.1

In the following, we briefly sketch the high-level ideas of the proof for Theorem 3.1.

**Closeness of the subgradient dynamics to its smooth counterpart.** To get the main intuition of our analysis, for simplicity let us first consider the noiseless case, i.e., $s_i = 0$ for all $i$. Now the objective $f(F)$ in (1.2) has expectation $\mathbb{E}f(F) = \frac{1}{2}\mathbb{E}\left|\langle A_1, FF^\top - X_\natural\rangle\right| = \sqrt{\frac{1}{2\pi}}\|FF^\top - X_\natural\|_F$. Hence, when minimizing this expected loss, the subgradient update at $F_t$ with $F_tF_t^\top \neq X_\natural$ becomes

$$F_0 \in \mathbb{R}^{d\times k}; \quad F_{t+1} = F_t - \eta_t \nabla \mathbb{E}f(F) \mid_{F=F_t} = F_t - \eta_t\sqrt{\frac{2}{\pi}}\frac{(F_tF_t^\top - X_\natural)}{\|F_tF_t^\top - X_\natural\|_F}F_t, \ t = 0, 1, 2, \ldots.$$

If we always have $\eta_t\sqrt{\frac{2}{\pi}} = \gamma\|F_tF_t^\top - X_\natural\|_F$ for some $\gamma > 0$, then the above iteration scheme becomes the gradient descent for the smooth problem $\min_{F\in\mathbb{R}^{d\times k}}\|FF^\top - X_\natural\|_F^2$:

$$F_0 \in \mathbb{R}^{d\times k}; \qquad F_{t+1} = F_t - \gamma\left(F_tF_t^\top - X_\natural\right)F_t, \ t = 0, 1, 2, \ldots. \tag{3.4}$$

Recent work [15] showed that when no corruption presents, the gradient decent dynamic (3.4) converges to the exact rank solution in the rank overspecified setting. Thus, under the robustness setting, our major intuition of stepsize design for the subgradient methods is to ensure that subgradient dynamics (3.1) for finite samples mimic the gradient descent dynamic (3.4). To show this, we need $\eta_t g_t$ in (3.1) to approximate $\left(F_tF_t^\top - X_\natural\right)F_t$ up to some multiplicative constant $\gamma$, which is indeed ensured by the conditions *(ii)* and *(iii)* in Theorem 3.1. Detailed analysis appears in Appendix A.

**Three-stage analysis of the subgradient dynamics.** However, recall our condition *(i)* in Theorem 3.1 only ensures that the initial $F_0F_0^\top$ is close to the ground truth $X_\natural$ up to some constant scaling factor. Thus, different from the analysis for matrix sensing with no corruptions [15], here we need to deal with the issue where the initial $\left\|F_0F_0^\top - X_\natural\right\|$ could be quite large. To deal with this issue, we introduce extra technicalities by decomposing the iterate $F_t$ into two parts, and conduct a three-stage analysis of the iterates. More specifically, since $X_\natural$ is rank $r$ and PSD, let its SVD be

$$X_\natural = \begin{bmatrix} U & V \end{bmatrix}\begin{bmatrix} D_S^* & 0 \\ 0 & 0 \end{bmatrix}\begin{bmatrix} U & V \end{bmatrix}^\top, \tag{3.5}$$

where $D_S^* \in \mathbb{R}^{r\times r}$, $U \in \mathbb{R}^{d\times r}$, and $V \in \mathbb{R}^{d\times(d-r)}$. Here $U$ and $V$ are orthonormal matrices and complement to each other, i.e., $U^\top V = 0$. Thus, for any $F_t \in \mathbb{R}^{d\times r}$, we can always decompose it as

$$F_t = US_t + VT_t, \tag{3.6}$$

where $S_t = U^\top F_t \in \mathbb{R}^{r\times k}$ stands for the signal term and $T_t = V^\top F_t \in \mathbb{R}^{(d-r)\times k}$ stands for an error term. Based on this, we show the following (we refer to Appendix A for the details.):

Stage 1. We characterize the evolution of the smallest singular value $\sigma_r(S_t)$, showing that it will increase above a certain threshold, so that $F_tF_t^\top$ is moving closer to the scaling of $X_\natural$.

Stage 2. We show that the quantity $Q_t = \max\{\left\|S_tS_t^\top - D_S^*\right\|, \left\|S_tT_t^\top\right\|\}$ decrease geometrically. In the analysis, for first two stages, $T_t$ may not decrease, and the first two stages consist of the burn-in phase $\mathcal{T}$.

Stage 3. We control $E_t = \max\{\left\|S_tS_t^\top - D_S^*\right\|, \left\|S_tT_t^\top\right\|, \left\|T_tT_t^\top\right\|\}$, showing that $E_t \to 0$ at a rate $O(1/t)$ due to the slower convergence of $\left\|T_tT_t^\top\right\|$. This further implies that $\left\|F_tF_t^\top - X_\natural\right\|$ converges to 0 at a sublinear rate $O(1/t)$. For the case $k = r$, the convergence speeds up to linear due to a better control of $\left\|T_tT_t^\top\right\|$.

### 3.4 Initialization schemes

---

[10]The work [13] actually allows for $p$ to be close to 1 under a simpler RC model. The main reason is that they assume the scale information $\|X_\natural\|_F$ is known a priori and their guarantee is only approximate $\|FF^\top - X_\natural\|_F \lesssim \delta\log\delta$ for a rank-1 $X_\natural$. However, estimating $\|X_\natural\|_F$ accurately requires $p$ to be close to 0.

---
**Algorithm 1:** Spectral Initialization
---
**Input:** the sensing matrix $\{A_i\}_{i=1}^m$ and observations $\{y_i\}_{i=1}^m$

  Form $D = \frac{1}{m}\sum_{i=1}^m \operatorname{sign}(y_i)A_i$, and compute the top-$k$ eigenpairs of $D$.

  Collect the eigenvectors as $U \in \mathbb{R}^{d\times k}$, and the eigenvalues as the diagonals of $\Sigma \in \mathbb{R}^{k\times k}$.

  Compute $B = U(\Sigma_+)^{\frac{1}{2}}$ where $\Sigma_+$ is the nonnegative part of $\Sigma$.

  Compute the median $\xi_{\frac{1}{2}}(\{|y_i|\}_{i=1}^m)$, and set $\gamma = \xi_{\frac{1}{2}}(\{|y_i|\}_{i=1}^m)/(\sqrt{2/\pi}\theta_{\frac{1}{2}})$.

**Output:** $F_0 = \sqrt{\gamma}B$.
---

In this section, we introduce a spectral initialization method as presented in Algorithm 1. The main intuition is that the cooked up matrix $D$ in Algorithm 1 has an expectation $\mathbb{E}(D) = \psi(X)\frac{X}{\|X\|_{\mathrm{F}}}$, so that the obtained factor $BB^\top$ from $D$ should be pointed to the same direction as $X_\natural$ up to a scaling. We then multiply $B$ by a suitable quantity to produce an initialization $F_0$ on the same scale as $X_\natural$. More specifically, given that $\theta_\epsilon$ is the left quantile of folded normal distribution introduced before Theorem 3.2, we state our result as follows.

**Proposition 3.3.** *Let $F_0$ be the output of Algorithm 1. Fix constant $c_0 < 0.1$. For model 1 with $p \le \frac{\tilde{c}_0}{\kappa^2\sqrt{r}}$ where $\tilde{c}_0$ depends only on $c_0$, there exist constants $c_1, c_2, c_3$ depending only on $c_0$ such that whenever $m \ge c_1 dr\kappa^4(\log\kappa + \log r)\log d$, we have*

$$\|F_0 F_0^\top - c^* X_\natural\| \le c_0\sigma_r/\kappa. \tag{3.7}$$

*with probability (w.p.) at least $1 - c_2\exp(-\frac{c_3 m}{\kappa^4 r}) - \exp(-(pm+d))$, where $c^* = 1$. For Model 2 with a fixed $p < 0.5$ and $m \ge c_4 dr\kappa^4(\log\kappa + \log r)\log d$. Then (3.7) also holds for $c^* \in \left[\sqrt{\frac{1}{2\pi}} \cdot \frac{\theta_{\frac{0.5-p}{3}}}{\theta_{\frac{1}{2}}}, \sqrt{\frac{2}{\pi}} \cdot \frac{\theta_{1-\frac{0.5-p}{3}}}{\theta_{\frac{1}{2}}}\right]$ w.p. at least $1 - c_5\exp(-\frac{c_6 m}{\kappa^4 r})$, where the constants $c_4, c_5, c_6$ depend only on $p$ and $c_0$.*

The details of the proof can be found in Appendix E. It should be noted in (3.7) that our guarantee ensures $F_0 F_0^\top$ is close to $X_\natural$ up to a scalar multiple $c^*$, instead of the usual guarantees on $\|F_0 F_0^\top - X_\natural\|$ [4, 5]. If $p \lesssim \frac{1}{\sqrt{r}\kappa}$, we obtain $\|F_0 F_0^\top - X_\natural\| \le c_0\sigma_r$ in Proposition E.2 and the sample complexity $m$ improves to $dr\kappa^2\log d(\log\kappa + \log r)$.

## 4 Experiments

In this section, we provide numerical evidence to verify our theoretical discoveries on robust matrix recovery in the rank overspecified regime. First, for robust low-rank matrix recovery, Section 4.1 shows that the subgradient method with proposed diminishing stepsizes can $(i)$ exactly recover the underlying low-rank matrix from its linear measurements even in the presence of corruptions and the rank is overspecified, and $(ii)$ prevent overfitting and produce high-quality solution even in the highly over-parameterized regime $k = d$. Furthermore, we apply the strategy of diminishing stepsizes for robust image recovery with DIP [14, 16, 32] in Section 4.2, demonstrating its effectiveness in alleviating overfitting. The experiments are conducted with Matlab 2018a and Google Colab [33].

### 4.1 Robust recovery of low-rank matrices

**Data generation and experiment setup.** For any given $d$ and $r$, we generate the ground truth $X_\natural = F_\natural F_\natural^\top$ with entries of $F_\natural \in \mathbb{R}^{d\times r}$ *i.i.d.* from standard normal distribution, and then normalize $X_\natural$ such that $\|X_\natural\|_{\mathrm{F}} = 1$. Similarly, we generate the $m$ sensing matrices $A_1, \ldots, A_m$ to be GOE matrices discussed in Section 2.1. For any given corruption ratio $p$, we generate the corruption vector $s \in \mathbb{R}^m$ according to the AC model, by randomly selecting $\lfloor pm \rfloor$ locations to be nonzero and generating those entries from a *i.i.d.* zero-mean Gaussian distribution with variance 100. We then generate the measurement according to (1.1), i.e., $y_i = \langle A_i, X_\natural\rangle + s_i, i = 1, \ldots, m$, and we set $n = 100$, $p = 0.2$, and $m = 10nr$. We run the subgradient methods for $10^3$ iterations using the following different stepsizes: $(i)$ the constant stepsize $\eta_t = 0.1$ for all $t \ge 0$, $(ii)$ sublinear diminishing stepsizes[11] $\eta_t = \eta_0/t$, $(iii)$ geometrically diminishing stepsizes $\eta_t = \eta_0 \cdot 0.9^t$, $(iv)$

---
[11] For the two diminising stepsizes, we set $\eta_0 = 2$ when $k = r$ or $2r$, and $\eta_0 = 5$ when $k = d$.

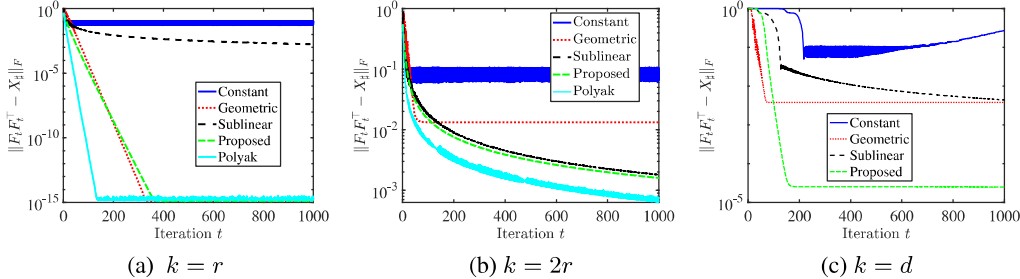

|  (a)  $k = r$  | (b) $k = 2r$ | (c) $k = d$ |

Figure 1: **Convergence of the subgradient method** (3.1) **with different stepsize rules** in the regimes of $(a)$ exact-parameterization $k = r$, $(b)$ rank overspecication $k = 2r$, and $(c)$ over-parameterization $k = d$. We use the following stepsizes: constant stepsizes $\eta_t = \eta_0$, geometric diminishing stepsizes $\eta_t = \eta_0 \cdot 0.9^t$, sublinear diminishing stepsizes $\eta_t = \eta_0/t$, the proposed stepsizes (3.2), and the Polyaks's stepsize.

the proposed stepsizes (3.2) with $C_\eta = \frac{1}{2}$, and $(v)$ the Polyak stepsize rule [34] which is given by $\eta_t = \frac{f(F) - f^\star}{\|g_t\|_{\mathrm{F}}^2}$, where $f^\star$ is the optimal value of (1.2) and $g_t$ is the subgradient in (3.1). We use the spectral initialization illustrated in Section 3.4 to generate the initialization $F_0$.

**Observations from the experimental results.** We run the subgradient method with different stepsize rules and different rank $k = r,\ 2r$, and $d$, and compute the reconstruction error $\|F_tF_t^\top - X_\natural\|_{\mathrm{F}}$ for the output of the algorithms. From Figure 2 and Figure 2a, our observations are the follows.

- *Exact rank case $k = r$.* As shown in Figure 1a, we can see that *(i)* the subgradient method with constant stepsizes does not converge to the target matrix, *(ii)* using sublinear diminishing stepsizes results in a sublinear convergence rate, *(iii)* geometrically diminishing or Polyak[12] stepsizes converge at a linear rate, consistent with the observation in [5], and *(iv)* the proposed stepsize rule also leads to convergence to the ground-truth at a linear rate, which is consistent with Theorem 3.1.

- *Overspecified rank case $k = 2r$.* As shown in Figure 2b, the subgradient method with any stepsize rule converges at most a sublinear rate, while constant or geometrically diminishing stepsizes result in poor solution accuracy. The proposed stepsize rule achieves on par convergence performance with the sublinear stepsize rule, which also demonstrates our Theorem 3.1 on the sublinear convergence in the rank overspecified setting. As shown in Figure 2a, the proposed strategy (3.2) gives stepsizes similar to Polyak's, and thus both resulting in a similar convergence rate (or the Polyak's is slightly better). However, the Polyak stepsize rule requires knowing the optimal value of the objective function, which is usually unknown a prior in practice.

- *Overparameterized case $k = d$.* The results are shown in Figure 1c.[13] In this case, inspired by [13, 27], we use an alternative tiny initialization $F_0$ by drawing its entries from *i.i.d.* zero-mean Gaussian distribution with standard deviation $10^{-7}$. We first note that the constant stepsize rule results in overfitting. In contrast, the proposed diminishing stepsizes can prevent overfitting issues and find a very high-quality solution. We conjecture this is due to certain implicit regularization effects [14, 26, 27] and we leave thorough theoretical analysis as future work.

### 4.2 Robust recovery of natural images with deep image prior

Finally, we conduct an exploratory experiment on robust image recovery with DIP [14, 16]. Here, the goal is to recover a natural image $X_\natural$ from its corrupted measurement $y = X_\natural + s$, where $s$ denotes noise corruptions. To achieve this, the DIP fits the observation by a highly overparameterized deep U-Net[14] $\phi(\Theta)$ by solving $\min_\Theta \|y - \phi(\Theta)\|_\square$ [37], where $\Theta$ denotes network parameters and the norm $\|\cdot\|_\square$ can be either $\ell_2$ or $\ell_1$ depending on the type of noises. As shown in [14, 16, 32], by learning the parameters $\Theta$, the network first reveals the underlying clear image and then *overfits* due to overparameterization, so that in many cases *early stopping* is *crucially* needed for its success.

In contrast to the constant stepsize rule, here we show a surprising result that, for sparse corruptions, optimizing a robust loss using the subgradient method with a diminishing stepsize rule does not

---

[12]As guaranteed by Theorem 2.3, in this case, the optimal value $f^\star$ is achieved at $F_\natural$.

[13]We omit the performance of the Polyak stepsizes since in this case the optimal value $f^\star$ is not easy to compute; it may not be achieved at $F_\natural$ due to the overfitting issue. Our experiments indicate that the Polyak stepsizes do not perform well when computed by setting the optimal value as either $0$ or the value at $F_\natural$.

[14]Following [16] (license obtained from [35]), we use the same U-shaped architecture with skip connections, each layer containing a convolution, a nonlinear LeakyReLU and a batch normalization units, Kaiming initialization for network parameters, and Adam optimizer [36]. We set the network width as 192 as used in [14].

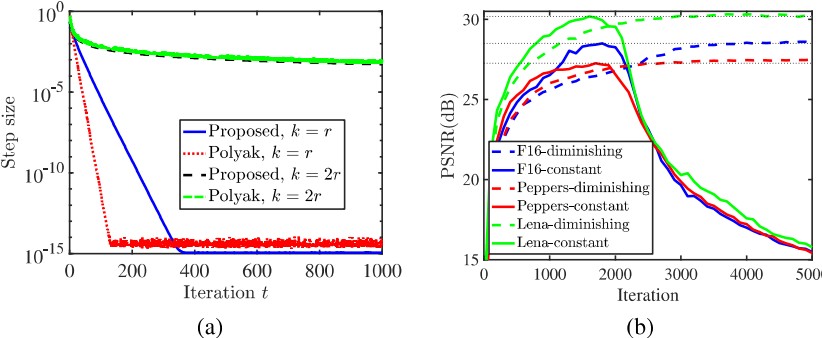

(a)                                      (b)

Figure 2: Figure 2a shows the **comparison of the proposed stepsize in** (3.2) **and the Polyak stepsize** in the regimes of the exact-parameterization $k = r$ and the rank overspecification $k = 2r$. Figure 2b shows the **learning curves for robust image recovery with DIP** on different test images with 50% salt-and-pepper noise. Here, we compare the constant and diminishing stepsize rules for DIP.

overfit. In particular, we consider a sparse corruption $s$ by impulse salt-and-pepper noise, and solve the robust DIP problem $\min_\Theta \|y - \phi(\Theta)\|_1$ using the subgradient method with a diminishing stepsize rule. We test the performance using three images, "F16", "Peppers" and "Lena" from a standard dataset [38], where we contaminate the image with salt-and-pepper noise by randomly replacing 50% pixels with either 1 or 0 (each happens with 50% probability). We compare the subgradient method with constant stepsize 0.01 (or learning rate) and diminishing stepsize (we use an initial stepsize 0.05 which is then scaled by 0.96 for every 50 iterations). As shown in **??**, the subgradient method with constant stepsize eventually overfits so that it requires early stopping. In comparison, optimization with diminishing stepsize prevents overfitting in the sense that the performance continues to improve as training proceeds, and it achieves on par performance with that of constant stepsizes when the learning is early stopped at the optimal PSNR. However, early stopping is not easy to implement in practice without knowing the clean image *a priori*.

## 5    Discussion

In this work, we designed and analyzed a subgradient method for the robust matrix recovery problem under the rank overspecified setting, showing that it converges to the exact solution with a sublinear rate. Based on our results, there are several directions worth further investigation.

- *Better sample complexity.* Our exact recovery result in Theorem 3.2 requires $\tilde{\mathcal{O}}(dk^3)$ many samples. When the rank is exactly specified ($k = r$), for factor approaches with corrupted measurements, the best sample complexity result is $\tilde{\mathcal{O}}(dr^2)$ [2, 4]. Hence there is an extra $k$ factor in the rank overspecified setting. Is it possible to improve the dependence on $k$? In particular, we think the worse dependence comes from the $\delta^{-4}$ dependence of the sample complexity of RDPP in Proposition 2.2. More specifically, the Cauchy-Schwartz inequality we applied in (C.31) might make the dependence on $\delta$ not tight.
- *Implicit regularization.* As explored and demonstrated in our experiments in Section 4, our stepsize choice combined with random initialization recovers $X_\natural$ with high accuracy, even if $k \geq r$ and the number of samples $m$ is way less than the degree of freedom (i.e., $m = O(dr)$). Theoretically justifying this phenomenon in the overparameterized regime would be of great interest.
- *Other sensing matrix.* In this work, we assume $A_i$ to be a GOE matrix. This specific assumption on the measurement matrix still seems to be restrictive. Generalizing our results to more generic sensing matrices would also be an interesting future direction.
- *Rectangular matrices.* In this work, we utilize the fact that $X_\natural$ is PSD, and optimize over one factor $F$. This is consistent with much existing analysis with rank overspecification [13, 15, 27]. For rectangular $X_\natural$, we need to optimize over two factors $U, V$. How to combine the analysis here with an extra regularization $\|UU^\top - VV^\top\|_F^2$ in the objective is another interesting direction to pursue.

## Acknowledgement

L. Ding would like to thank Jiacheng Zhuo for inspiring discussions. Y. Chen is partially supported by NSF grant CCF-1704828 and CAREER Award CCF-2047910. Z. Zhu is partially supported by NSF grants CCF-2008460 and CCF-2106881.

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
