# A Analysis of algorithm under conditions of Theorem 3.1

Here we recall some basic setup introduced in the sketch of analysis in Section 3.3. Recall the singular value decomposition of $X_\natural$ is

$$X_\natural = [U \quad V] \begin{bmatrix} D_S^* & 0 \\ 0 & 0 \end{bmatrix} [U \quad V]^\top, \tag{A.1}$$

where $U \in \mathbb{R}^{d \times r}, V \in \mathbb{R}^{d \times d-r}, D_S^* \in \mathbb{R}^{r \times r}$. $U$ and $V$ has orthonormal columns and $U^\top V = 0$. The $i$-th largest singular values of $X_\natural$ is denoted as $\sigma_i$. Thus, $\sigma_1$ and $\sigma_r$ are the largest and smallest diagonal entries of $D_S^*$ respectively. Since $X_\natural$ is assumed to have rank $r$, we have $\sigma_{r+1} = \sigma_{r+2} = \ldots = \sigma_d = 0$. The condition number is defined to be $\kappa = \frac{\sigma_1}{\sigma_r}$. Since union of column space of $U$ and $V$ spans the whole space, for any $F_t \in \mathbb{R}^{d \times r}$, we can write

$$F_t = US_t + VT_t, \tag{A.2}$$

where $S_t = U^\top F_t \in \mathbb{R}^{r \times k}$ and $T_t = V^\top F_t \in \mathbb{R}^{(d-r) \times k}$.

We now formalize the idea of closeness of subgradient dynamics to its smooth counter part described in Section 3.3. By assumption (iii) in Theorem 3.1, the RDPP holds with parameters $(k + r, \sqrt{\frac{1}{2\pi}}\delta)$ and $\delta = \frac{c}{\kappa^3 \sqrt{k}}$ for some small constant $c$ depending on $c_3$ in Theorem 3.1. Since the RDPP holds, let

$$\gamma_t = \frac{\eta_t \psi(F_t F_t^\top - X_\natural)}{\|F_t F_t^\top - X_\natural\|_F}, \quad D_t \in D(F_t F_t^\top - X_\natural), \tag{A.3}$$

we have

$$\left\| \eta_t D_t - \gamma_t (F_t F_t^\top - X_\natural) \right\|_{F,k+r} \leq \eta_t \sqrt{\frac{1}{2\pi}}\delta \tag{A.4}$$

$$\leq \eta_t \psi(F_t F_t^\top - X_\natural)\delta \tag{A.5}$$

$$= \delta \gamma_t \|F_t F_t^\top - X_\natural\|_F. \tag{A.6}$$

Define the following shorthand $\Delta_t$,

$$\Delta_t = \frac{\eta_t}{\gamma_t} D_t - (F_t F_t^\top - X_\natural). \tag{A.7}$$

Then (A.4) becomes

$$\|\Delta_t\| \leq \delta \|F_t F_t^\top - X_\natural\|_F \overset{(a)}{\leq} \delta\sqrt{k+r} \left\| F_t F_t^\top - X_\natural \right\|. \tag{A.8}$$

Here step $(a)$ is because $F_t F_t^\top - X_\natural$ has rank no more than $k + r$.

Using that fact that the subgradient we used in algorithm 3.1 can be written as $g_t = D_t F_t$, we have

$$F_{t+1} = F_t - \gamma_t (F_t F_t^\top - X_\natural)F_t + \gamma_t \Delta_t F_t. \tag{A.9}$$

Note that if we ignore the error term $\gamma_t \Delta_t F_t$ in (A.9), the update equation becomes

$$\tilde{F}_{t+1} = F_t - \gamma_t \left( F_t F_t^\top - X_\natural \right) F_t. \tag{A.10}$$

This update is the update of gradient descent for the smooth function $\tilde{f}(F) = \frac{1}{4}\|FF^\top - X_\natural\|_F^2$ with stepsize $\gamma_t$. We will refer (A.10) as the "population-level" update and we will leverage the properties of this update throughout the analysis. We are now ready to start our full analysis of the subgradient dynamics. We first characterize the initialization quality in terms of $S$ and $T$.

**Proposition A.1** (Initialization quality). *Under the condition on $F_0$ sated in (3.3) of Theorem 3.1 we have*

$$\sigma_r(S_0) \geq \frac{\sqrt{\epsilon \sigma_r}}{2}, \tag{A.11}$$

$$\|T_0\| \leq \min\{\frac{\sigma_r}{200\sqrt{\sigma_1}}, \frac{\sqrt{\epsilon \sigma_r}}{40}\}, \tag{A.12}$$

$$\|S_0\| \leq 2\sqrt{\sigma_1}, \tag{A.13}$$

$$4\|T_0\|^2 \leq 0.001\sigma_r(S_0)\frac{\sigma_r}{\sqrt{\sigma_1}}. \tag{A.14}$$

In the analysis, we denote $\sigma_r(S_0)$ by $\rho$ and let $c_\rho = \min\{\frac{\sigma_r}{200\sqrt{\sigma_1}\rho}, \frac{1}{20}\}$. Then we have

$$\sigma_r(S_0) = \rho \geq \frac{\sqrt{\epsilon\sigma_r}}{2} \tag{A.15}$$

$$\|T_0\| \leq c_\rho\rho \tag{A.16}$$

$$\|S_0\| \leq 2\sqrt{\sigma_1}. \tag{A.17}$$

The parameters satisfy $4(c_\rho\rho)^2 \leq 0.001\rho\frac{\sigma_r}{\sqrt{\sigma_1}}$ and $c_\rho\rho \leq \min\{0.1\sqrt{\sigma_1}, \frac{\sigma_r}{200\sqrt{\sigma_1}}\} = \frac{\sigma_r}{200\sqrt{\sigma_1}}$, which will be applied multiple times in the following analysis.

The next proposition illustrates the evolution of $S_t$ and $T_t$.

**Proposition A.2** (Updates of $S_t, T_t$). *For any $t \geq 0$, we have*

$$S_{t+1} = S_t - \gamma_t\left(S_t S_t^\top S_t + S_t T_t^\top T_t - D_S^* S_t\right) + \gamma_t U^\top \Delta_t F_t, \tag{A.18}$$

$$T_{t+1} = T_t - \gamma_t\left(T_t T_t^\top T_t + T_t S_t^\top S_t\right) + \gamma_t V^\top \Delta_t F_t. \tag{A.19}$$

We introduce notations

$$\mathcal{M}_t(S_t) = S_t - \gamma_t\left(S_t S_t^\top S_t + S_t T_t^\top T_t - D_S^* S_t\right) \tag{A.20}$$

$$\mathcal{N}_t(T_t) = T_t - \gamma_t\left(T_t T_t^\top T_t + T_t S_t^\top S_t\right). \tag{A.21}$$

They are "population-level" updates for $S_t$ and $T_t$.

**Proposition A.3** (Uniform upper bound). *Suppose $\gamma_t$ satisfies $\gamma_t \leq \frac{0.01}{\sigma_1}$ for all $t \geq 0$ and $(50\sqrt{k}\delta)^{\frac{1}{3}} \leq \frac{c_\rho\rho}{2\sqrt{\sigma_1}} = \frac{\sigma_r}{400\sigma_1}$, we have*

$$\|T_t\| \leq c_\rho\rho \leq 0.1\sqrt{\sigma_r} \leq 0.1\sqrt{\sigma_1} \tag{A.22}$$

$$\|S_t\| \leq 2\sqrt{\sigma_1} \tag{A.23}$$

*for all $t \geq 0$.*

The analysis of algorithm consists of three stages:

- In stage 1, we show at $\sigma_r(S_t)$ increases geometrically to level $\sqrt{\frac{\sigma_r}{2}}$ by time $\mathcal{T}_1'$, then $\left\|S_t S_t^\top - D_S^*\right\|$ will decrease geometrically to $\frac{100(c_\rho\rho)^2\sigma_1}{\sigma_r}(\leq \frac{\sigma_r}{100})$ by $\mathcal{T}_1$. The iterate will then enter a good region.

- In stage 2, we show that $D_t = \max\{\left\|S_t S_t^\top - D_S^*\right\|, \left\|S_t T_t^\top\right\|\}$ decreases geometrically if it is bigger than $10\delta\sqrt{2k}\sigma_1$, which is the computational threshold. In other words, $\left\|S_t S_t^\top - D_S^*\right\|$ decrease to a $\frac{100(c_\rho\rho)^2\sigma_1}{\sigma_r}$ geometrically, and this will happen by $\mathcal{T}_2$.

- In stage 3, after $\mathcal{T}_2$, $E_t = \max\{\left\|S_t S_t^\top - D_S^*\right\|, \left\|S_t T_t^\top\right\|, \left\|T_t T_t^\top\right\|\}$ converges to 0 sublinearly.

In the above statement,

$$\mathcal{T}_1' = \left\lceil \log\left(\frac{\sqrt{\sigma_r}}{\sqrt{2}\rho}\right) / \log\left(1 + \frac{c_\gamma\sigma_r^2}{6\sigma_1^2}\right) \right\rceil, \tag{A.24}$$

$$\mathcal{T}_1 = \mathcal{T}_1' + \left\lceil \frac{\log(\frac{20(c_\rho\rho)^2}{\sigma_r})}{\log\left(1 - \frac{c_\gamma\sigma_r^2}{2\sigma_1^2}\right)} \right\rceil \tag{A.25}$$

and

$$\mathcal{T}_2 = \mathcal{T}_1 + \left\lceil \frac{\log\left(\frac{1000\delta\sqrt{k}\sigma_1}{\sigma_r}\right)}{\log\left(1 - \frac{c_\gamma\sigma_r^2}{6\sigma_1^2}\right)} \right\rceil. \tag{A.26}$$

Stage 1 consists of all the iterations up to time $\mathcal{T}_1$. Stage 2 consists of all the iterations between $\mathcal{T}_1 + 1$ and $\mathcal{T}_2$. Stage 3 consists of all the iterations afterwards.

## A.1 Analysis of $\mathcal{M}_t(S_t)$ and $\mathcal{N}_t(T_t)$.

In this sections we prove some facts about $\mathcal{M}_t(S_t)$ and $\mathcal{N}_t(T_t)$ that will be useful in the analysis.

**Proposition A.4.** *Suppose* $\gamma_t \leq \min\{\frac{0.01}{\sigma_1}, \frac{0.01\sigma_r}{\sigma_1^2}\}$, $\|S_t\| \leq 2\sqrt{\sigma_1}$, $\sigma_r(S_t) \geq \sqrt{\frac{\sigma_r}{2}}$ *and* $\|T_t\| \leq 0.1\sqrt{\sigma_r}$, *we have the following:*

1. $\left\| \mathcal{M}_t(S_t)\mathcal{M}_t(S_t)^\top - D_S^* \right\| \leq \left(1 - \frac{3\gamma_t\sigma_r}{4}\right)\left\| S_t S_t^\top - D_S^* \right\| + 3\gamma_t \left\| S_t T_t^\top \right\|^2.$

*Suppose* $\gamma_t \leq \frac{0.01}{\sigma_1}$, $\|S_t\| \leq 2\sqrt{\sigma_1}$, *and* $\|T_t\| \leq 0.1\sqrt{\sigma_r}$, *we have the following:*

2. $\left\| \mathcal{N}_t(T_t)\mathcal{N}_t(T_t)^\top \right\| \leq \|T_t\|^2 \left(1 - \frac{3\gamma_t}{2}\|T_t\|^2\right) = \left\| T_t T_t^\top \right\|\left(1 - \frac{3\gamma_t}{2}\left\| T_t T_t^\top \right\|\right).$

*Furthermore, suppose* $\gamma_t \leq \frac{0.01}{\sigma_1}$, $\|S_t\| \leq 2\sqrt{\sigma_1}$, $\sigma_r(S_t) \geq \sqrt{\frac{\sigma_r}{2}}$, $\|T_t\| \leq 0.1\sqrt{\sigma_r}$, *and* $\left\| S_t S_t^\top - D_S^* \right\| \leq \frac{\sigma_r}{10}$, *we have same inequalities as 1, 2 and*

3. $\left\| \mathcal{M}_t(S_t)\mathcal{N}_t(T_t)^\top \right\| \leq \left(1 - \frac{\gamma_t\sigma_r}{3}\right)\left\| S_t T_t^\top \right\|.$

**Proposition A.5.** *Suppose* $\gamma_t \leq \frac{0.01}{\sigma_1}$, $\|S_t\| \leq 2\sqrt{\sigma_1}$, $\sigma_r(S_t) \geq \sqrt{\frac{\sigma_r}{2}}$ *and* $\|T_t\| \leq 0.1\sqrt{\sigma_r}$, *we have the following:*

1. $\left\| D_S^* - \mathcal{M}_t(S_t)S_t^\top \right\| \leq \left(1 - \frac{\gamma\sigma_r}{2}\right)\left\| D_S^* - S_t S_t^\top \right\| + \gamma_t \left\| S_t T_t^\top \right\|^2.$

2. $\left\| \mathcal{M}_t(S_t)T_t^\top \right\| \leq 2\left\| S_t T_t^\top \right\|.$

3. $\left\| \mathcal{N}_t(T_t)S_t^\top \right\| \leq \left\| T_t S_t^\top \right\|.$

4. $\left\| \mathcal{N}_t(T_t)T_t^\top \right\| \leq \|T_t\|^2\left(1 - \gamma_t\|T_t\|^2\right) = \left\| T_t T_t^\top \right\|\left(1 - \gamma_t\left\| T_t T_t^\top \right\|\right).$

## A.2 Analysis of Stage 1

The following proposition characterize the evolution of $\sigma_r(S_t)$. In stage one, we start with a initialization satisfies conditions in Proposition A.1.

**Proposition A.6.** *Suppose there is some constant* $c_\gamma > 0$ *such that the parameters satisfy* $\frac{c_\gamma\sigma_r}{\sigma_1^2} \leq \gamma_t \leq \min\{\frac{1}{100\sigma_1}, \frac{\sigma_r}{100\sigma_1^2}\} = \frac{\sigma_r}{100\sigma_1^2}$, $(50\sqrt{k}\delta)^{\frac{1}{3}} \leq \frac{c_\rho\rho}{2\sqrt{\sigma_1}} = \frac{\sigma_r}{400\sigma_1}$, *we have*

$$\sigma_r(S_t) \geq \min\left\{\left(1 + \frac{\sigma_r^2 c_\gamma}{6\sigma_1^2}\right)^t \sigma_r(S_0), \sqrt{\frac{\sigma_r}{2}}\right\} \tag{A.27}$$

*for all* $t \geq 0$. *In particular, we have*

$$\sigma_r(S_{\mathcal{T}_1'} + t) \geq \sqrt{\frac{\sigma_r}{2}}. \tag{A.28}$$

*for all* $t \geq 0$.

Next, we show that $\left\| S_t S_t^\top - D_S^* \right\|$ decays geometrically to $\frac{100(c_\rho\rho)^2\sigma_1}{\sigma_r}$.

**Proposition A.7.** *Suppose there is some constant* $c_\gamma > 0$ *such that the parameters satisfy* $\frac{c_\gamma\sigma_r}{\sigma_1^2} \leq \gamma_t \leq \min\{\frac{1}{100\sigma_1}, \frac{\sigma_r}{100\sigma_1^2}\} = \frac{\sigma_r}{100\sigma_1^2}$, $(50\sqrt{k}\delta)^{\frac{1}{3}} \leq \frac{c_\rho\rho}{2\sqrt{\sigma_1}}$, *we have for any* $t \geq 0$, *we have*

$$\left\| S_{\mathcal{T}_1+t}S_{\mathcal{T}_1+t}^\top - D_S^* \right\| \leq \max\{5\sigma_1\left(1 - \frac{c_\gamma\sigma_r^2}{2\sigma_1^2}\right)^t, \frac{100(c_\rho\rho)^2\sigma_1}{\sigma_r}\}. \tag{A.29}$$

*In particular, for* $\mathcal{T}_1 = \mathcal{T}_1' + \left\lceil \frac{\log(\frac{20(c_\rho\rho)^2}{\sigma_r})}{\log\left(1 - \frac{c_\gamma\sigma_r^2}{2\sigma_1^2}\right)} \right\rceil$, *we have*

$$\left\| S_t S_t^\top - D_S^* \right\| \leq \frac{100(c_\rho\rho)^2\sigma_1}{\sigma_r} \leq \frac{\sigma_r}{100}, \qquad \forall t \geq \mathcal{T}_1. \tag{A.30}$$

When $t = \mathcal{T}_1$, by Proposition A.7,

$$\left\| D_S^* - S_{\mathcal{T}_1} S_{\mathcal{T}_1}^\top \right\| \leq \frac{\sigma_r}{100}. \tag{A.31}$$

By Proposition A.3 and our assumption that $c_\rho \rho \leq \frac{\sigma_r}{200\sqrt{\sigma_1}}$,

$$\left\| S_{\mathcal{T}_1} T_{\mathcal{T}_1} \right\| \leq 2(c_\rho \rho)\sqrt{\sigma_1} \leq \frac{\sigma_r}{100}. \tag{A.32}$$

Combining, we obtain

$$D_{\mathcal{T}_1} \leq \frac{\sigma_r}{100}. \tag{A.33}$$

### A.3 Analysis of Stage 2

Recall

$$D_t = \max\{\left\| S_t S_t^\top - D_S^* \right\|, \left\| S_t T_t^\top \right\|\}. \tag{A.34}$$

We show that $D_t$ decreases to $10\delta\sqrt{k+r}\sigma_1$ geometrically after $\mathcal{T}_1$.

**Proposition A.8.** *Suppose there is some constant $c_\gamma > 0$ such that the parameters satisfy $\frac{c_\gamma \sigma_r}{\sigma_1^2} \leq \gamma_t \leq \frac{0.01}{\sigma_1}$, $\delta\sqrt{k+r} \leq \frac{0.001\sigma_r}{\sigma_1}$. Also, we suppose $\|T_t\| \leq 0.1\sqrt{\sigma_r}$ for all $t$. If for some $\mathcal{T}_1 > 0$,*

$$D_{\mathcal{T}_1} \leq \max\{\frac{\sigma_r}{100}, 10\delta\sqrt{k+r}\sigma_1\} = \frac{\sigma_r}{100}, \tag{A.35}$$

*then for any $t \geq 0$, we have*

$$D_{\mathcal{T}_1+t} \leq \max\left\{\left(1 - \frac{c_\gamma \sigma_r^2}{6\sigma_1^2}\right)^t \cdot \frac{\sigma_r}{100}, 10\delta\sqrt{k+r}\sigma_1\right\}. \tag{A.36}$$

*In particular, for $\mathcal{T}_2 = \mathcal{T}_1 + \left\lceil \frac{\log\left(\frac{1000\delta\sqrt{k+r}\sigma_1}{\sigma_r}\right)}{\log\left(1 - \frac{c_\gamma \sigma_r^2}{6\sigma_1^2}\right)} \right\rceil$, we have*

$$\left\| S_t S_t^\top - D_S^* \right\| \leq 10\delta\sqrt{k+r}\sigma_1, \tag{A.37}$$
$$\left\| S_t T_t^\top \right\| \leq 10\delta\sqrt{k+r}\sigma_1, \qquad \forall t \geq \mathcal{T}_2. \tag{A.38}$$

### A.4 Analysis of Stage 3

Define

$$E_t = \max\{\left\| S_t S_t^\top - D_S^* \right\|, \left\| S_t T_t^\top \right\|, \left\| T_t T_t^\top \right\|\}. \tag{A.39}$$

We are going to show the sublinear convergence of $E_t$ in stage three.

**Proposition A.9.** *Suppose we have $\gamma_t \leq \frac{0.01}{\sigma_1}$, $\delta\sqrt{k+r} \leq \frac{0.001\sigma_r}{\sigma_1}$ and $E_t \leq 0.01\sigma_r$ for some $t > 0$. Then we have*

$$E_{t+1} \leq \max\{(1 - \frac{\gamma_t \sigma_r}{6})E_t, E_t(1 - \gamma_t E_t)\} = E_t(1 - \gamma_t E_t). \tag{A.40}$$

Indeed, we can prove a better rate if there is no overparametrization.

**Proposition A.10.** *Suppose we have $\gamma_t \leq \frac{0.01}{\sigma_1}$, $\delta\sqrt{k+r} \leq \frac{0.001\sigma_r}{\sigma_1}$ and $E_t \leq 0.01\sigma_r$ for some $t > 0$. If $k = r$, then we have*

$$E_{t+1} \leq (1 - \frac{\gamma_t \sigma_r}{3})E_t. \tag{A.41}$$

### A.5 Proof of Theorem 3.1

The proof is a combination of all the propositions in this section. First, we show that under suitable choice of $c_0$ and $c_3$, all the assumptions are satisfied. First, if we take $c_3$ to be small enough, we know

that $(50\sqrt{k}\delta)^{\frac{1}{3}} \le \frac{\sigma_r}{400\sigma_1}$ holds. Hence, all the conditions related to $\delta$ are satisfied. Next, by definition, $\gamma_t = \frac{\eta_t \psi(F_t F_t^\top - X_\natural)}{\|F_t F_t^\top - X_\natural\|_F}$. By the second assumption and the assumption on range of $\psi$, we know

$$\gamma_t \in [c_1\sqrt{\frac{1}{2\pi}}\frac{\sigma_r}{\sigma_1^2}, c_2\sqrt{\frac{2}{\pi}}\frac{\sigma_r}{\sigma_1^2}]. \tag{A.42}$$

Since we assumed $c_2 \le 0.01$, so the step size condition $\gamma_t \le \frac{\sigma_r}{100\sigma_1^2}$ is satisfied. Moreover, $c_\gamma \ge c_1\sqrt{\frac{1}{2\pi}}$. Now, applying theorems for initialization, stage 1 and stage 2, we know that

$$\left\|S_{\mathcal{T}_2}S_{\mathcal{T}_2}^\top - D_S^*\right\| \le 10\delta\sqrt{k+r}\sigma_1 \le \frac{0.01\sigma_r^2}{\sigma_1}, \tag{A.43}$$

$$\left\|S_{\mathcal{T}_2}T_{\mathcal{T}_2}^\top\right\| \le 10\delta\sqrt{k+r}\sigma_1 \le \frac{0.01\sigma_r^2}{\sigma_1}. \tag{A.44}$$

In addition, by Proposition A.3, we know

$$\left\|T_{\mathcal{T}_2}T_{\mathcal{T}_2}^\top\right\| = \|T_{\mathcal{T}_2}\|^2 \le (c_\rho\rho)^2 \le \frac{0.01\sigma_r^2}{\sigma_1}. \tag{A.45}$$

Hence, $E_{\mathcal{T}_2} \le \frac{0.01\sigma_r^2}{\sigma_1}$. Here are two cases:

- $k > r$, By Proposition A.9 and induction, we know

$$E_{t+1} \le E_t(1 - \gamma_t E_t) \le E_t(1 - \frac{c_\gamma\sigma_r}{\sigma_1^2}E_t), \qquad \forall t \ge \mathcal{T}_2. \tag{A.46}$$

  where $c_1\sqrt{\frac{2}{\pi}} \le c_\gamma \le 0.01$. Define $G_t = \frac{c_\gamma\sigma_r}{\sigma_1^2}E_t$, then we have $G_{\tau_2} < 1$ and

$$G_{t+1} \le G_t(1 - G_t), \qquad \forall t \ge \mathcal{T}_2. \tag{A.47}$$

  Taking reciprocal, we obtain

$$\frac{1}{G_{t+1}} \ge \frac{1}{G_t} + \frac{1}{1 - G_t} \ge \frac{1}{G_t} + 1, \qquad \forall t \ge \mathcal{T}_2 \tag{A.48}$$

  So we obtain

$$G_{\mathcal{T}_2+t} \le \frac{1}{\frac{1}{G_{\mathcal{T}_2}} + t}, \qquad \forall t \ge 0. \tag{A.49}$$

  Plugging in the definition of $G_t$, we obtain

$$E_{\tau_2+t} \le \frac{\sigma_1^2}{c_\gamma\sigma_r}\frac{1}{\frac{\sigma_1^2}{c_\gamma\sigma_r E_{\mathcal{T}_2}} + t} \le \frac{\sigma_1^2}{c_\gamma\sigma_r}\frac{1}{\frac{100\sigma_1^3}{c_\gamma\sigma_r^3} + t} = \frac{\sigma_1}{c_\gamma}\frac{\kappa}{\frac{100}{c_\gamma}\kappa^3 + t} \le \frac{\sigma_1}{c_\gamma}\frac{\kappa}{\kappa^3 + t}. \tag{A.50}$$

  Since $c_\gamma \ge c_1\sqrt{\frac{2}{\pi}}$, we can simply take $c_5 = \frac{1}{4c_1}\sqrt{\frac{\pi}{2}}$, $\mathcal{T} = \mathcal{T}_2$, apply Lemma I.5, and get

$$\left\|F_{\mathcal{T}+t}F_{\mathcal{T}+t}^\top - X_\natural\right\| \le c_5\sigma_1\frac{\kappa}{\kappa^3 + t}, \qquad \forall t \ge 0. \tag{A.51}$$

The last thing to justify is $\mathcal{T}_2 \lesssim \kappa^2\log\kappa$. Recall

$$\mathcal{T}_1' = \left\lceil \log\left(\frac{\sqrt{\sigma_r}}{\sqrt{2}\rho}\right) \Big/ \log\left(1 + \frac{c_\gamma\sigma_r^2}{6\sigma_1^2}\right) \right\rceil, \tag{A.52}$$

$$\mathcal{T}_1 = \mathcal{T}_1' + \left\lceil \frac{\log(\frac{20(c_\rho\rho)^2}{\sigma_r})}{\log\left(1 - \frac{c_\gamma\sigma_r^2}{2\sigma_1^2}\right)} \right\rceil \tag{A.53}$$

and

$$\mathcal{T}_2 = \mathcal{T}_1 + \left\lceil \frac{\log\left(\frac{1000\delta\sqrt{k}\sigma_1}{\sigma_r}\right)}{\log\left(1 - \frac{c_\gamma\sigma_r^2}{6\sigma_1^2}\right)} \right\rceil. \tag{A.54}$$

Simple calculus yield that each integer above is $O(\kappa^2\log\kappa)$. So the proof is complete in overspecified case.

- $k = r$. By Proposition A.10 and induction, we obtain

$$E_{t+1} \leq (1 - \frac{\gamma_t \sigma_r}{3})E_t \leq (1 - \frac{c_\gamma \sigma_r^2}{\sigma_1^2})E_t, \forall t \geq \mathcal{T}_2. \tag{A.55}$$

Applying this inequality recursively and noting $c_\gamma \geq c_1 \sqrt{\frac{2}{\pi}}$, we obtain

$$E_{\mathcal{T}_2+t} \leq (1 - \frac{c_\gamma \sigma_r^2}{\sigma_1^2})^t E_{\mathcal{T}_2} \leq \left(1 - \frac{c_1\sqrt{\frac{2}{\pi}}}{\kappa^2}\right)^t \frac{0.01\sigma_r}{\kappa}, \forall t \geq 0. \tag{A.56}$$

Thus, we can take $c_6 = 0.01/4$, $c_7 = c_1\sqrt{\frac{2}{\pi}}$, $\mathcal{T} = \mathcal{T}_2$, apply Lemma I.5 and get

$$\left\| F_{\mathcal{T}+t}F_{\mathcal{T}+t}^\top - X_\natural \right\| \leq \frac{c_6 \sigma_r}{\kappa}\left(1 - \frac{c_7}{\kappa^2}\right)^t, \qquad \forall t \geq 0. \tag{A.57}$$

The validity of $\mathcal{T}$ is proved in the last part. The proof is complete.

# B Proof of Propositions

## B.1 Proof of Proposition A.1

First, we note that the $r$-th singular value of $c^* X_\natural$ is at least $\epsilon \sigma_r$. By almost the same proof as Lemma I.5, we get

$$\max\{\left\| S_0 S_0^\top - c^* D_S^* \right\|, \left\| S_0 T_0^\top \right\|, \left\| T_0 T_0^\top \right\|\} \leq \left\| F_0 F_0^\top - c^* X_\natural \right\| \leq \frac{\tilde{c}_0 \epsilon \sigma_r}{\kappa}. \tag{B.1}$$

We take $\tilde{c}_0 = \left(\frac{1}{200}\right)^2$. By Weyl's inequality (I.3),

$$\sigma_r(S_0 S_0^\top) \geq \sigma_r(c^* D_S^*) - \left\| S_0 S_0^\top - c^* D_S^* \right\| \geq \frac{c^* \sigma_r}{4} \geq \frac{\epsilon \sigma_r}{4}. \tag{B.2}$$

Hence, $\rho = \sigma_r(S_0) \geq \frac{\sqrt{\epsilon \sigma_r}}{2}$. On the other hand,

$$\|T_0\| \leq \sqrt{\frac{\tilde{c}_0 \epsilon \sigma_r}{\kappa}} \tag{B.3}$$

$$\leq \min\{\frac{\sigma_r}{200\sqrt{\sigma_1}}, \frac{\sqrt{\epsilon \sigma_r}}{40}\}. \tag{B.4}$$

We can simply assume $\sigma_1(S_0) \leq 2\sqrt{\sigma_1}$. If not so, we can normalize $F_0$ so that $\sigma_1(S_0) = 2\sqrt{\sigma_1}$ and use normalized $F_0$ as our initialization. By Weyl's inequality (I.3),

$$\sigma_1(S_0 S_0^\top) \leq 1.01 c^* \sigma_1. \tag{B.5}$$

Hence, $c^* \geq 3$. In this case, it is easy to show that $\rho = \sigma_r(S_0) \geq \frac{\sqrt{\epsilon \sigma_r}}{2}$ and $\|T_0\| \leq \frac{\sigma_r}{200\sqrt{\sigma_1}}$ still holds. Therefore, the initialization quality is proved.

## B.2 Proof of Proposition A.2

The algorithm A.9 updates $F_t$ by

$$F_{t+1} = F_t - \gamma_t \left(F_t F_t^\top - X_\natural\right) F_t + \gamma_t \Delta_t F_t. \tag{B.6}$$

Using the definition of $U, V, S_t, T_t$, we have

$$S_{t+1} = U^\top F_{t+1}$$
$$= U^\top F_t - \gamma_t U^\top \left(F_t F_t^\top - X_\natural\right) F_t + \gamma_t U^\top \Delta_t F_t$$
$$= U^\top (US_t + VT_t) - \gamma_t U^\top \left[(US_t + VT_t)(US_t + VT_t)^\top - UD_S^* U^\top - VD_T^* V^\top\right](US_t + VT_t)$$
$$\quad + \gamma_t U^\top \Delta_t F_t$$
$$\overset{(\natural)}{=} S_t - \gamma_t (S_t S_t^\top S_t + S_t T_t^\top T_t - D_S^* S_t) + \gamma_t U^\top \Delta_t F_t.$$

Here $(\natural)$ follows from the fact that $U^\top V = 0$ and $U, V$ are orthonormal.
By the same token, we can show

$$T_{t+1} = T_t - \gamma_t \left(T_t T_t^\top T_t + T_t S_t^\top S_t\right) + \gamma_t V^\top \Delta_t F_t. \tag{B.7}$$

## B.3 Proof of Proposition A.3

We prove the proposition by induction. By Proposition A.1, it's clear that the proposition holds for $t = 0$. Suppose for $t \geq 0$, we have

$$\|T_t\| \leq c_\rho \rho \leq 0.1\sqrt{\sigma_1} \tag{B.8}$$

$$\|S_t\| \leq 2\sqrt{\sigma_1}. \tag{B.9}$$

By Proposition A.2, we know

$$S_{t+1} = S_t - \gamma_t(S_t S_t^\top S_t + S_t T_t^\top T_t - D_S^* S_t) + \gamma_t U^\top \Delta_t F_t. \tag{B.10}$$

Since $\|T_t\| \leq c_\rho \rho \leq 0.1\sqrt{\sigma_1}$, $\|S_t\| \leq 2\sqrt{\sigma_1}$ and our assumption that $\gamma_t \leq \frac{0.01}{\sigma_1}$, $I - \gamma_t S_t^\top S_t - \gamma_t T_t^\top T_t$ is a PSD matrix. By lemma I.2,

$$\left\| S_t \left( I - \gamma_t S_t^\top S_t - \gamma_t T_t^\top T_t \right) \right\| \leq \left\| S_t \left( I - \gamma_t S_t^\top S_t \right) \right\| + \gamma_t \|S_t\| \|T_t\|^2 \tag{B.11}$$

$$= \|S_t\| - \gamma_t \|S_t\|^3 + 0.1\gamma_t \sigma_1^{\frac{3}{2}}. \tag{B.12}$$

On the other hand, simple triangle inequality yields

$$\|F_t\| = \|US_t + VT_t\| \leq \|S_t\| + \|T_t\| \leq 3\sqrt{\sigma_1}. \tag{B.13}$$

By A.8 and lemma I.4, we get

$$\|\Delta_t F_t\| \leq \|\Delta_t\| \|F_t\| \tag{B.14}$$

$$\leq 3\sigma_1^{\frac{1}{2}} \delta \|F_t F_t^\top - X_\natural\|_{\mathrm{F}} \tag{B.15}$$

$$\leq 3\sigma_1^{\frac{1}{2}} \delta \sqrt{k+r} \left\| F_t F_t^\top - X_\natural \right\| \tag{B.16}$$

$$\leq 3\sigma_1^{\frac{1}{2}} \delta \sqrt{k+r} \left( \|F_t\|^2 + \|X_\natural\| \right) \tag{B.17}$$

$$\leq 50\delta\sqrt{k}\sigma_1^{\frac{3}{2}} \tag{B.18}$$

$$\leq 0.1\sigma_1^{\frac{3}{2}} \tag{B.19}$$

Combining, we have

$$\|S_{t+1}\| \leq \left\| S_t \left( I - \gamma_t S_t^\top S_t - \gamma_t T_t^\top T_t \right) \right\| + \gamma_t \|D_S^* S_t\| + \gamma_t \left\| U^\top \Delta_t F_t \right\| \tag{B.20}$$

$$\leq \|S_t\| - \gamma_t \|S_t\|^3 + \gamma_t \sigma_1 \|S_t\| + 0.2\gamma_t \sigma_1^{\frac{3}{2}} \tag{B.21}$$

$$= \|S_t\| \left( 1 + \gamma_t \sigma_1 - \gamma_t \|S_t\|^2 \right) + 0.2\gamma_t \sigma_1^{\frac{3}{2}}. \tag{B.22}$$

We consider two different cases:

- $\|S_t\| \leq 1.5\sqrt{\sigma_1}$. By the inequality above, we have

$$\|S_{t+1}\| \leq \|S_t\| (1 + \gamma_t \sigma_1) + 0.1\gamma_t \sigma_1^{\frac{3}{2}} \leq 1.6\sqrt{\sigma_1} + 0.2\sqrt{\sigma_1} \leq 2\sqrt{\sigma_1} \tag{B.23}$$

- $1.5\sqrt{\sigma_1} < \|S_t\| \leq 2\sqrt{\sigma_1}$. In this case, we have

$$\|S_{t+1}\| \leq \|S_t\| (1 + \gamma_t \sigma_1 - 2.25\gamma_t \sigma_1) + 0.2\gamma_t \sigma_1^{\frac{3}{2}} \tag{B.24}$$

$$\leq \|S_t\| (1 - \gamma_t \sigma_1) + 0.2\gamma_t \sigma_1 \|S_t\| \tag{B.25}$$

$$\leq \|S_t\| \tag{B.26}$$

$$\leq 2\sqrt{\sigma_1}. \tag{B.27}$$

The desired bound for $S_{t+1}$ is established. For $T_{t+1}$, we note

$$T_{t+1} = T_t(I - \gamma_t T_t^\top T_t - \gamma_t S_t^\top S_t) + \gamma_t V^\top \Delta_t F_t. \tag{B.28}$$

We expand $T_{t+1} T_{t+1}^\top$ and obtain

$$T_{t+1} T_{t+1}^\top = (\mathcal{N}_t(T_t) + \gamma_t V^\top \Delta_t F_t)(\mathcal{N}_t(T_t) + \gamma_t V^\top \Delta_t F_t)^\top \tag{B.29}$$

$$\leq \underbrace{\mathcal{N}_t(T_t)\mathcal{N}_t(T_t)^\top}_{Z_1} + \underbrace{\gamma_t V^\top \Delta_t F_t \mathcal{N}_t(T_t)^\top + \gamma_t \mathcal{N}_t(T_t) F_t^\top \Delta_t^\top V}_{Z_2} \tag{B.30}$$

$$+ \underbrace{\gamma_t^2 V^\top \Delta_t F_t F_t^\top \Delta_t^\top V}_{Z_3} \tag{B.31}$$

By Proposition A.4, we have $\|Z_1\| \leq \|T_t T_t^\top\|(1 - \frac{3\gamma_t}{2}\|T_t T_t^\top\|)$. By induction hypothesis and triangle inequality, we have

$$\|Z_2\| \leq 2\gamma_t 50\delta\sqrt{k}\sigma_1^{\frac{3}{2}}\|\mathcal{N}_t(T_t)\| \tag{B.32}$$

$$\leq 2\gamma_t 50\delta\sqrt{k}\sigma_1^{\frac{3}{2}}\|T_t\| \tag{B.33}$$

and

$$\|Z_3\| \leq \gamma_t^2 \left(50\delta\sqrt{k}\sigma_1^{\frac{3}{2}}\right)^2 \leq 0.01\gamma_t(c_\rho\rho)^4. \tag{B.34}$$

By triangle inequality, we have

$$\left\|T_{t+1}T_{t+1}^\top\right\| \leq \|Z_1\| + \|Z_2\| + \|Z_3\| \tag{B.35}$$

$$\leq \|T_t T_t^\top\|(1 - \frac{3\gamma_t}{2}\|T_t T_t^\top\|) + 2\gamma_t 50\delta\sqrt{k}\sigma_1^{\frac{3}{2}}\|T_t\| + 0.01\gamma_t(c_\rho\rho)^4 \tag{B.36}$$

We consider two different cases:

- $\left\|T_t T_t^\top\right\| \leq \frac{(c_\rho\rho)^2}{2}$. We have

$$\left\|T_{t+1}T_{t+1}^\top\right\| \leq \|T_t T_t^\top\| + 2\gamma_t 50\delta\sqrt{k}\sigma_1^{\frac{3}{2}}\|T_t\| + 0.01\gamma_t(c_\rho\rho)^4 \tag{B.37}$$

$$\leq \frac{(c_\rho\rho)^2}{2} + \frac{\gamma_t(c_\rho\rho)^4}{4} + 0.01\gamma_t(c_\rho\rho)^4 \tag{B.38}$$

$$\leq (c_\rho\rho)^2. \tag{B.39}$$

- $\frac{(c_\rho\rho)^2}{2} < \left\|T_t T_t^\top\right\| \leq (c_\rho\rho)^2$. We have

$$\left\|T_{t+1}T_{t+1}^\top\right\| \leq \|T_t T_t^\top\|(1 - \frac{3\gamma_t}{2}\|T_t T_t^\top\|) + 2\gamma_t 50\delta\sqrt{k}\sigma_1^{\frac{3}{2}}\|T_t\| + 0.01\gamma_t(c_\rho\rho)^4 \tag{B.40}$$

$$\leq \|T_t T_t^\top\| - \frac{3\gamma_t}{8}(c_\rho\rho)^4 + 2\gamma_t 50\delta\sqrt{k}\sigma_1^{\frac{3}{2}}\|T_t\| + 0.01\gamma_t(c_\rho\rho)^4 \tag{B.41}$$

$$\leq \|T_t T_t^\top\| - \frac{3\gamma_t}{8}(c_\rho\rho)^4 + \frac{\gamma_t(c_\rho\rho)^4}{4} + 0.01\gamma_t(c_\rho\rho)^4 \tag{B.42}$$

$$\leq \|T_t T_t^\top\|^2 \tag{B.43}$$

$$\leq (c_\rho\rho)^2. \tag{B.44}$$

Hence, we proved the inequality for $\|T_{t+1}\|$. By induction, the proof is complete.

## B.4 Proof of Proposition A.4

1. $\left\|\mathcal{M}_t(S_t)\mathcal{M}_t(S_t)^\top - D_S^*\right\| \leq (1 - \frac{3\gamma_t\sigma_r}{4})\left\|S_t S_t^\top - D_S^*\right\| + 3\gamma_t\left\|S_t T_t^\top\right\|^2$.
   First, we suppose that we have $\gamma_t \leq \min\{\frac{0.01}{\sigma_1}, \frac{0.01\sigma_r}{\sigma_1^2}\}$. By definition,

$$\mathcal{M}_t(S_t) = S_t - \gamma_t\left(S_t S_t^\top S_t + S_t T_t^\top T_t - D_S^* S_t\right). \tag{B.45}$$

This yields

$$D_S^* - \mathcal{M}_t(S_t)\mathcal{M}_t(S_t)^\top \tag{B.46}$$

$$= D_S^* - \left[S_t - \gamma_t\left(S_t S_t^\top S_t + S_t T_t^\top T_t - D_S^* S_t\right)\right]\left[S_t - \gamma_t\left(S_t S_t^\top S_t + S_t T_t^\top T_t - D_S^* S_t\right)\right]^\top \tag{B.47}$$

$$= Z_1 + Z_2 + Z_3, \tag{B.48}$$

where

$$Z_1 = D_S^* - S_t S_t^\top - \gamma_t(D_S^* - S_t S_t^\top)S_t S_t^\top - \gamma_t S_t S_t^\top(D_S^* - S_t S_t^\top), \tag{B.49}$$

$$Z_2 = 2\gamma_t S_t T_t^\top T_t S_t^\top, \tag{B.50}$$

and
$$Z_3 = -\gamma_t^2 (S_t S_t^\top S_t + S_t T_t^\top T_t - D_S^* S_t)(S_t S_t^\top S_t + S_t T_t^\top T_t - D_S^* S_t)^\top. \quad \text{(B.51)}$$

We bound each of them separately. For $Z_1$, by triangle inequality,

$$\|Z_1\| = \left\| (D_S^* - S_t S_t^\top) \left( \frac{1}{2} I - \gamma_t S_t S_t^\top \right) + \left( \frac{1}{2} I - \gamma_t S_t S_t^\top \right) (D_S^* - S_t S_t^\top) \right\| \quad \text{(B.52)}$$

$$\leq \left\| (D_S^* - S_t S_t^\top) \left( \frac{1}{2} I - \gamma_t S_t S_t^\top \right) \right\| + \left\| \left( \frac{1}{2} I - \gamma_t S_t S_t^\top \right) (D_S^* - S_t S_t^\top) \right\| \quad \text{(B.53)}$$

$$\leq (\frac{1}{2} - \gamma_t \sigma_r^2(S_t)) \left\| D_S^* - S_t S_t^\top \right\| + (\frac{1}{2} - \gamma_t \sigma_r^2(S_t)) \left\| D_S^* - S_t S_t^\top \right\| \quad \text{(B.54)}$$

$$\leq (1 - \gamma_t \sigma_r) \left\| D_S^* - S_t S_t^\top \right\| \quad \text{(B.55)}$$

The norm of $Z_2$ can be simply bounded by

$$\|Z_2\| \leq 2\gamma_t \left\| S_t T_t^\top \right\|^2 \quad \text{(B.56)}$$

For $Z_3$, we can split it as

$$Z_3 = -\gamma_t^2 (S_t S_t^\top - D_S^*) S_t S_t^\top (S_t S_t^\top - D_S^*)^\top - \gamma_t^2 S_T T_t^\top T_t T_t^\top T_t S_t^\top \quad \text{(B.57)}$$
$$- \gamma_t^2 \left( S_t T_t^\top T_t \left( S_t^\top S_t S_t^\top - S_t^\top D_S^* \right) + \left( S_t S_t^\top S_t - D_S^* S_t \right) T_t^\top T_t S_t^\top \right) \quad \text{(B.58)}$$

By triangle inequality and our assumption that $\|S_t\| \leq 2\sqrt{\sigma_1}$, we have $\left\| S_t S_t^\top - D_S^* \right\| \leq 5\sigma_1$. Hence

$$\|Z_3\| \leq 20\gamma_t^2 \sigma_1^2 \left\| S_t S_t^\top - D_S^* \right\| + 0.01\gamma_t^2 \sigma_1 \left\| S_t T_t^\top \right\|^2 + 2\gamma_t^2 \left\| S_t S_t^\top - D_S^* \right\| \left\| S_t T_t^\top \right\|^2 \quad \text{(B.59)}$$

$$\overset{(\sharp)}{\leq} 20\gamma_t^2 \sigma_1^2 \left\| S_t S_t^\top - D_S^* \right\| + 0.01\gamma_t^2 \sigma_1 \left\| S_t T_t^\top \right\|^2 + \gamma_t^2 \sigma_1 \sigma_r \left\| S_t S_t^\top - D_S^* \right\| \quad \text{(B.60)}$$

$$\leq (20\gamma_t^2 \sigma_1^2 + \gamma_t^2 \sigma_1 \sigma_r) \left\| S_t S_t^\top - D_S^* \right\| + \gamma_t \left\| S_t T_t^\top \right\|^2 \quad \text{(B.61)}$$

$$\leq (\frac{20}{100} \gamma_t \sigma_r + \frac{1}{100} \gamma_t \sigma_r) \left\| S_t S_t^\top - D_S^* \right\| + \gamma_t \left\| S_t T_t^\top \right\|^2 \quad \text{(B.62)}$$

Here $(\sharp)$ follows from our assumption that $\|S_t\| \leq 2\sqrt{\sigma_1}$ and $\|T_t\| \leq 0.1\sqrt{\sigma_r}$. Combining, we have

$$\left\| D_S^* - S_{t+1} S_{t+1}^\top \right\| \leq \|Z_1\| + \|Z_2\| + \|Z_3\| \quad \text{(B.63)}$$

$$\leq (1 - \gamma_t \sigma_r + \frac{21}{100} \gamma_t \sigma_r) \left\| D_S^* - S_t S_t^\top \right\| + (2\gamma_t + \gamma_t) \left\| S_t T_t^\top \right\|^2 \quad \text{(B.64)}$$

$$\leq \left( 1 - \frac{3\gamma_t \sigma_r}{4} \right) \left\| D_S^* - S_t S_t^\top \right\| + 3\gamma_t \left\| S_t T_t^\top \right\|^2 \quad \text{(B.65)}$$

If we assume $\left\| S_t S_t^\top - D_S^* \right\| \leq \frac{\sigma_r}{10}$ and $\gamma_t \leq \frac{0.01}{\sigma_1}$ instead, the only bound that will change is

$$\|Z_3\| \leq 4\gamma_t^2 \sigma_1 \left\| S_t S_t^\top - D_S^* \right\|^2 + 0.01\gamma_t^2 \sigma_r \left\| S_t T_t^\top \right\|^2 + 2\gamma_t^2 \left\| S_t S_t^\top - D_S^* \right\| \left\| S_t T_t^\top \right\|^2 \quad \text{(B.66)}$$

$$\leq 4\gamma_t^2 \sigma_1 \frac{\sigma_r}{10} \left\| S_t S_t^\top - D_S^* \right\| + 0.01\gamma_t^2 \sigma_1 \left\| S_t T_t^\top \right\|^2 + \gamma_t^2 \sigma_1 \sigma_r \left\| S_t S_t^\top - D_S^* \right\| \quad \text{(B.67)}$$

$$\leq \left( \frac{4}{1000} \gamma_t \sigma_r + \frac{1}{100} \gamma_t \sigma_r \right) \left\| S_t S_t^\top - D_S^* \right\| + \gamma_t \left\| S_t T_t^\top \right\|^2. \quad \text{(B.68)}$$

With this bound, we can do same argument except only with $\gamma_t \leq \frac{0.01}{\sigma_1}$ to get same bound

$$\left\| D_S^* - S_{t+1} S_{t+1}^\top \right\| \leq \left( 1 - \frac{3\gamma_t \sigma_r}{4} \right) \left\| D_S^* - S_t S_t^\top \right\| + 3\gamma_t \left\| S_t T_t^\top \right\|^2 \quad \text{(B.69)}$$

2. $\left\|\mathcal{N}_t(T_t)\mathcal{N}_t(T_t)^\top\right\| \le \|T_t\|^2 \left(1 - \gamma_t \|T_t\|^2\right) = \left\|T_t T_t^\top\right\| \left(1 - \gamma_t \left\|T_t T_t^\top\right\|\right)$.
By definition,

$$\mathcal{N}_t(T_t) = T_t - \gamma_t(T_t T_t^\top T_t + T_t S_t^\top S_t). \tag{B.70}$$

Plug this into $\mathcal{N}_t(T_t)\mathcal{N}_t(T_t)^\top$, we obtain

$$\mathcal{N}_t(T_t)\mathcal{N}_t(T_t)^\top = \left(T_t - \gamma_t(T_t T_t^\top T_t + T_t S_t^\top S_t)\right)\left(T_t - \gamma_t(T_t T_t^\top T_t + T_t S_t^\top S_t)\right)^\top \tag{B.71}$$

$$= Z_4 + Z_5 + Z_6 \tag{B.72}$$

where

$$Z_4 = T_t T_t^\top - 2\gamma_t T_t T_t^\top T_t T_t^\top, \tag{B.73}$$

$$Z_5 = -2\gamma_t T_t S_t^\top S_t T_t^\top + \gamma_t^2 T_t S_t^\top S_t S_t^\top S_t T_t^\top, \tag{B.74}$$

and

$$Z_6 = \gamma_t^2 \left[T_t T_t^\top \left(T_t S_t^\top S_t T_t^\top\right) + \left(T_t S_t^\top S_t T_t^\top\right) T_t T_t^\top\right] \tag{B.75}$$

We bound each of them separately. By lemma I.1, we obtain

$$\|Z_4\| \le \left\|T_t T_t^\top - 2\gamma_t T_t T_t^\top T_t T_t^\top\right\| \tag{B.76}$$

$$\le \left\|T_t T_t^\top\right\|\left(1 - 2\gamma_t \left\|T_t T_t^\top\right\|\right) \tag{B.77}$$

$$= \|T_t\|^2 \left(1 - 2\gamma_t \|T_t\|^2\right). \tag{B.78}$$

On the other hand,

$$Z_5 = -2\gamma_t T_t S_t^\top S_t T_t^\top + \gamma_t^2 T_t S_t^\top S_t S_t^\top S_t T_t^\top \tag{B.79}$$

$$= -2\gamma_t T_t S_t^\top (I - \gamma_t S_t S_t^\top) S_t T_t^\top \tag{B.80}$$

$$\preceq 0. \tag{B.81}$$

Furthermore,

$$\|Z_6\| \le \gamma_t^2 \left\|S_t^\top S_t\right\| \left\|T_t T_t^\top\right\|^2 \tag{B.82}$$

$$\le \frac{\gamma_t}{2} \left\|T_t T_t^\top\right\|^2. \tag{B.83}$$

Combining, we obtain

$$\left\|\mathcal{N}_t(T_t)\mathcal{N}_t(T_t)^\top\right\| = \|Z_4 + Z_5 + Z_6\| \tag{B.84}$$

$$\le \|Z_4 + Z_5\| + \|Z_6\| \tag{B.85}$$

$$\le \|Z_4\| + \|Z_6\| \tag{B.86}$$

$$\le \left\|T_t T_t^\top\right\|\left(1 - \frac{3\gamma_t}{2}\left\|T_t T_t^\top\right\|\right). \tag{B.87}$$

The second inequality follows from the fact that $Z_5 \preceq 0$. In this proof, we only need $\gamma_t \le \frac{0.01}{\sigma_1}$.

3. $\left\|\mathcal{M}_t(S_t)\mathcal{N}_t(T_t)^\top\right\| \le (1 - \frac{\gamma_t \sigma_r}{3}) \left\|S_t T_t^\top\right\|$.
By definition of $\mathcal{M}_t(S_t), \mathcal{N}_t(T_t)$, we have

$$\mathcal{M}_t(S_t)\mathcal{N}_t(T_t)^\top = \left(S_t - \gamma_t(S_t S_t^\top S_t + S_t T_t^\top T_t - D_S^* S_t)\right)\left(T_t - \gamma_t(T_t T_t^\top T_t + T_t S_t^\top S_t)\right)^\top \tag{B.88}$$

$$= Z_7 + Z_8 + Z_9, \tag{B.89}$$

where

$$Z_7 = (I - \gamma_t S_t S_t^\top) S_t T_t^\top, \tag{B.90}$$

$$Z_8 = \gamma_t(D_S^* - S_t S_t^\top) S_t T_t^\top \tag{B.91}$$

and

$$Z_9 = -2\gamma_t S_t T_t^\top T_t T_t^\top + \gamma_t^2(S_t S_t^\top S_t + S_t T_t^\top T_t - D_S^* S_t)(T_t T_t^\top T_t + T_t S_t^\top S_t)^\top. \tag{B.92}$$

We bound each of them. By our assumption that $\sigma_r(S_t) \geq \sqrt{\frac{\sigma_r}{2}}$,

$$\|Z_7\| \leq (1 - \frac{\gamma_t \sigma_r}{2}) \|S_t T_t^\top\|. \tag{B.93}$$

By the assumption that $\|D_S^* - S_t S_t^\top\| \leq \frac{\sigma_r}{10}$,

$$\|Z_8\| \leq \frac{\gamma_t \sigma_r}{10} \|S_t T_t^\top\|. \tag{B.94}$$

For $Z_9$, we use triangle inequality and get

$$\|Z_9\| \leq \|2\gamma_t S_t T_t^\top T_t T_t^\top\| + \gamma_t^2 \|(S_t S_t^\top - D_S^*) S_t T_t^\top T_t T_t^\top\| \tag{B.95}$$
$$+ \gamma_t^2 \|(S_t S_t^\top - D_S^*) S_t S_t^\top S_t T_t^\top)\| + \gamma_t^2 \|S_t T_t^\top T_t (T_t^\top T_t T_t^\top + S_t^\top S_t T_t^\top)\| \tag{B.96}$$

$$\stackrel{(\sharp)}{\leq} \frac{2}{100} \gamma_t \sigma_r \|S_t T_t^\top\| + \gamma_t^2 \frac{\sigma_r}{10} \frac{\sigma_r}{100} \|S_t T_T^\top\| + 4\gamma_t^2 \frac{\sigma_r}{10} \sigma_1 \|S_t T_t^\top\| \tag{B.97}$$

$$+ \gamma_t^2 \left(\frac{\sigma_r}{100}\right)^2 \|S_t T_t^\top\| + \gamma_t^2 \left(\frac{2\sqrt{\sigma_1 \sigma_r}}{10}\right)^2 \|S_t T_t^\top\| \tag{B.98}$$

$$\stackrel{(\star)}{\leq} \frac{\gamma_t \sigma_r}{20} \|S_t T_t^\top\| \tag{B.99}$$

In $(\sharp)$, we used the bound that $\|S_t T_t^\top\| \leq \|S_t\| \|T_t\| \leq \frac{2\sqrt{\sigma_1 \sigma_r}}{10}$ and $\|T_t T_t^\top\| \leq \frac{\sigma_r}{100}$. $(\star)$ follows from our assumption that $\gamma_t \leq \frac{0.01}{\sigma_1}$. Combining, we obtain

$$\|\mathcal{M}_t(S_t)\mathcal{N}_t(T_t)^\top\| \leq \|Z_7\| + \|Z_8\| + \|Z_9\| \tag{B.100}$$
$$\leq \left(1 - \frac{\gamma_t \sigma_r}{3}\right) \|S_t T_t^\top\|. \tag{B.101}$$

## B.5 Proof of Proposition A.5

We prove them one by one.

1. $\|D_S^* - \mathcal{M}_t(S_t)S_t^\top\| \leq (1 - \frac{\gamma_t \sigma_r}{2}) \|D_S^* - S_t S_t^\top\| + \gamma_t \|S_t T_t^\top\|^2$. By definition of $\mathcal{M}_t(S_t)$, we know that

$$\mathcal{M}_t(S_t)S_t^\top - D_S^* = S_t S_t^\top - \gamma_t(S_t S_t^\top S_t + S_t T_t^\top T_t - D_S^* S_t)S_t^\top - D_S^* \tag{B.102}$$
$$= (S_t S_t^\top - D_S^*)(I - \gamma_t S_t S_t^\top) - \gamma_t S_t T_t^\top T_t S_t^\top. \tag{B.103}$$

By our assumption that $\sigma_r(S_t) \geq \sqrt{\frac{\sigma_r}{2}}$, we know

$$\|(S_t S_t^\top - D_S^*)(I - \gamma_t S_t S_t^\top)\| \leq (1 - \frac{\sigma_r}{2}) \|S_t S_t^\top - D_S^*\|. \tag{B.104}$$

By triangle inequality, the result follows.

2. $\|\mathcal{M}_t(S_t)T_t^\top\| \leq 2 \|S_t T_t^\top\|$. By definition of $\mathcal{M}_t(S_t)$, we have

$$\mathcal{M}_t(S_t)T_t^\top = S_t T_t^\top - \gamma_t(S_t S_t^\top S_t + S_t T_t^\top T_t - D_S^* S_t)T_t^\top \tag{B.105}$$

Triangle inequality yields

$$\gamma_t \|(S_t S_t^\top S_t + S_t T_t^\top T_t - D_S^* S_t)T_t^\top\| \leq \gamma_t(\|S_t\|^2 + \|T_t\|^2 + \|D_S^*\|) \|S_t T_t^\top\| \tag{B.106}$$
$$\leq \gamma_t(4\sigma_1 + 0.01\sigma_r + \sigma_1) \|S_t T_t^\top\| \tag{B.107}$$
$$\leq \|S_t T_t^\top\| \tag{B.108}$$

The last inequality follows from our assumption that $\gamma_t \leq \frac{0.01}{\sigma_1}$. By triangle inequality again, we obtain

$$\|\mathcal{M}_t(S_t)T_t^\top\| \leq \|S_t T_t^\top\| + \|S_t T_t^\top\| = 2 \|S_t T_t^\top\|. \tag{B.109}$$

3. $\left\|\mathcal{N}_t(T_t)S_t^\top\right\| \le \left\|T_tS_t^\top\right\|$. By definition of $\mathcal{N}_t(T_t)$,

$$\mathcal{N}_t(T_t)S_t^\top = T_tS_t^\top - \gamma_t(T_tT_t^\top T_t + T_tS_t^\top S_t)S_t^\top \tag{B.110}$$

$$= \left(\frac{1}{2}I - \gamma_tT_tT_t^\top\right)T_tS_t^\top + T_tS_t^\top\left(\frac{1}{2}I - \gamma_tS_tS_t^\top\right) \tag{B.111}$$

By triangle inequality,

$$\left\|\mathcal{N}_t(T_T)S_t^\top\right\| \le \left\|\left(\frac{1}{2}I - \gamma_tT_tT_t^\top\right)T_tS_t^\top\right\| + \left\|T_tS_t^\top\left(\frac{1}{2}I - \gamma_tS_tS_t^\top\right)\right\| \tag{B.112}$$

$$\le \left\|T_tS_t^\top\right\|. \tag{B.113}$$

The last inequality follows from the choice of $\gamma_t$ and the fact that $\left\|\left(\frac{1}{2}I - \gamma_tT_tT_t^\top\right)\right\| \le \frac{1}{2}$, $\left\|\left(\frac{1}{2}I - \gamma_tS_tS_t^\top\right)\right\| \le \frac{1}{2}$.

4. $\left\|\mathcal{N}_t(T_t)T_t^\top\right\| \le \left\|T_t\right\|^2\left(1 - \gamma_t\left\|T_t\right\|^2\right) = \left\|T_tT_t^\top\right\|\left(1 - \gamma_t\left\|T_tT_t^\top\right\|\right)$. By definition of $\mathcal{N}_t(T_t)$, we have

$$\mathcal{N}_t(T_t)T_t^\top = T_tT_t^\top - \gamma_t(T_tT_t^\top T_t + T_tS_t^\top S_t)T_t^\top \tag{B.114}$$

$$= T_tT_t^\top(I - \gamma_tT_tT_t^\top) - \gamma_tT_tS_t^\top S_tT_t^\top \tag{B.115}$$

$$\preceq T_tT_t^\top(I - \gamma_tT_tT_t^\top) \tag{B.116}$$

As a result of lemma I.1, we have

$$\left\|\mathcal{N}_t(T_t)T_t^\top\right\| \le \left\|T_tT_t^\top\right\|\left(1 - \gamma_t\left\|T_tT_t^\top\right\|\right) = \left\|T_t\right\|^2\left(1 - \gamma_t\left\|T_t\right\|^2\right). \tag{B.117}$$

## B.6 Proof of Proposition A.6

We prove this proposition by induction. Note that the inequality A.27 holds trivially when $s = 0$. Suppose it holds for $t \ge 0$. By Proposition A.2, we can write $S_{t+1}$ as

$$S_{t+1} = \left(I - \gamma_tS_tS_t^\top + \gamma_tD_S^*\right)S_t - \gamma_tS_tT_t^\top T_t + \gamma_tU^\top\Delta_tF_t \tag{B.118}$$

$$= (I + \gamma_tD_S^*)S_t(I - \gamma_tS_t^\top S_t) + \gamma_t^2D_S^*S_tS_t^\top S_t - \gamma_tS_tT_t^\top T_t + \gamma_tU^\top\Delta_tF_t \tag{B.119}$$

These two ways of expressing $S_{t+1}$ are crucial to the proof.

For the ease of notation, we introduce some notations. Let

$$H_t = I - \gamma_tS_tS_t^\top + \gamma_tD_S^* \tag{B.120}$$

$$E_t = S_tT_t^\top T_t - U^\top\Delta_tF_t \tag{B.121}$$

By Proposition A.3 and our assumption that $(50\sqrt{k}\delta)^{\frac{1}{3}} \le \frac{c_\rho\rho}{2\sqrt{\sigma_1}}$, we have

$$\left\|E_t\right\| \le \left\|S_t\right\|\left\|T_t\right\|^2 + \left\|\Delta_tF_t\right\| \tag{B.122}$$

$$\le 2(c_\rho\rho)^2\sqrt{\sigma_1} + 50\delta\sqrt{k}\sigma_1^{\frac{3}{2}} \tag{B.123}$$

$$\le 2(c_\rho\rho)^2\sqrt{\sigma_1} + \frac{(c_\rho\rho)^3}{8} \tag{B.124}$$

$$\le 3(c_\rho\rho)^2\sqrt{\sigma_1}. \tag{B.125}$$

In the last inequality, we used our assumption that $c_\rho\rho \le 0.1\sqrt{\sigma_1}$. By lemma I.1 and our choice of $\gamma_t$, we know $H_t$ is invertible and

$$\left\|H_t^{-1}\right\| \le \frac{1}{1 - \gamma_t\left\|S_t\right\|^2 - \gamma_t\left\|D_S^*\right\|} \le \frac{1}{1 - 0.04 - 0.01} \le 2. \tag{B.126}$$

By B.118, we can write

$$S_t = H_t^{-1}S_{t+1} + \gamma_tH_t^{-1}E_t \tag{B.127}$$

Plug this in to B.119 and rearrange, we get

$$\left(I - \gamma_t^2D_S^*S_tS_t^\top H_t^{-1}\right)S_{t+1} = \underbrace{(I + \gamma_tD_S^*)S_t(I - \gamma_tS_t^\top S_t)}_{Z_1} + \underbrace{\gamma_t^3D_S^*S_tS_t^\top H_t^{-1}E_t - \gamma_tE_t}_{Z_2}$$

$$\tag{B.128}$$

Let's consider the $r$-th singular value of both sides. For LHS, by lemma I.2 and lemma I.1

$$\sigma_r\left(\left(I - \gamma_t^2 D_S^* S_t S_t^\top H_t^{-1}\right) S_{t+1}\right) \leq \left\|\left(I - \gamma_t^2 D_S^* S_t S_t^\top H_t^{-1}\right)\right\| \sigma_r(S_{t+1}) \tag{B.129}$$

$$\leq \frac{1}{1 - \gamma_t^2 \left\|D_S^* S_t S_t^\top H_t^{-1}\right\|} \sigma_r(S_{t+1}) \tag{B.130}$$

$$\leq \frac{1}{1 - 8\gamma_t^2 \sigma_1^2} \sigma_r(S_{t+1}). \tag{B.131}$$

For RHS, we consider $Z_1$ and $Z_2$ separately. For $Z_1$, by lemma I.2, we have

$$\sigma_r(Z_1) \geq \sigma_r(I + \gamma_t D_S^*) \cdot \sigma_r(S_t(I - \gamma_t S_t^\top S_t)) \tag{B.132}$$

$$= (1 + \gamma_t \sigma_r)\sigma_r(S_t)(1 - \gamma_t \sigma_r^2(S_t)) \tag{B.133}$$

For $Z_2$, by triangle inequality,

$$\left\|\gamma_t^3 D_S^* S_t S_t^\top H_t^{-1} E_t - \gamma_t E_t\right\| \leq \gamma_t^3 \left\|D_S^* S_t S_t^\top H_t^{-1} E_t\right\| + \gamma_t \left\|E_t\right\| \tag{B.134}$$

$$\leq (8\gamma_t^3 \sigma_1^2 + \gamma_t) \left\|E_t\right\| \tag{B.135}$$

$$\leq 3(8\gamma_t^3 \sigma_1^2 + \gamma_t)(c_\rho \rho)^2 \sqrt{\sigma_1}. \tag{B.136}$$

Combining, by lemma I.3, we obtain

$$\sigma_r\left((I + \gamma_t D_S^*) S_t(I - \gamma_t S_t^\top S_t) + \gamma_t^3 D_S^* S_t S_t^\top H_t^{-1} E_t - \gamma_t E_t\right) \tag{B.137}$$

$$\geq \sigma_r(Z_1) - \gamma_t \left\|E_t\right\| - \left\|Z_2\right\| \tag{B.138}$$

$$\geq (1 + \gamma_t \sigma_r)\sigma_r(S_t)(1 - \gamma_t \sigma_r^2(S_t)) - 3(8\gamma_t^3 \sigma_1^2 + \gamma_t)(c_\rho \rho)^2 \sqrt{\sigma_1}. \tag{B.139}$$

By induction hypothesis, we know $\sigma_r(S_t) \geq \rho$. Note we assumed that $4c_\rho^2 \rho \leq 0.01 \frac{\sigma_r}{\sqrt{\sigma_1}}$, so we have

$$3(8\gamma_t^3 \sigma_1^2 + \gamma_t)(c_\rho \rho)^2 \sqrt{\sigma_1} \leq 4\gamma_t(c_\rho \rho)^2 \sqrt{\sigma_1} \leq 0.01\gamma_t \sigma_r \sigma_r(S_t) \tag{B.140}$$

Consequently, we get

$$\sigma_r\left((I + \gamma_t D_S^*) S_t(I - \gamma_t S_t^\top S_t) + \gamma_t^3 D_S^* S_t S_t^\top H_t^{-1} E_t - \gamma_t E_t\right) \tag{B.141}$$

$$\geq (1 + \gamma_t \sigma_r)\sigma_r(S_t)(1 - \gamma_t \sigma_r^2(S_t)) - 0.01\sigma_r \sigma_r(S_t) \tag{B.142}$$

$$= \sigma_r(S_t)\left(1 + 0.99\gamma_t \sigma_r - \gamma_t \sigma_r^2(S_t) - \gamma_t^2 \sigma_r \sigma_r^2(S_t)\right) \tag{B.143}$$

Combining the LHS and RHS, we finally get

$$\sigma_r(S_{t+1}) \geq (1 - 8\gamma_t^2 \sigma_1^2)\left(1 + 0.99\gamma_t \sigma_r - \gamma_t \sigma_r^2(S_t) - \gamma_t^2 \sigma_r \sigma_r^2(S_t)\right)\sigma_r(S_t) \tag{B.144}$$

We consider two cases(recall $\sigma_r = \frac{1}{\kappa}$):

- $\sigma_r(S_t) \geq \sqrt{\frac{3\sigma_r}{4}}$. By B.144, we know that

$$\sigma_r(S_{t+1}) \geq (1 - 8\gamma_t^2 \sigma_1^2)(1 - 5\gamma_t \sigma_1)\sigma_r(S_t). \tag{B.145}$$

Here we used Proposition A.3 to bound $\sigma_r(S_t)$ by $2\sqrt{\sigma_1}$. Since $\gamma_t \leq \frac{0.01}{\sigma_1}$, simple calculation shows that

$$\sigma_r(S_{t+1}) \geq (1 - 8\gamma_t^2 \sigma_1^2)(1 - 5\gamma_t \sigma_1)\sqrt{\frac{3\sigma_r}{4}} \geq \sqrt{\frac{\sigma_r}{2}}. \tag{B.146}$$

- $\sigma_r(S_t) < \sqrt{\frac{3\sigma_r}{4}}$. By B.144 and induction hypothesis, we know

$$\sigma_r(S_{t+1}) \geq (1 - 8\gamma_t^2 \sigma_1^2)\left(1 + 0.99\gamma_t \sigma_r - \gamma_t \sigma_r^2(S_t) - \gamma_t^2 \sigma_r \sigma_r^2(S_t)\right)\sigma_r(S_t) \tag{B.147}$$

$$\geq (1 - 8\gamma_t^2 \sigma_1^2)\left(1 + \frac{\gamma_t \sigma_r}{5}\right)\sigma_r(S_t) \tag{B.148}$$

$$\geq \left(1 + \frac{\gamma_t \sigma_r^2}{6\sigma_1^2}\right)\sigma_r(S_t) \tag{B.149}$$

$$\geq \min\{(1 + \frac{c_\gamma \sigma_r^2}{6\sigma_1^2})^{t+1}\sigma_r(S_0), \sqrt{\frac{\sigma_r}{2}}\}. \tag{B.150}$$

We used the bound $\gamma_t \geq \frac{c_\gamma \sigma_r}{\sigma_1^2}$ in the last inequality.

By induction, we proved inequality A.27 for $\sigma_r(S_t)$. By our choice of $\mathcal{T}_1$, it's easy to verify that

$$\sigma_r(S_{\mathcal{T}_1 + t}) \geq \sqrt{\frac{\sigma_r}{2}}, \qquad \forall t \geq 0. \tag{B.151}$$

## B.7 Proof of Proposition A.7

We prove it by induction. For the ease of notation, we use index $t$ for $t \geq \mathcal{T}_1'$ instead of $\mathcal{T}_1' + t$. The inequality A.29 holds for $t = \mathcal{T}_1'$ by Proposition A.3 and triangle inequality that

$$\left\| S_{\mathcal{T}_1} S_{\mathcal{T}_1}^\top - D_S^* \right\| \leq \|S_{\mathcal{T}_1}\|^2 + \|D_S^*\| \leq 5\sigma_1. \tag{B.152}$$

Suppose that A.29 holds for some $t \geq \mathcal{T}_1'$. By Proposition A.2, we have

$$S_{t+1} = \mathcal{M}_t(S_t) + \gamma_t U^\top \Delta_t F_t. \tag{B.153}$$

As a result,

$$S_{t+1} S_{t+1}^\top - D_S^* = \underbrace{\mathcal{M}_t(S_t)\mathcal{M}_t(S_t)^\top - D_S^*}_{Z_1} + \underbrace{\gamma_t(U^\top \Delta_t F_t \mathcal{M}_t(S_t)^\top + \mathcal{M}_t(S_t) F_t^\top \Delta_t^\top U)}_{Z_2} \tag{B.154}$$

$$+ \underbrace{\gamma_t^2 U^\top \Delta_t F_t F_t^\top \Delta_t^\top U}_{Z_3} \tag{B.155}$$

By Proposition A.4, we know

$$\|Z_1\| \leq (1 - \frac{3\gamma_t \sigma_r}{4}) \left\| S_t S_t^\top - D_S^* \right\| + 3\gamma_t \left\| S_t T_t^\top \right\|^2 \tag{B.156}$$

$$\overset{(\sharp)}{\leq} (1 - \frac{3\gamma_t \sigma_r}{4}) \left\| S_t S_t^\top - D_S^* \right\| + 12\gamma_t \sigma_1 (c_\rho \rho)^2 \tag{B.157}$$

Here $(\sharp)$ follows from Proposition A.3.
On the other hand, it's easy to see $\|\mathcal{M}_t(S_t)\| \leq 3\sqrt{\sigma_1}$ by its definition and Proposition A.3. By triangle inequality,

$$\|Z_2\| \leq 2\gamma_t \left\| U^\top \Delta_t F_t \mathcal{M}_t(S_t)^\top \right\| \tag{B.158}$$

$$\leq 2\gamma_t \|\Delta_t\| \|U S_t + V T_t\| \|\mathcal{M}_t(S_t)\| \tag{B.159}$$

$$\leq 18\gamma_t \|\Delta_t\| \sigma_1 \tag{B.160}$$

$$\overset{(\sharp)}{\leq} 18\gamma_t \delta \sqrt{k+r} \left\| F_t F_t^\top - X_\natural \right\| \sigma_1 \tag{B.161}$$

$$\overset{(\star)}{\leq} 270\gamma_t \delta \sqrt{k} \sigma_1^2 \tag{B.162}$$

$$\overset{(*)}{\leq} \gamma_t (c_\rho \rho)^3 \sqrt{\sigma_1} \tag{B.163}$$

Here $(\sharp)$ follows from A.8, $(\star)$ follows from uniform bound $\|F_t\| \leq 3\sqrt{\sigma_1}$, and $(*)$ follows from the assumption that $(50\sqrt{k}\delta)^{\frac{1}{3}} \leq \frac{c_\rho \rho}{2\sqrt{\sigma_1}}$.
Furthermore,

$$\|Z_3\| \leq \gamma_t^2 \|\Delta_t\|^2 \|F_t\|^2 \tag{B.164}$$

$$\leq 9\gamma_t^2 (10\delta \sqrt{k+r})^2 \sigma_1^3 \tag{B.165}$$

$$\leq \gamma_t^2 (c_\rho \rho)^6 \tag{B.166}$$

The last inequality follows simply from our assumption that $(50\sqrt{k}\delta)^{\frac{1}{3}} \leq \frac{c_\rho \rho}{2\sqrt{\sigma_1}}$. Combining, we obtain

$$\left\| S_{t+1} S_{t+1}^\top - D_S^* \right\| \leq \|Z_1\| + \|Z_2\| + \|Z_3\| \tag{B.167}$$

$$\leq (1 - \frac{3\gamma_t \sigma_r}{4}) \left\| S_t S_t^\top - D_S^* \right\| + 12\gamma_t (c_\rho \rho)^2 \sigma_1 + \gamma_t (c_\rho \rho)^3 \sqrt{\sigma_1} + \gamma_t^2 (c_\rho \rho)^6 \tag{B.168}$$

$$\leq (1 - \frac{3\gamma_t \sigma_r}{4}) \left\| S_t S_t^\top - D_S^* \right\| + 13\gamma_t (c_\rho \rho)^2 \sigma_1 \tag{B.169}$$

In the last inequality, we used $c_\rho \rho \leq 0.1\sqrt{\sigma_1}$ and $\gamma_t \leq \frac{0.01}{\sigma_1}$. We consider two cases:

- $\left\|S_t S_t^\top - D_S^*\right\| \le \frac{52(c_\rho\rho)^2\sigma_1}{\sigma_r}$. By above inequality, we simply have

$$\left\|S_{t+1}S_{t+1}^\top - D_S^*\right\| \le \left\|S_t S_t^\top - D_S^*\right\| + 13\gamma_t(c_\rho\rho)^2\sigma_1 \le \frac{100(c_\rho\rho)^2\sigma_1}{\sigma_r}. \quad \text{(B.170)}$$

The last inequality follows from the assumption that $\gamma_t \le \frac{0.01}{\sigma_1} \le \frac{0.01}{\sigma_r}$.

- $\left\|S_t S_t^\top - D_S^*\right\| > \frac{52(c_\rho\rho)^2\sigma_1}{\sigma_r}$. In this case, $13\gamma_t(c_\rho\rho)^2\sigma_1 \le \frac{\gamma_t\sigma_r}{4}\left\|S_t S_t^\top - D_S^*\right\|$. Consequently,

$$\left\|S_{t+1}S_{t+1}^\top - D_S^*\right\| \le (1 - \frac{3\gamma_t\sigma_r}{4})\left\|S_t S_t^\top - D_S^*\right\| + \frac{\gamma_t\sigma_r}{4}\left\|S_t S_t^\top - D_S^*\right\| \quad \text{(B.171)}$$

$$\le (1 - \frac{\gamma_t\sigma_r}{2})\left\|S_t S_t^\top - D_S^*\right\| \quad \text{(B.172)}$$

$$\le \max\{5(1 - \frac{c_\gamma\sigma_r^2}{2\sigma_1^2})^{t+1-\mathcal{T}_1'}, \frac{100(c_\rho\rho)^2\sigma_1}{\sigma_r}\}. \quad \text{(B.173)}$$

We used the induction hypothesis in the last inequality. By induction, inequality A.29 is proved. Moreover, $\mathcal{T}_2'$ is the smallest integer such that

$$5(1 - \frac{c_\gamma\sigma_r^2}{2\sigma_1^2})^{t-\mathcal{T}_1'} \le \frac{100(c_\rho\rho)^2\sigma_1}{\sigma_r}. \quad \text{(B.174)}$$

Therefore, the second claim in Proposition A.7 follows from A.29.

## B.8  Proof of Proposition A.8

We prove it by induction. For the ease of notation, we use index $t$ for $t \ge \mathcal{T}_1$ instead of $\mathcal{T}_1 + t$. When $t = \mathcal{T}_1$, A.36 holds by assumption. Now suppose A.36 holds for some $t \ge \mathcal{T}_1$. By induction hypothesis, we have

$$\left\|S_t T_t^\top\right\| \le 0.01\sigma_r. \quad \text{(B.175)}$$

Moreover,

$$\left\|S_t S_t^\top\right\| \le \|D_S^*\| + \left\|S_t S_t^\top - D_S^*\right\| \le 1.01\sigma_1. \quad \text{(B.176)}$$

Therefore, $\|S_t\| \le 2\sqrt{\sigma_1}$. Also,

$$\sigma_r(S_t S_t^\top) \ge \sigma_r(D_S^*) - \left\|S_t S_t^\top - D_S^*\right\| \ge \frac{\sigma_r}{2}. \quad \text{(B.177)}$$

Hence, $\sigma_r(S_t) \ge \sqrt{\frac{\sigma_r}{2}}$ and the conditions of Proposition A.4 and Proposition A.5 are satisfied. We consider $\left\|S_{t+1}S_{t+1}^\top - D_S^*\right\|$ and $\left\|S_{t+1}T_{t+1}^\top\right\|$ separately.

1. For $\left\|S_{t+1}S_{t+1}^\top - D_S^*\right\|$, we apply the same idea as proof of Proposition A.7 and write

$$S_{t+1}S_{t+1}^\top - D_S^* = \underbrace{\mathcal{M}_t(S_t)\mathcal{M}_t(S_t)^\top - D_S^*}_{Z_1} - \underbrace{\gamma_t(U^\top\Delta_t F_t \mathcal{M}_t(S_t)^\top + \mathcal{M}_t(S_t)F_t^\top\Delta_t^\top U)}_{Z_2}$$

$$\text{(B.178)}$$

$$+ \underbrace{\gamma_t^2 U^\top\Delta_t F_t F_t^\top\Delta_t^\top U}_{Z_3} \quad \text{(B.179)}$$

By Proposition A.4, we know

$$\|Z_1\| \le (1 - \frac{3\gamma_t\sigma_r}{4})\left\|S_t S_t^\top - D_S^*\right\| + 3\gamma_t\left\|S_t T_t^\top\right\|^2 \quad \text{(B.180)}$$

$$\le (1 - \frac{3\gamma_t\sigma_r}{4})\left\|S_t S_t^\top - D_S^*\right\| + 0.03\gamma_t\sigma_r\left\|S_t T_t^\top\right\| \quad \text{(B.181)}$$

$$\le (1 - \frac{3\gamma_t\sigma_r}{4} + 0.03\gamma_t\sigma_r)D_t. \quad \text{(B.182)}$$

On the other hand, By triangle inequality,

$$\|Z_2\| \le 2\gamma_t \left\|U^\top \Delta_t F_t \mathcal{M}_t(S_t)^\top\right\| \tag{B.183}$$

$$\le 2\gamma_t \|\Delta_t\| \|US_t + VT_t\| \|\mathcal{M}_t(S_t)\| \tag{B.184}$$

$$\le 18\gamma_t\sigma_1 \|\Delta_t\| \tag{B.185}$$

$$\overset{(\sharp)}{\le} 18\gamma_t\sigma_1\delta\sqrt{k+r} \left\|F_tF_t^\top - X_\natural\right\| \tag{B.186}$$

$$\tag{B.187}$$

Here $(\sharp)$ follows from A.8. By lemma I.5, we see that

$$\left\|F_tF_t^\top - X_\natural\right\| \le \left\|S_tS_t^\top - D_S^*\right\| + 2\left\|S_tT_t^\top\right\| + \left\|T_tT_t^\top\right\| \tag{B.188}$$

$$\le \frac{3\sigma_r}{100} + \frac{\sigma_r}{100} \tag{B.189}$$

$$\le \frac{4\sigma_r}{100}. \tag{B.190}$$

Hence, we obtain

$$\|Z_2\| \le \frac{72}{100}\gamma_t\sigma_r\delta\sqrt{k+r}\sigma_1. \tag{B.191}$$

Similarly,

$$\|Z_3\| \le \gamma_t^2 \|\Delta_t\|^2 \|F_t\|^2 \tag{B.192}$$

$$\le 9\sigma_1\gamma_t^2(\delta\sqrt{k+r})^2 \left\|F_tF_t^\top - X_\natural\right\|^2 \tag{B.193}$$

$$\le 9\sigma_1\gamma_t^2(\delta\sqrt{k+r})^2\left(\frac{4\sigma_r}{100}\right)^2 \tag{B.194}$$

$$\le \frac{1}{100}\gamma_t\sigma_r\delta\sqrt{k+r}\sigma_1. \tag{B.195}$$

In the last inequality, we used our assumption that $\gamma_t\sigma_r \le \gamma_t\sigma_1 \le 0.01$ and $\delta\sqrt{k+r} \le 0.001$. Combining, we obtain

$$\left\|S_{t+1}S_{t+1}^\top - D_S^*\right\| \le (1 - \frac{\gamma_t\sigma_r}{2})D_t + \gamma_t\sigma_r\delta\sqrt{k+r}\sigma_1. \tag{B.196}$$

We consider two cases:

- $D_t \le 3\delta\sqrt{k+r}\sigma_1$. In this case, we simply have

$$\left\|S_{t+1}S_{t+1}^\top - D_S^*\right\| \le D_t + 3\gamma_t\sigma_r\delta\sqrt{k+r}\sigma_1 \le D_t + \delta\sqrt{k+r}\sigma_1 \le 10\delta\sqrt{k+r}\sigma_1. \tag{B.197}$$

- $3\delta\sqrt{k+r}\sigma_1 < D_t \le 10\delta\sqrt{k+r}\sigma_1$. In this case, we clearly have

$$\gamma_t\delta\sqrt{k+r}\sigma_1\sigma_r \le \frac{\gamma_t\sigma_r}{3}D_t. \tag{B.198}$$

Consequently,

$$\left\|S_{t+1}S_{t+1}^\top - D_S^*\right\| \le (1 - \frac{\gamma_t\sigma_r}{6\sigma_1})D_t \le \max\left\{ \left(1 - \frac{c_\gamma\sigma_r^2}{6\sigma_1^2}\right)^{t+1-\mathcal{T}_1} \cdot \frac{\sigma_r}{10}, 10\delta\sqrt{k+r}\sigma_1 \right\}. \tag{B.199}$$

Here we used the induction hypothesis on $D_t$.

2. For $\left\|S_{t+1}T_{t+1}^\top\right\|$, we can expand it and get

$$S_{t+1}T_{t+1}^\top = (\mathcal{M}_t(S_t) + \gamma_t U^\top \Delta_t F_t)(\mathcal{N}_t(T_t) + \gamma_t V^\top \Delta_t F_t)^\top \tag{B.200}$$

$$= \underbrace{\mathcal{M}_t(S_t)\mathcal{N}_t(T_t)^\top}_{Z_4} + \underbrace{\gamma_t U^\top \Delta_t F_t \mathcal{N}_t(T_t) + \gamma_t \mathcal{M}_t(S_t)F_t^\top \Delta_t^\top V}_{Z_5} \tag{B.201}$$

$$+ \underbrace{\gamma_t^2 U^\top \Delta_t F_t F_t^\top \Delta_t^\top V}_{Z_6}. \tag{B.202}$$

By assumption, we have

$$\left\| S_t S_t^\top - D_S^* \right\| \leq D_t \tag{B.203}$$

$$\leq \max\{\frac{\sigma_r}{100}, 10\delta\sqrt{k+r}\sigma_1\} \tag{B.204}$$

$$\leq \frac{\sigma_r}{100}. \tag{B.205}$$

By Proposition A.4, we know

$$\|Z_4\| \leq (1 - \frac{\gamma_t \sigma_r}{3}) \left\| S_t T_t^\top \right\| \leq (1 - \frac{\gamma_t \sigma_r}{3}) D_t \tag{B.206}$$

On the other hand, it's easy to see the $\|\mathcal{M}_t(S_t)\| \leq 3\sqrt{\sigma_1}$ and $\|\mathcal{N}_t(T_t)\| \leq \sqrt{\sigma_1}$, by triangle inequality and the same argument as $\left\| S_{t+1} S_{t+1}^\top - D_S^* \right\|$,

$$\|Z_5\| \leq \gamma_t \left( \|F_t\| \|\mathcal{N}_t(T_t)\| + \|F_t\| \|\mathcal{M}_t(S_t)\| \right) \|\Delta_t\| \tag{B.207}$$

$$\leq 12\gamma_t \sigma_1 \|\Delta_t\| \tag{B.208}$$

$$\leq 12\gamma_t \sigma_1 \delta\sqrt{k+r} \left\| F_t F_t^\top - X_\natural \right\| \tag{B.209}$$

$$\leq \frac{48}{100}\gamma_t \sigma_r \delta\sqrt{k+r}\sigma_1. \tag{B.210}$$

We used $\left\| F_t F_t^\top - X_\natural \right\| \leq \frac{4\sigma_r}{100}$, which was proved above. Similar as calculation for $\left\| S_{t+1} S_{t+1}^\top - D_S^* \right\|$, we have

$$\|Z_6\| \leq \frac{1}{100}\gamma_t \sigma_r \delta\sqrt{k+r}\sigma_1. \tag{B.211}$$

Combining, we obtain

$$\left\| S_{t+1} T_{t+1}^\top \right\| \leq \|Z_4\| + \|Z_5\| + \|Z_6\| \tag{B.212}$$

$$\leq \left(1 - \frac{\gamma_t \sigma_r}{3}\right) D_t + \gamma_t \sigma_r \delta\sqrt{k+r}\sigma_1. \tag{B.213}$$

We consider two cases:

- $D_t \leq 6\delta\sqrt{k}\sigma_1$. In this case, we simply have

$$\left\| S_{t+1} T_{t+1}^\top \right\| \leq D_t + \gamma_t \sigma_r \delta\sqrt{k+r}\sigma_1 \leq D_t + \delta\sqrt{k+r}\sigma_1 \leq 10\delta\sqrt{k+r}\sigma_1. \tag{B.214}$$

- $6\delta\sqrt{k+r}\sigma_1 < D_t \leq 10\delta\sqrt{k+r}\sigma_1$. In this case, we clearly have

$$\gamma_t \sigma_r \delta\sqrt{k+r}\sigma_1 \leq \frac{\gamma_t \sigma_r}{6} D_t. \tag{B.215}$$

Consequently,

$$\left\| S_{t+1} T_{t+1}^\top \right\| \leq (1 - \frac{\gamma_t \sigma_r}{6}) D_t \leq \max\left\{ \left(1 - \frac{c_\gamma \sigma_r^2}{6\sigma_1^2}\right)^{t+1-\mathcal{T}_1} \cdot \frac{\sigma_r}{10}, 10\delta\sqrt{k+r}\sigma_1 \right\}. \tag{B.216}$$

Here we used the induction hypothesis on $D_t$.

Combining, we see that

$$D_{t+1} \leq \max\left\{ \left(1 - \frac{c_\gamma \sigma_r^2}{6\sigma_1^2}\right)^{t+1-\mathcal{T}_1} \cdot \frac{\sigma_r}{10}, 10\delta\sqrt{k+r}\sigma_1 \right\}. \tag{B.217}$$

So the induction step is proved. Note that $\mathcal{T}_2$ is chosen to be the smallest integer $t$ that

$$\left(1 - \frac{c_\gamma \sigma_r^2}{6\sigma_1^2}\right)^{t-\mathcal{T}_1} \cdot \frac{\sigma_r}{10} \leq 10\delta\sqrt{k+r}\sigma_1, \tag{B.218}$$

the second part of Proposition A.8 follows.

## B.9 Proof of Proposition A.9

The proof is inspired by [15]. By our assumption that $E_t \leq 0.01\sigma_r$, we have

$$\left\|S_t S_t^\top\right\| \leq \left\|S_t S_t^\top - D_S^*\right\| + \left\|D_S^*\right\| \leq 1.01\sigma_1. \tag{B.219}$$

As a result, $\|S_t\| \leq 2\sqrt{\sigma_1}$. Similarly,

$$\|T_t\| \leq \sqrt{\left\|T_t T_t^\top\right\|} \leq 0.1\sqrt{\sigma_r}. \tag{B.220}$$

Moreover,

$$\sigma_r(S_t S_t^\top) \geq \sigma_r(D_S^*) - \left\|S_t S_t^\top - D_S^*\right\| \geq \frac{\sigma_r}{2}. \tag{B.221}$$

We obtain

$$\sigma_r(S_t) \geq \sqrt{\frac{\sigma_r}{2}}. \tag{B.222}$$

Thus, $S_t, T_t$ satisfy all the conditions in Proposition A.4 and Proposition A.5. We will bound $\left\|S_{t+1} S_{t+1}^\top - D_S^*\right\|$, $\left\|S_{t+1} T_{t+1}^\top\right\|$, $\left\|T_{t+1} T_{t+1}^\top\right\|$ separately.

- $\left\|S_{t+1} S_{t+1}^\top - D_S^*\right\|$. Simple algebra yields

$$S_{t+1} S_{t+1}^\top - D_S^* = \underbrace{\mathcal{M}_t(S_t)\mathcal{M}_t(S_t)^\top - D_S^*}_{Z_1} + \underbrace{\gamma_t(U^\top \Delta_t F_t \mathcal{M}_t(S_t)^\top + \mathcal{M}_t(S_t)F_t^\top \Delta_t^\top U)}_{Z_2}$$
$$\tag{B.223}$$

$$+ \underbrace{\gamma_t^2 U^\top \Delta_t F_t F_t^\top \Delta_t^\top U}_{Z_3} \tag{B.224}$$

By Proposition A.4, we obtain

$$\|Z_1\| \leq (1 - \frac{3\gamma_t\sigma_r}{4})\left\|S_t S_t^\top - D_S^*\right\| + 3\gamma_t\left\|S_t T_t^\top\right\|^2 \tag{B.225}$$

$$\overset{(\sharp)}{\leq} (1 - \frac{3\gamma_t\sigma_r}{4})\left\|S_t S_t^\top - D_S^*\right\| + 0.03\gamma_t\sigma_r\left\|S_t T_t^\top\right\| \tag{B.226}$$

$$\leq (1 - \frac{3\gamma_t\sigma_r}{4} + 0.03\gamma_t\sigma_r)E_t. \tag{B.227}$$

In $(\sharp)$, we used our assumption that $\left\|S_t T_t^\top\right\| \leq 0.01\sigma_r$. On the other hand, it's easy to see $\|\mathcal{M}_t(S_t)\| \leq 3\sqrt{\sigma_1}$ by its definition and the fact that $\|S_t\| \leq 2\sqrt{\sigma_r}$. By triangle inequality,

$$\|Z_2\| \leq 2\gamma_t\left\|U^\top \Delta_t F_t \mathcal{M}_t(S_t)^\top\right\| \tag{B.228}$$

$$\leq 2\gamma_t\|\Delta_t\|\|US_t + VT_t\|\|\mathcal{M}_t(S_t)\| \tag{B.229}$$

$$\leq 18\gamma_t\sigma_1\|\Delta_t\| \tag{B.230}$$

$$\overset{(\sharp)}{\leq} 18\gamma_t\sigma_1\delta\sqrt{k+r}\left\|F_t F_t^\top - X_\natural\right\| \tag{B.231}$$

$$\overset{(\star)}{\leq} 0.018\gamma_t\sigma_r\left\|F_t F_t^\top - X_\natural\right\| \tag{B.232}$$

Here $(\sharp)$ follows from A.8, $(\star)$ follows from our assumption that $\delta\sqrt{k+r} \leq \frac{0.001\sigma_r}{\sigma_1}$. By lemma I.5, we see that

$$\left\|F_t F_t^\top - X_\natural\right\| \leq \left\|S_t S_t^\top - D_S^*\right\| + 2\left\|S_t T_t^\top\right\| + \left\|T_t T_t^\top\right\| \tag{B.233}$$

$$\leq 4E_t \tag{B.234}$$

Hence, we obtain

$$\|Z_2\| \leq 0.1\gamma_t\sigma_r E_t. \tag{B.235}$$

Similarly,

$$\|Z_3\| \leq \gamma_t^2\|\Delta_t\|^2\|F_t\|^2 \tag{B.236}$$

$$\leq 9\sigma_1\gamma_t^2(\delta\sqrt{k+r})^2\left\|F_t F_t^\top - X_\natural\right\|^2 \tag{B.237}$$

$$\leq 144\sigma_1\gamma_t^2(\delta\sqrt{k+r})^2 E_t^2 \tag{B.238}$$

$$\leq 0.1\gamma_t\sigma_r E_t. \tag{B.239}$$

In the last inequality, we used our assumption that $\delta\sqrt{k+r} \leq \frac{0.001\sigma_r}{\sigma_1} \leq 0.001$, $\gamma_t \leq \frac{0.01}{\sigma_1}$ and $\|E_t\| \leq 0.01\sigma_r$. Combining, we obtain

$$\left\|S_{t+1}S_{t+1}^\top - D_S^*\right\| \leq \|Z_1\| + \|Z_2\| + \|Z_3\| \tag{B.240}$$

$$\leq (1 - \frac{\gamma_t\sigma_r}{2})E_t \tag{B.241}$$

- $\left\|S_{t+1}T_{t+1}^\top\right\|$. We can expand it and get

$$S_{t+1}T_{t+1}^\top = (\mathcal{M}_t(S_t) + \gamma_t U^\top \Delta_t F_t)(\mathcal{N}_t(T_t) + \gamma_t V^\top \Delta_t F_t)^\top \tag{B.242}$$

$$= \underbrace{\mathcal{M}_t(S_t)\mathcal{N}_t(T_t)^\top}_{Z_4} + \underbrace{\gamma_t U^\top \Delta_t F_t \mathcal{N}_t(T_t) + \gamma_t \mathcal{M}_t(S_t)F_t^\top \Delta_t^\top V}_{Z_5} \tag{B.243}$$

$$+ \underbrace{\gamma_t^2 U^\top \Delta_t F_t F_t^\top \Delta_t^\top V}_{Z_6}. \tag{B.244}$$

By Proposition A.4, we know

$$\|Z_4\| \leq (1 - \frac{\gamma_t\sigma_r}{3})\left\|S_t T_t^\top\right\| \leq (1 - \frac{\gamma_t\sigma_r}{3})E_t \tag{B.245}$$

On the other hand, we see that $\|\mathcal{M}_t(S_t)\| \leq 3\sqrt{\sigma_1}$ and $\|\mathcal{N}_t(T_t)\| \leq \sqrt{\sigma_1}$ (by bound on $S_t$ and $T_t$ and the update rule), by triangle inequality and the same argument as $\left\|S_{t+1}S_{t+1}^\top - D_S^*\right\|$,

$$\|Z_5\| \leq \gamma_t (\|F_t\|\|\mathcal{N}_t(T_t)\| + \|F_t\|\|\mathcal{M}_t(S_t)\|)\|\Delta_t\| \tag{B.246}$$

$$\leq 12\gamma_t\sigma_1\|\Delta_t\| \tag{B.247}$$

$$\leq 12\gamma_t\sigma_1\delta\sqrt{k+r}\left\|F_t F_t^\top - X_\natural\right\| \tag{B.248}$$

$$\leq 0.05\gamma_t\sigma_r E_t. \tag{B.249}$$

Same as calculation for $\left\|S_{t+1}S_{t+1}^\top - D_S^*\right\|$, we have

$$\|Z_6\| \leq 0.1\gamma_t\sigma_r E_t. \tag{B.250}$$

Combining, we obtain

$$\left\|S_{t+1}T_{t+1}^\top\right\| \leq \|Z_4\| + \|Z_5\| + \|Z_6\| \tag{B.251}$$

$$\leq \left(1 - \frac{\gamma_t\sigma_r}{6}\right)E_t. \tag{B.252}$$

- $\left\|T_{t+1}T_{t+1}^\top\right\|$. We expand it and obtain

$$T_{t+1}T_{t+1}^\top = (\mathcal{N}_t(T_t) + \gamma_t V^\top \Delta_t F_t)(\mathcal{N}_t(T_t) + \gamma_t V^\top \Delta_t F_t)^\top \tag{B.253}$$

$$\leq \underbrace{\mathcal{N}_t(T_t)\mathcal{N}_t(T_t)^\top}_{Z_7} + \underbrace{\gamma_t V^\top \Delta_t F_t \mathcal{N}_t(T_t)^\top + \gamma_t \mathcal{N}_t(T_t)F_t^\top \Delta_t^\top V}_{Z_8} \tag{B.254}$$

$$+ \underbrace{\gamma_t^2 V^\top \Delta_t F_t F_t^\top \Delta_t^\top V}_{Z_9} \tag{B.255}$$

By Proposition A.4,

$$\|Z_7\| \leq \left\|T_t T_t^\top\right\|(1 - 2\gamma_t\left\|T_t T_t^\top\right\|) \leq E_t(1 - 2\gamma_t E_t). \tag{B.256}$$

The last inequality follows from the fact that $x \to x(1 - 2\gamma_t x)$ is non-decreasing on interval $[0, \frac{1}{4\gamma_t}]$. On the other hand,

$$V^\top \Delta_t F_t \mathcal{N}_t(T_t)^\top = V^\top \Delta_t(US_t + VT_t)\mathcal{N}_t(T_t)^\top \tag{B.257}$$

$$= V^\top \Delta_t US_t \mathcal{N}_t(T_t)^\top + V^\top \Delta_t VT_t \mathcal{N}_t(T_t)^\top \tag{B.258}$$

By Proposition A.5, we obtain

$$\left\|V^\top\Delta_t F_t \mathcal{N}_t(T_t)^\top\right\| \leq \left\|V^\top\Delta_t U S_t \mathcal{N}_t(T_t)^\top\right\| + \left\|V^\top\Delta_t V T_t \mathcal{N}_t(T_t)^\top\right\| \tag{B.259}$$

$$\leq \left(\left\|S_t \mathcal{N}_t(T_t)^\top\right\| + \left\|T_t \mathcal{N}_t(T_t)^\top\right\|\right)\|\Delta_t\| \tag{B.260}$$

$$\leq \left(\left\|S_t T_t^\top\right\| + \left\|T_t T_t^\top\right\|\right)\delta\sqrt{k+r}\left\|F_t F_t^\top - X_\natural\right\| \tag{B.261}$$

$$\leq 8\delta\sqrt{k+r}E_t^2. \tag{B.262}$$

$$\leq 0.01 E_t^2 \tag{B.263}$$

Consequently,

$$\|Z_8\| \leq 2\gamma_t \left\|V^\top\Delta_t F_t \mathcal{N}_t(T_t)^\top\right\| \leq 0.02\gamma_t E_t^2. \tag{B.264}$$

Furthermore,

$$\|Z_9\| \leq \gamma_t^2 \|F_t\|^2 \|\Delta_t\|^2 \tag{B.265}$$

$$\leq 9\gamma_t^2\sigma_1(\delta\sqrt{k+r})^2\left\|F_t F_t^\top - X_\natural\right\|^2 \tag{B.266}$$

$$\leq 144\gamma_t^2\sigma_1(\delta\sqrt{k+r})^2 E_t^2 \tag{B.267}$$

$$\leq 0.1\gamma_t E_t^2. \tag{B.268}$$

In the last inequality, we used our assumption that $\gamma_t \leq 0.01\sigma_1$ and $\delta\sqrt{k+r} \leq 0.001$. Combining, we obtain

$$\left\|T_{t+1}T_{t+1}^\top\right\| \leq E_t(1 - \gamma_t E_t). \tag{B.269}$$

The result follows.

## B.10   Proof of Proposition A.10

The proof of this proposition has lots of overlap with Proposition A.9. By our assumption that $E_t \leq 0.01\sigma_r$, we have

$$\left\|S_t S_t^\top\right\| \leq \left\|S_t S_t^\top - D_S^*\right\| + \|D_S^*\| \leq 1.01\sigma_1. \tag{B.270}$$

As a result, $\|S_t\| \leq 2\sqrt{\sigma_1}$. Similarly,

$$\|T_t\| \leq \sqrt{\left\|T_t T_t^\top\right\|} \leq 0.1\sqrt{\sigma_r}. \tag{B.271}$$

Moreover,

$$\sigma_r(S_t S_t^\top) \geq \sigma_r(D_S^*) - \left\|S_t S_t^\top - D_S^*\right\| \geq \frac{\sigma_r}{2}. \tag{B.272}$$

We obtain

$$\sigma_r(S_t) \geq \sqrt{\frac{\sigma_r}{2}}. \tag{B.273}$$

Thus, $S_t, T_t$ satisfy all the conditions in Proposition A.4 and Proposition A.5. We will bound $\left\|S_{t+1}S_{t+1}^\top - D_S^*\right\|, \left\|S_{t+1}T_{t+1}^\top\right\|, \left\|T_{t+1}T_{t+1}^\top\right\|$ separately. Note that the proof of Proposition A.9 doesn't use $k > r$, so it also holds for the case when $k = r$. So, we already have

$$\left\|S_{t+1}S_{t+1}^\top - D_S^*\right\| \leq (1 - \frac{\gamma_t\sigma_r}{2})E_t \tag{B.274}$$

and

$$\left\|S_{t+1}T_{t+1}^\top\right\| \leq (1 - \frac{\gamma_t\sigma_r}{3})E_t. \tag{B.275}$$

Next, we obtain a better bound for $\left\|T_{t+1}T_{t+1}^\top\right\|$. We expand $T_{t+1}T_{t+1}^\top$ and obtain

$$T_{t+1}T_{t+1}^\top = (\mathcal{N}_t(T_t) + \gamma_t V^\top\Delta_t F_t)(\mathcal{N}_t(T_t) + \gamma_t V^\top\Delta_t F_t)^\top \tag{B.276}$$

$$= \underbrace{\mathcal{N}_t(T_t)\mathcal{N}_t(T_t)^\top}_{Z_1} + \underbrace{\gamma_t V^\top\Delta_t F_t \mathcal{N}_t(T_t)^\top + \gamma_t \mathcal{N}_t(T_t)F_t^\top\Delta_t^\top V}_{Z_2} \tag{B.277}$$

$$+ \underbrace{\gamma_t^2 V^\top\Delta_t F_t F_t^\top\Delta_t^\top V}_{Z_3} \tag{B.278}$$

By definition,
$$\mathcal{N}_t(T_t) = T_t - \gamma_t(T_t T_t^\top T_t + T_t S_t^\top S_t). \tag{B.279}$$
Plug this into $\mathcal{N}_t(T_t)\mathcal{N}_t(T_t)^\top$, we obtain
$$Z_1 = \mathcal{N}_t(T_t)\mathcal{N}_t(T_t)^\top \tag{B.280}$$
$$= \left(T_t - \gamma_t(T_t T_t^\top T_t + T_t S_t^\top S_t)\right)\left(T_t - \gamma_t(T_t T_t^\top T_t + T_t S_t^\top S_t)\right)^\top \tag{B.281}$$
$$= Z_4 + Z_5, \tag{B.282}$$
where
$$Z_4 = T_t T_t^\top - 2\gamma_t T_t T_t^\top T_t T_t^\top - \gamma_t T_t S_t^\top S_t T_t^\top \tag{B.283}$$
and
$$Z_5 = -\gamma_t T_t S_t^\top S_t T_t^\top + \gamma_t^2 (T_t T_t^\top T_t + T_t S_t^\top S_t)(T_t T_t^\top T_t + T_t S_t^\top S_t)^\top. \tag{B.284}$$
We bound each of them separately. Since $k = r$, $S_t^\top S_t$ is a $r$-by-$r$. Moreover,
$$\sigma_r(S_t^\top S_t) = \sigma_r(S_t)^2 \geq \frac{\sigma_r}{2}, \tag{B.285}$$
By $\gamma_t \leq \frac{0.01}{\sigma_1}$,
$$\left\| I - \gamma_t S_t^\top S_t - 2\gamma_t T_t T_t^\top \right\| \leq \left\| I - \gamma_t S_t^\top S_t \right\| \tag{B.286}$$
$$\leq 1 - \frac{\gamma_t \sigma_r}{2}. \tag{B.287}$$
Consequently,
$$\|Z_4\| = \left\| T_t(I - \gamma_t S_t^\top S_t - 2\gamma_t T_t^\top T_t)T_t^\top \right\| \tag{B.288}$$
$$\leq \|T_t\|^2 \left\| (I - \gamma_t S_t^\top S_t - 2\gamma_t T_t^\top T_t) \right\| \tag{B.289}$$
$$\leq (1 - \frac{\gamma_t \sigma_r}{2}) \|T_t\|^2. \tag{B.290}$$
In addition,
$$Z_5 = -\gamma_t T_t S_t^\top S_t T_t^\top + \gamma_t^2 \left[ T_t T_t^\top \left( T_t S_t^\top S_t T_t^\top \right) + \left( T_t S_t^\top S_t T_t^\top \right) T_t T_t^\top \right] + \gamma_t^2 T_t S_t^\top S_t S_t^\top S_t T_t^\top \tag{B.291}$$
$$\preceq (-\gamma_t + \frac{2}{100}\gamma_t^2 \sigma_r + 4\sigma_1 \gamma_t^2) T_t S_t^\top S_t T_t^\top \tag{B.292}$$
$$\preceq 0 \tag{B.293}$$
Combining, we obtain
$$\left\| \mathcal{N}_t(T_t)\mathcal{N}_t(T_t)^\top \right\| \leq \|Z_4\| \leq (1 - \frac{\gamma_t \sigma_r}{2}) \left\| T_t T_t^\top \right\|. \tag{B.294}$$
On the other hand, we see that $\|\mathcal{M}_t(S_t)\| \leq 3\sqrt{\sigma_1}$ and $\|\mathcal{N}_t(T_t)\| \leq \sqrt{\sigma_1}$(by bound on $S_t$ and $T_t$ and the update rule). As a result,
$$\left\| V^\top \Delta_t F_t \mathcal{N}_t(T_t)^\top \right\| \leq \|F_t\| \|\mathcal{N}_t(T_t)\| \|\Delta_t\| \tag{B.295}$$
$$\leq (\|S_t\| + \|T_t\|) \|\mathcal{N}_t(T_t)\| \delta\sqrt{k+r} \left\| F_t F_t^\top - X_\natural \right\| \tag{B.296}$$
$$\leq 3\sigma_1 \delta\sqrt{k+r} \left\| F_t F_t^\top - X_\natural \right\|. \tag{B.297}$$
$$\leq 12\sigma_1 \delta\sqrt{k+r} E_t. \tag{B.298}$$
Consequently,
$$\|Z_2\| \leq 2\gamma_t \left\| V^\top \Delta_t F_t \mathcal{N}_t(T_t)^\top \right\| \leq 0.03\gamma_t \sigma_r E_t \tag{B.299}$$
Furthermore,
$$\|Z_3\| \leq \gamma_t^2 \|F_t\|^2 \|\Delta_t\|^2 \tag{B.300}$$
$$\leq 9\gamma_t^2 \sigma_1 (\delta\sqrt{k+r})^2 \left\| F_t F_t^\top - X_\natural \right\|^2 \tag{B.301}$$
$$\leq 144\gamma_t^2 \sigma_1 (\delta\sqrt{k+r})^2 E_t^2 \tag{B.302}$$
$$\leq 0.01\gamma_t \sigma_r E_t. \tag{B.303}$$

In the last inequality, we used our assumption that $\gamma_t \leq 0.01\sigma_1$ and $\delta\sqrt{k+r} \leq 0.001$. Combining, we obtain

$$\left\|T_{t+1}T_{t+1}^\top\right\| \leq \|Z_1\| + \|Z_2\| + \|Z_3\| \tag{B.304}$$

$$\leq (1 - \frac{\gamma_t \sigma_r}{3})\left\|T_t T_t^\top\right\| \tag{B.305}$$

## C  Proof of RDPP

Throughout this section, we denote

$$\mathbb{S} := \{X \in \mathcal{S}^{d\times d}\colon \|X\|_{\mathrm{F}} = 1\}, \quad \mathbb{S}_r := \{X \in \mathcal{S}^{d\times d}\colon \|X\|_{\mathrm{F}} = 1, \mathrm{rank}(X) \leq r\}.$$

Here we split the Proposition 2.2 into two parts and prove them separately. For the ease of notation, we use $r$ to denote the rank, instead of $k'$.

**Proposition C.1.** *Assume that the sensing matrix $A_i \overset{i.i.d.}{\sim} GOE(d)$,[15] and the corruption is from model 2. Then RDPP holds with parameters $(r, \delta)$ and a scaling function $\psi(X) = \frac{1}{m}\sum_{i=1}^m \sqrt{\frac{2}{\pi}}\left(1 - p + p\mathbb{E}_{s_i \sim \mathbb{P}_i}\left[\exp(-\frac{s_i^2}{2\|X\|_{\mathrm{F}}^2})\right]\right)$ with probability at least $1 - Ce^{-cm\delta^4}$, given $m \gtrsim \frac{dr\left(\log(\frac{1}{\delta})\vee 1\right)}{\delta^4}$.*

**Proposition C.2.** *Assume that the sensing matrices $\{A_i\}_{i=1}^m$ have i.i.d. standard Gaussian entries, and the corruption is from model I. Moreover, we modify function $\mathrm{sign}(x)$ such that $\mathrm{sign}(x) = \begin{cases} \{-1\} & x < 0 \\ \{-1, 1\} & x = 0 \\ \{1\} & x > 0 \end{cases}$ . Then, RDPP-II holds with parameter $(r, \delta + 3\sqrt{\frac{dp}{m}} + 3p)$ and a scaling function $\psi(X) = \sqrt{\frac{2}{\pi}}$ with probability at least $1 - \exp(-(pm+d)) - \exp(-c'm\delta^4)$, given $m \gtrsim \frac{dr\left(\log(\frac{1}{\delta})\vee 1\right)}{\delta^4}$.*

### C.1  Proof of Proposition C.1

In the probability bounds that we obtained, the $c$ might be different from bounds to bounds, but they are all universal constants.

**Lemma C.3.** *Suppose that we are under Model 2. Then, for every nonzero $X \in \mathbb{S}^{d\times d}$, and every $D \in \mathcal{D}(X)$, the expectation $\mathbb{E}[D]$ is*

$$\mathbb{E}[D] = \psi(X)\frac{X}{\|X\|_{\mathrm{F}}}, \text{ where } \psi(X) = \frac{1}{m}\sum_{i=1}^m \sqrt{\frac{2}{\pi}}\left(1 - p + p\mathbb{E}_{s_i \sim \mathbb{P}_i}\left[e^{-s_i^2/2\|X\|_{\mathrm{F}}^2}\right]\right). \tag{C.1}$$

*Proof.* We may drop the subscript under expectation when the distribution is clear. Firstly, we show that for any $X, Y \in \mathbb{S}^{d\times d}$, if $s$ follows distribution $\mathbb{P}$, $A$ is GOE matrix and they are independent, then

$$\mathbb{E}\left[\mathrm{sign}(\langle A, X\rangle - s)\langle A, Y\rangle\right] = \sqrt{\frac{2}{\pi}}\mathbb{E}\left[e^{-s^2/2\|X\|_{\mathrm{F}}^2}\right]\left\langle \frac{X}{\|X\|_{\mathrm{F}}}, Y\right\rangle. \tag{C.2}$$

In this section, $\mathrm{sign}(\langle A, x\rangle - s)$ should be thought of as any element chosen from the corresponding set. There is ambiguity when $\langle A, x\rangle - s = 0$, but this happens with probability 0, so it won't affect the result. Without loss of generality, we assume $\|X\|_{\mathrm{F}} = \|Y\|_{\mathrm{F}} = 1$. To leverage the fact that $A$ is GOE matrix, we denote $u = \langle A, X\rangle$, $v = \langle A, Y\rangle$ and $\rho = \mathrm{cov}(u, v)$. Simple calculation yields $u \sim N(0, 1)$, $v \sim N(0, 1)$ and $\rho = \langle X, Y\rangle$. By coupling, we can write $v = \rho u + \sqrt{1 - \rho^2}w$, where $w$ is another standard Gaussian independent of others. Using the definition of $u, v, \rho, w$, we have

$$\mathbb{E}\left[\mathrm{sign}(\langle A, X\rangle - s)\langle A, Y\rangle\right] = \mathbb{E}\left[\mathrm{sign}(u - s)v\right] = \rho\mathbb{E}\left[\mathrm{sign}(u - s)u\right]. \tag{C.3}$$

---

[15]Gaussian orthogonal ensemble(GOE): $A$ is symmetric with $A_{ij} = A_{ji} \sim N(0, \frac{1}{2})$ for $i \neq j$ and $A_{ii} \sim N(0, 1)$ independently.

We continue the above equality using the properties of Gaussian:

$$\rho\mathbb{E}\left[\mathrm{sign}(u-s)u\right] = \rho\mathbb{E}_s\left[\int_s^{+\infty} u\frac{1}{\sqrt{2\pi}}e^{-u^2/2}du - \int_{-\infty}^s u\frac{1}{\sqrt{2\pi}}e^{-u^2/2}du\right] \tag{C.4}$$

$$\overset{(a)}{=}\rho\mathbb{E}_s\left[\int_s^{+\infty} u\frac{1}{\sqrt{2\pi}}e^{-u^2/2}du + \int_{-s}^{+\infty} u\frac{1}{\sqrt{2\pi}}e^{-u^2/2}du\right] \tag{C.5}$$

$$\overset{(b)}{=}2\rho\mathbb{E}_s\left[\int_{|s|}^{+\infty} u\frac{1}{\sqrt{2\pi}}e^{-u^2/2}du\right] \tag{C.6}$$

$$=\sqrt{\frac{2}{\pi}}\rho\mathbb{E}_s\left[\int_{|s|}^{+\infty} d(-e^{-u^2/2})\right] = \sqrt{\frac{2}{\pi}}\rho\mathbb{E}_s\left[e^{-s^2/2}\right]. \tag{C.7}$$

Here, in the steps $(a)$, we do a change of variable $u \mapsto -u$. In the step $(b)$, we use the fact that the density of standard Gaussian is symmetric. Recall that $\rho = \langle X, Y\rangle$. Hence, the equation (C.2) follows from (C.3) - (C.7). Since it holds for all symmetric $Y$, we obtain

$$\mathbb{E}\left[\mathrm{sign}(\langle A, X\rangle - s)A\right] = \sqrt{\frac{2}{\pi}}\mathbb{E}\left[e^{-s^2/2\|X\|_{\mathrm{F}}^2}\right]\frac{X}{\|X\|_{\mathrm{F}}}. \tag{C.8}$$

On the other hand, if we apply the above result to the case when $s \equiv 0$, we get

$$\mathbb{E}\left[\mathrm{sign}(\langle A, X\rangle)A\right] = \sqrt{\frac{2}{\pi}}\frac{X}{\|X\|_{\mathrm{F}}}. \tag{C.9}$$

When $s_i$'s are form model 2, by tower property and results above,

$$\mathbb{E}\left[\mathrm{sign}(\langle A_i, X\rangle - s_i)A_i\right] = \mathbb{E}\left[\mathbb{E}[\mathrm{sign}(\langle A_i, X\rangle - s_i)A_i \mid s_i]\right] \tag{C.10}$$

$$= (1-p)\mathbb{E}\left[\mathrm{sign}(\langle A_i, X\rangle)A_i\right] + p\mathbb{E}_{s_i \sim \mathbb{P}_i, A_i}\left[\mathrm{sign}(\langle A_i, X\rangle - s_i)A_i\right] \tag{C.11}$$

$$= \sqrt{\frac{2}{\pi}}\left((1-p) + p\mathbb{E}\left[e^{-s^2/2\|X\|_{\mathrm{F}}^2}\right]\right)\frac{X}{\|X\|_{\mathrm{F}}} \tag{C.12}$$

The lemma follows from the linearity of expectation. $\qquad\square$

Lemma C.3 is an analogue of [13, Lemma 3]. Note that the function $\psi$ is not necessarily the quantity $\sqrt{\frac{2}{\pi}}\left((1-p) + p\mathbb{E}\left[e^{-s_i^2/2\|X\|_{\mathrm{F}}^2}\right]\right)\frac{X}{\|X\|_{\mathrm{F}}}$, which appears in [13, Lemma 3], since the corruptions are not assumed to be i.i.d in this paper.

Next, we prove a probability bound that holds for any fixed $X, Y \in \mathbb{S}$.

**Lemma C.4.** *Under Model 2, there exists a universal constant $c$ such that for any $\delta > 0, X \in \mathbb{S}, Y \in \mathbb{S}$, with probablity at most $2e^{-cm\delta^2}$, the following event happens*

$$\left|\frac{1}{m}\sum_{i=1}^m \mathrm{sign}\left(\langle A_i, X\rangle - s_i\right)\langle A_i, Y\rangle - \psi(X)\langle X, Y\rangle\right| > \delta, \tag{C.13}$$

*where $\psi(X) = \frac{1}{m}\sum_{i=1}^m \sqrt{\frac{2}{\pi}}\left(1 - p + p\mathbb{E}_{s_i \sim \mathbb{P}_i}\left[e^{-s_i^2/2\|X\|_{\mathrm{F}}^2}\right]\right)$.*

*Proof.* We first show that $\mathrm{sign}(\langle A_i, X\rangle - s_i)\langle A_i, Y\rangle$ is a sub-Gaussian random variable. Let consider the Orlicz norm [11] with $\psi_2(x) = e^{x^2} - 1$. $\langle A_i, Y\rangle$ is standard Gaussian, so it has sub-Gaussian parameter 1. By property of Orlicz norm, $\|\langle A_i, Y\rangle\|_{\psi_2} \leq C$ for some constant $C$. Moreover, $|\mathrm{sign}(\langle A_i, X\rangle - s_i)| \leq 1$, so

$$\|\mathrm{sign}(\langle A_i, X\rangle - s_i)\langle A_i, Y\rangle\|_{\psi_2} \leq \|\langle A_i, Y\rangle\|_{\psi_2} \leq C. \tag{C.14}$$

By property of Orlicz norm again, we know $\mathrm{sign}(\langle A_i, X\rangle - s_i)\langle A_i, Y\rangle$ is sub-Gaussian with constant sub-Gaussian parameter. By Lemma C.3, we have

$$\mathbb{E}\left[\frac{1}{m}\sum_{i=1}^m \mathrm{sign}\left(\langle A_i, X\rangle - s_i\right)\langle A_i, Y\rangle\right] = \psi(X)\langle X, Y\rangle. \tag{C.15}$$

By Chernoff bound, we can find some constant $c > 0$ such that

$$P\left(\left|\frac{1}{m}\sum_{i=1}^{m}\text{sign}\left(\langle A_i, X\rangle - s_i\right)\langle A_i, Y\rangle - \psi(X)\langle X, Y\rangle\right| \geq \delta\right) \tag{C.16}$$

$$\leq 2e^{-cm\delta^2} \tag{C.17}$$

$\square$

Lemma C.4 is an analogue of [13, Lemma 4]. Since the corruptions are not assumed to be i.i.d., the function $\psi$ is different from the quantity $\sqrt{\frac{2}{\pi}}\left((1-p) + p\mathbb{E}\left[e^{-s_i^2/2\|X\|_{\text{F}}^2}\right]\right)\frac{X}{\|X\|_{\text{F}}}$, which appears in [13, Lemma 3]. Moreover, we need to apply a (generalized) Chernoff bound for a sum of random variables with different sub-Gaussian parameters in the end of our proof rather than a concentration bound for i.i.d. random variables as done in [13, Lemma 4].

*Proof of Proposition C.1.* Without loss of generality, we only need to prove the bound holds for all $X \in \mathbb{S}_r$ with high probability. By Lemma I.8, we can find $\epsilon$-nets $\mathbb{S}_{\epsilon,r} \subset \mathbb{S}_r$, $\mathbb{S}_{\epsilon,1} \subset \mathbb{S}_1$ with respect to Frobenius norm and satisfy $|\mathbb{S}_{\epsilon,r}| \leq \left(\frac{9}{\epsilon}\right)^{(2d+1)r}$, $|\mathbb{S}_{\epsilon,1}| \leq \left(\frac{9}{\epsilon}\right)^{2d+1}$. For any $\bar{X} \in \mathbb{S}_{\epsilon,r}$, define $B_r(\bar{X},\epsilon) = \{X \in \mathbb{S}_r : \|X - \bar{X}\|_{\text{F}} \leq \epsilon\}$. $B_1(\bar{X},\epsilon)$ is defined similarly by $B_1(\bar{X},\epsilon) = \{X \in \mathbb{S}_1 : \|X - \bar{X}\|_{\text{F}} \leq \epsilon\}$. Then, for any $\bar{X}, \bar{Y}$ and $X \in B_r(\bar{X},\epsilon)$, $Y \in B_1(\bar{Y},\epsilon)$, we have $\langle X, Y\rangle - \langle \bar{X}, \bar{Y}\rangle = \langle X, Y - \bar{Y}\rangle + \langle X - \bar{X}, \bar{Y}\rangle$. By bounding the two terms on the RHS of the previous equality via the Cauchy-Schwarz's inequality, we have

$$\left|\langle X, Y\rangle - \langle \bar{X}, \bar{Y}\rangle\right| \leq 2\epsilon. \tag{C.18}$$

Let us also decompose the quantity of interest, $R := \frac{1}{m}\sum_{i=1}^{m}\text{sign}(\langle A_i, X\rangle - s_i)\langle A_i, Y\rangle - \psi(X)\langle X, Y\rangle$, into four terms:

$$R := \frac{1}{m}\sum_{i=1}^{m}\text{sign}(\langle A_i, X\rangle - s_i)\langle A_i, Y\rangle - \psi(X)\langle X, Y\rangle \tag{C.19}$$

$$= \underbrace{\frac{1}{m}\sum_{i=1}^{m}\text{sign}(\langle A_i, \bar{X}\rangle - s_i)\langle A_i, \bar{Y}\rangle - \psi(\bar{X})\langle \bar{X}, \bar{Y}\rangle}_{=:R_1} \tag{C.20}$$

$$+ \underbrace{\frac{1}{m}\sum_{i=1}^{m}\text{sign}(\langle A_i, \bar{X}\rangle - s_i)\langle A_i, Y\rangle - \text{sign}(\langle A_i, \bar{X}\rangle - s_i)\langle A_i, \bar{Y}\rangle}_{=:R_2} \tag{C.21}$$

$$+ \underbrace{\frac{1}{m}\sum_{i=1}^{m}\text{sign}(\langle A_i, X\rangle - s_i)\langle A_i, Y\rangle - \text{sign}(\langle A_i, \bar{X}\rangle - s_i)\langle A_i, Y\rangle}_{=:R_3} \tag{C.22}$$

$$+ \underbrace{\psi(\bar{X})\langle \bar{X}, \bar{Y}\rangle - \psi(X)\langle X, Y\rangle}_{=:R_4} \tag{C.23}$$

Recall our goal is to give a high probablity bound on $\sup_{X \in \mathbb{S}_r, Y \in \mathbb{S}_1}|R|$. To achieve this goal, we use the above decomposition and the triangle inequality, and have the following bound.

$$\sup_{X \in \mathbb{S}_r, Y \in \mathbb{S}_1}|R| = \sup_{\substack{\bar{X} \in \mathbb{S}_{\epsilon,r}, \\ \bar{Y} \in \mathbb{S}_{\epsilon,1}}} \sup_{\substack{X \in B_r(\bar{X},\epsilon) \\ Y \in B_1(\bar{Y},\epsilon)}}|R| \tag{C.24}$$

$$\leq \underbrace{\sup_{\substack{\bar{X} \in \mathbb{S}_{\epsilon,r} \\ \bar{Y} \in \mathbb{S}_{\epsilon,1}}}|R_1|}_{Z_1} + \underbrace{\sup_{\substack{\bar{X} \in \mathbb{S}_{\epsilon,r} \\ \bar{Y} \in \mathbb{S}_{\epsilon,1}}} \sup_{Y \in B_r(\bar{Y},\epsilon)}|R_2|}_{Z_2} + \underbrace{\sup_{\substack{\bar{X} \in \mathbb{S}_{\epsilon,r} \\ Y \in \mathbb{S}_1}} \sup_{X \in B_r(\bar{X},\epsilon)}|R_3|}_{Z_3} + \underbrace{\sup_{\substack{\bar{X} \in \mathbb{S}_{\epsilon,r} \\ \bar{Y} \in \mathbb{S}_{\epsilon,1}}} \sup_{\substack{X \in B_r(\bar{X},\epsilon) \\ Y \in B_1(\bar{Y},\epsilon)}}|R_4|}_{Z_4} \tag{C.25}$$

By C.18 and $\psi(X) = \psi(\bar{X}) \leq 1$, we obtain

$$Z_4 \leq 2\epsilon. \tag{C.26}$$

Then we hope to bound $Z_1, Z_2, Z_3$ separately. By union bound and Lemma C.4, we have $Z_1 \leq \delta_1$ with probability at least $1 - 2\left|S_{\epsilon,r}\right|\left|S_{\epsilon,1}\right| e^{-cm\delta_1^2}$. On the other hand, by $\ell_1/\ell_2$-rip (I.6),

$$Z_2 \leq \sup_{\bar{Y} \in \mathbb{S}_{\epsilon,1}, Y \in B_1(\bar{Y}, \epsilon)} \frac{1}{m} \sum_{i=1}^{m} \left|\langle A_i, Y - \bar{Y} \rangle\right| \tag{C.27}$$

$$\leq \epsilon \sup_{Z \in \mathbb{S}_2} \frac{1}{m} \sum_{i=1}^{m} \left|\langle A_i, Z \rangle\right| \tag{C.28}$$

$$\leq \epsilon \left(\sqrt{\frac{2}{\pi}} + \delta_2\right) \tag{C.29}$$

with probability at least $1 - e^{-cm\delta_2^2}$, given $m \gtrsim d$.
Moreover, by Cauchy-Schwartz inequality,

$$Z_3 \leq \sup_{\substack{\bar{X} \in S_{\epsilon,r} \\ X \in B_r(\bar{X}, \epsilon)}} \left(\frac{1}{m} \sum_{i=1}^{m} \left(\mathrm{sign}(\langle A_i, \bar{X} \rangle - s_i) - \mathrm{sign}(\langle A_i, X \rangle - s_i)\right)^2\right)^{\frac{1}{2}} \sup_{Y \in \mathbb{S}_1} \left(\frac{1}{m} \sum_{i=1}^{m} \langle A_i, Y \rangle^2\right)^{\frac{1}{2}}.$$
$$\tag{C.30}$$

By $\ell_2$-rip (I.7), we know

$$\sup_{Y \in \mathbb{S}_1} \frac{1}{m} \sum_{i=1}^{m} \langle A_i, Y \rangle^2 \leq 1 + \delta_3 \tag{C.31}$$

with probability $1 - C\exp(-Dm)$ given $m \gtrsim \frac{1}{\delta_3^2} \log(\frac{1}{\delta_3}) d$. Note that $\mathrm{sign}(\langle A_i, \bar{X} \rangle - s_i) = \mathrm{sign}(\langle A_i, X \rangle - s_i)$ if $\left|\langle A_i, X - \bar{X} \rangle\right| \leq \left|\langle A_i, \bar{X} \rangle - s_i\right|$, as a result, for any $t > 0$,

$$\sup_{\substack{\bar{X} \in S_{\epsilon,r} \\ X \in B_r(\bar{X}, \epsilon)}} \frac{1}{m} \sum_{i=1}^{m} \left(\mathrm{sign}(\langle A_i, \bar{X} \rangle - s_i) - \mathrm{sign}(\langle A_i, X \rangle - s_i)\right)^2 \tag{C.32}$$

$$\leq \sup_{\substack{\bar{X} \in S_{\epsilon,r} \\ X \in B_r(\bar{X}, \epsilon)}} \frac{4}{m} \sum_{i=1}^{m} \mathbf{1}\left(\left|\langle A_i, X - \bar{X} \rangle\right| \geq \left|\langle A_i, \bar{X} \rangle - s_i\right|\right) \tag{C.33}$$

$$\leq \sup_{\substack{\bar{X} \in S_{\epsilon,r} \\ X \in B_r(\bar{X}, \epsilon)}} \frac{4}{m} \sum_{i=1}^{m} \mathbf{1}\left(\left|\langle A_i, X - \bar{X} \rangle\right| \geq t\right) + \mathbf{1}\left(\left|\langle A_i, \bar{X} \rangle - s_i\right| \leq t\right) \tag{C.34}$$

$$\leq \underbrace{\sup_{Z \in \epsilon\mathbb{S}_{2r}} \frac{4}{m} \sum_{i=1}^{m} \mathbf{1}\left(\left|\langle A_i, Z \rangle\right| \geq t\right)}_{Z_5} + \underbrace{\sup_{\bar{X} \in \mathbb{S}_{\epsilon,r}} \frac{4}{m} \sum_{i=1}^{m} \mathbf{1}\left(\left|\langle A_i, \bar{X} \rangle - s_i\right| \leq t\right)}_{Z_6} \tag{C.35}$$

For $Z_5$, we use the simple inequality $\mathbf{1}(\left|\langle A_i, Z \rangle\right| \geq t) \leq \frac{|\langle A_i, Z \rangle|}{t}$ and $\ell_1/\ell_2$-rip (I.6) and obtain

$$Z_5 \leq \sup_{Z \in \epsilon\mathbb{S}_{2r}} \frac{4}{m} \sum_{i=1}^{m} \frac{\left|\langle A_i, Z \rangle\right|}{t} \tag{C.36}$$

$$\leq \sup_{Z \in \mathbb{S}_{2r}} \frac{4\epsilon}{m} \sum_{i=1}^{m} \frac{\left|\langle A_i, Z \rangle\right|}{t} \tag{C.37}$$

$$\leq \frac{4\epsilon(1 + \delta_4)}{t} \tag{C.38}$$

with probability at least $1 - e^{-cm\delta_4^2}$ given $m \gtrsim dr$.
For $Z_6$, we firstly use Chernoff's bound for each fixed $\bar{X}$ and get

$$\frac{1}{m} \sum_{i=1}^{m} \mathbf{1}\left(\left|\langle A_i, \bar{X} \rangle - s_i\right| \leq t\right) \leq \mathbb{E}\left[\mathbf{1}\left(\left|\langle A_i, \bar{X} \rangle - s_i\right| \leq t\right)\right] + \delta_5 \tag{C.39}$$

with probability at least $1 - e^{cm\delta_5^2}$. On the other hand, for fixed $\bar{X} \in \mathbb{S}_{\epsilon,r}$, $\langle A_i, \bar{X} \rangle$ is standard Gaussian. Since the density function of Gaussian is bounded above by $\frac{1}{2\pi}$, we always have

$$\mathbb{E}\left[1\left(\left|\langle A_i, \bar{X} \rangle - s_i\right| \leq t\right)\right] \leq \frac{2}{\sqrt{2\pi}} t \leq t. \tag{C.40}$$

Consequently,

$$\frac{1}{m} \sum_{i=1}^{m} 1\left(\left|\langle A_i, \bar{X} \rangle - s_i\right| \leq t\right) \leq t + \delta_5 \tag{C.41}$$

with probability at least $1 - e^{-cm\delta_5^2}$. By union bound, we have

$$Z_6 \leq 4t + 4\delta_5 \tag{C.42}$$

with probability at least $1 - |\mathbb{S}_{\epsilon,r}| e^{-cm\delta_5^2}$. Combining, we have

$$\sup_{X,Y \in \mathbb{S}_r} \left|\frac{1}{m} \sum_{i=1}^{m} \text{sign}(\langle A_i, X \rangle - s_i) \langle A_i, Y \rangle - \psi(X) \langle X, Y \rangle\right| \tag{C.43}$$

$$\leq \delta_1 + \epsilon\left(\sqrt{\frac{2}{\pi}} + \delta_2\right) + \sqrt{1 + \delta_3}\sqrt{\frac{4\epsilon(1 + \delta_4)}{t} + 4t + 4\delta_5} + 2\epsilon \tag{C.44}$$

with probability as least $1 - 2|S_{\epsilon,r}||S_{\epsilon,1}| e^{-cm\delta_1^2} - e^{-cm\delta_2^2} - C\exp(-Dm) - e^{-cm\delta_4^2} - |\mathbb{S}_{\epsilon,r}| e^{-cm\delta_5^2}$, given $m \gtrsim \max\{\frac{1}{\delta_3^2} \log\left(\frac{1}{\delta_3}\right) d, dr\}$. Take $\delta_1 = \delta, \delta_2 = \delta_3 = \delta_4 = \frac{1}{2}, \delta_5 = \delta^2, t = \delta^2, \epsilon = \delta^4$, we have

$$\sup_{X,Y \in \mathbb{S}_r} \left|\frac{1}{m} \sum_{i=1}^{m} \text{sign}(\langle A_i, X \rangle - s_i) \langle A_i, Y \rangle - \psi(X) \langle X, Y \rangle\right| \lesssim \delta \tag{C.45}$$

with probability at least(given $m \gtrsim dr$)

$$1 - 2\left(\frac{9}{\delta^4}\right)^{(r+1)(2d+1)} e^{-cm\delta^2} - C'\exp(-D'm) - \left(\frac{9}{\delta^4}\right)^{(2d+1)r} e^{-cm\delta^4} \tag{C.46}$$

Given $m \gtrsim dr\delta^4 \log\left(\frac{1}{\delta}\right)$, we have

$$2\left(\frac{9}{\delta^4}\right)^{(r+1)(2d+1)} e^{-cm\delta^2} + C'\exp(-D'm) + \left(\frac{9}{\delta^4}\right)^{(2d+1)r} e^{-cm\delta^4} \tag{C.47}$$

$$\lesssim \exp\left(8r(2d+1)\log\left(\frac{9}{\delta}\right) - cm\delta\right) + \exp\left(4r(2d+1)\log\left(\frac{9}{\delta}\right) - cm\delta^4\right) \tag{C.48}$$

$$\lesssim \exp(-c'm\delta^4) \tag{C.49}$$

So if $m \gtrsim dr\delta^4 \log\left(\frac{1}{\delta}\right)$,

$$\sup_{X \in \mathbb{S}_r, Y \in \mathbb{S}_1} \left|\frac{1}{m} \sum_{i=1}^{m} \text{sign}(\langle A_i, X \rangle - s_i) \langle A_i, Y \rangle - \psi(X) \langle X, Y \rangle\right| \lesssim \delta \tag{C.50}$$

with probability at least $1 - C\exp(-c'm\delta^4)$. This implies

$$\sup_{X \in \mathbb{S}_r} \left\|\frac{1}{m} \sum_{i=1}^{m} \text{sign}(\langle A_i, X \rangle - s_i) A_i - \psi(X) X\right\| \lesssim \delta \tag{C.51}$$

by variational expression of operator norm. The proof is complete since we only need to prove RDPP for matrices with unit Frobenius norm.

$$\square$$

Proposition C.1 is an analogue of [13, Proposition 5]. Note that the function $\psi$ is different from the function $\psi$ in [13, Proposition 5] as the corruptions are not assumed to be i.i.d. in this paper. Our proof also deviates from the proof of [13, Proposition 5] in bounding the term $Z_5$, which appears in (C.35). This term corresponds to the first term on the RHS of the last line of eq. (38) in [13]. In [13], this term is bounded by [13, Lemma 8] using empirical processes tools such as Talagrand's inequality. Here, we bound the term $Z_5$ using a simple contraction argument (stated as an inline inequality before (C.36)) and the $\ell_1/\ell_2$-RIP; see (C.36)-(C.38).

## C.2 Proof of Proposition C.2

We assume for simplicity that $pm$ and $(1-p)m$ are integers. Note that

$$\frac{1}{m}\sum_{i=1}^{m}\text{sign}(\langle A_i, X\rangle - s_i)A_i - \psi(X)\frac{X}{\|X\|_F} \tag{C.52}$$

$$=\frac{1}{m}\sum_{i\in S}\text{sign}(\langle A_i, X\rangle - s_i)A_i + \frac{1}{m}\sum_{i\notin S}\text{sign}(\langle A_i, X\rangle - s_i)A_i - \sqrt{\frac{2}{\pi}}\frac{X}{\|X\|_F} \tag{C.53}$$

$$=\underbrace{\frac{1}{m}\sum_{i\in S}\text{sign}(\langle A_i, X\rangle - s_i)A_i}_{Z_1} + \underbrace{\frac{1}{m}\sum_{i\notin S}\text{sign}(\langle A_i, X\rangle)A_i - (1-p)\sqrt{\frac{2}{\pi}}\frac{X}{\|X\|_F}}_{Z_2} \tag{C.54}$$

$$\underbrace{-p\sqrt{\frac{2}{\pi}}\frac{X}{\|X\|_F}}_{Z_3} \tag{C.55}$$

We bound $Z_1, Z_2, Z_3$ separately.

- For $Z_1$, we observe the following fact: let $e_i \in \{-1, 1\}$ be sign variables. For any fixed $\{e_i\}_{i\in S}$, $\sum_{i\in S}e_iA_i$ is a GOE matrix with $N(0, pm)$ diagonal elements and $N(0, \frac{pm}{2})$ off-diagonal elements. By lemma I.10, we have

$$P\left(\left\|\sum_{i\in S}e_iA_i\right\| \geq \sqrt{pm}(\sqrt{d}+t)\right) \leq e^{-\frac{t^2}{2}}. \tag{C.56}$$

  Take $t = 2\sqrt{pm+d}$, we obtain

$$P\left(\left\|\sum_{i\in S}e_iA_i\right\| \geq \sqrt{pm}(\sqrt{d}+2\sqrt{pm+d})\right) \leq e^{-2(pm+d)}. \tag{C.57}$$

  As a result, by union bound(the union of all the possible signs), with probability at least $1 - 2^{pm}e^{-2(pm+d)} \geq 1 - e^{-(pm+d)}$,

$$\left\|\sum_{i\in S}\text{sign}(\langle A_i, X\rangle - s_i)A_i\right\| \leq \sqrt{pm}(\sqrt{d}+2\sqrt{pm+d}) \tag{C.58}$$

  for any $X$. Note also that $\sqrt{d}+2\sqrt{pm+d} \leq 3\sqrt{d}+2\sqrt{pm}$, so with probability at least $1 - \exp(-(pm+d))$,

$$\|Z_1\| \leq 3\sqrt{\frac{dp}{m}} + 2p \tag{C.59}$$

  for any $X$.

- For $Z_2$, applying Proposition C.1 with zero corruption and the assumption that $p < \frac{1}{2}$, we obtain that with probability exceeding $1 - \exp(-cm(1-p)\delta^2) \geq 1 - \exp(-c'm\delta^4)$, the following holds for all matrix $X$ with rank at most $r$,

$$\left\|\frac{1}{(1-p)m}\sum_{i\notin S}\text{sign}(\langle A_i, X\rangle)A_i - \sqrt{\frac{2}{\pi}}\frac{X}{\|X\|_F}\right\| \leq \delta, \tag{C.60}$$

  given $m \gtrsim \frac{dr(\log(\frac{1}{\delta})\vee 1)}{\delta^4}$. Consequently, given $m \gtrsim \frac{dr(\log(\frac{1}{\delta})\vee 1)}{\delta^4}$, with probability exceeding $1 - \exp(-cm(1-p)\delta^2) \geq 1 - \exp(-c'm\delta^4)$,

$$\|Z_2\| \leq \delta \tag{C.61}$$

  for any $X$ with rank at most $r$.

- For $Z_3$, we have a deterministic bound

$$\|Z_3\| \leq \sqrt{\frac{2}{\pi}}p. \tag{C.62}$$

Combining, we obtain that given $m \gtrsim \frac{dr\left(\log(\frac{1}{\delta}) \vee 1\right)}{\delta^4}$, then with probability exceeding $1 - \exp(-(pm+d)) - \exp(-c'm\delta^4)$,

$$\left\| \frac{1}{m}\sum_{i=1}^{m} \text{sign}(\langle A_i, X\rangle - s_i)A_i - \psi(X)\frac{X}{\|X\|_F} \right\| \leq 3\sqrt{\frac{dp}{m}} + 3p + \delta \tag{C.63}$$

for any $X$ with rank at most $r$.

# D   Choice of stepsize

First, we present a proposition that is the cornerstone for the choice of stepsize.

**Proposition D.1.** *Fix $p \in (0,1)$, $\epsilon \in (0,1)$. If $m \geq c_0(\epsilon^{-2}\log\epsilon^{-1})dr\log d$ for some large enough constant $c_0$, then with probability at least $1 - c_1\exp(-c_2m\epsilon^2)$, where $c_1$ and $c_2$ are some constants, we have for all symmetric matrix $G \in \mathbb{R}^{d\times d}$ with rank at most $r$,*

$$\xi_p\left(\{|\langle A_i, G\rangle|\}_{i=1}^m\right) \in [\theta_p - 2\epsilon, \theta_p + 2\epsilon]\|G\|_F, \tag{D.1}$$

*where $\xi_p(\{|\langle A_i, G\rangle|\}_{i=1}^m)$ is $p$-quantile of samples. (see Definition 5.1 in [4])*

Next, we prove a proposition that can be used to estimate $\|F_tF_t^\top - X^*\|_F$ and $\|X^*\|_F$ under corruption model 1.

**Proposition D.2.** *Suppose we are under model 1 and $y_i = \langle A_i, G\rangle + s_i$'s are given. Fix $\epsilon < 0.1$ and corruption probability $p < 0.1$. Then if $m \geq c_0(\epsilon^{-2}\log\epsilon^{-1})dr\log d$ for some large enough constant $c_0$, then with probability at least $1 - c_1\exp(-c_2m\epsilon^2)$, where $c_1$ and $c_2$ are some constants, we have for any symmetric matrix $G \in \mathbb{R}^{d\times d}$ with rank at most $r$,*

$$\xi_{\frac{1}{2}}\left(\{|y_i|\}_{i=1}^m\right) \in [\theta_{\frac{1}{2}-p-\epsilon}, \theta_{\frac{1}{2}+p+\epsilon}]\|G\|_F \tag{D.2}$$

$$\subset [\theta_{\frac{1}{2}} - L(p+\epsilon), \theta_{\frac{1}{2}} + L(p+\epsilon)]\|G\|_F, \tag{D.3}$$

*where $L > 0$ is some universal constant.*

The following proposition can be used to estimate $\|F_tF_t^\top - X^*\|_F$ and $\|X^*\|_F$ under corruption model 2.

**Proposition D.3.** *Suppose we are under model 2 and $y_i = \langle A_i, G\rangle + s_i$'s are given. Fix corruption probability $p < 0.5$. Let $\epsilon = \frac{0.5-p}{3}$. Then if $m \geq c_0dr\log d$ for some large enough constant $c_0$ depending on $p$, then with probability at least $1 - c_1\exp(-c_2m\epsilon^2)$, where $c_1$ and $c_2$ are some constants, we have for all symmetric matrix $G \in \mathbb{R}^{d\times d}$ with rank at most $r$,*

$$\xi_{\frac{1}{2}}\left(\{|y_i|\}_{i=1}^m\right) \in [\theta_{\frac{0.5-p}{3}}, \theta_{1-\frac{0.5-p}{3}}]\|G\|_F. \tag{D.4}$$

## D.1   Proof of Proposition D.1

The proof is modified from Proposition 5.1 in [4]. We first note $\langle A_i, G\rangle \sim N(0, \|G\|_F^2)$ and

$$\theta_p(|N(0, \|G\|_F^2)|) = \theta_p \cdot \|G\|_F. \tag{D.5}$$

Here $\theta_p(|N(0, \|G\|_F^2)|)$ denote the $p$-quantile of folded $N(0, \|G\|_F^2)$. It suffices to prove the bound for all symmetric matrices that have rank at most $r$ and unit Frobenius norm. For each fixed symmetric $G_0$ with $\|G_0\|_F = 1$, we know from Lemma I.9 that

$$\xi_p\left(\{|\langle A_i, G\rangle|\}_{i=1}^m\right) \in [\theta_p - \epsilon, \theta_p + \epsilon] \tag{D.6}$$

with probability at least $1 - 2\exp(-cm\epsilon^2)$ for some constant $c$ that depends on $p$. Next, we extend this result to all symmetric matrices with rank at most $r$ via a covering argument. Let $S_{\tau,r}$ be a

$\tau$-net for all symmetric matrices with rank at most $r$ and unit Frobenius norm. By Lemma I.8, $|S_{\tau,r}| \leq \left(\frac{9}{\tau}\right)^{r(2d+1)}$. Taking union bound, we obtain

$$\xi_p\left(\{|\langle A_i, G_0\rangle|\}_{i=1}^m\right) \in [\theta_p - \epsilon, \theta_p + \epsilon], \qquad \forall G_0 \in \mathbb{S}_{\tau,r} \tag{D.7}$$

with probability at least $1 - 2\left(\frac{9}{\tau}\right)^{r(2d+1)}\exp(-cm\epsilon^2)$. Set $\tau = \epsilon/(2\sqrt{d(d+m)})$. Under this event and the event that

$$\max_{i=1,2,\ldots,m}\|A_i\|_{\mathrm{F}} \leq 2\sqrt{d(d+m)}, \tag{D.8}$$

which holds with probability at least $1 - m\exp(-d(d+m)/2)$ by Lemma I.12, for any rank-$r$ matrix $G$ with $\|G\|_{\mathrm{F}} = 1$, there exists $G_0 \in \mathbb{S}_{\tau,r}$ such that $\|G - G_0\|_{\mathrm{F}} \leq \tau$, and

$$|\xi_p\left(\{|\langle A_i, G\rangle|\}_{i=1}^m\right) - \xi_p\left(\{|\langle A_i, G_0\rangle|\}_{i=1}^m\right)| \leq \max_{i=1,2,\ldots,m}||\langle A_i, G\rangle| - |\langle A_i, G_0\rangle|| \tag{D.9}$$

$$\leq \max_{i=1,2,\ldots,m}|\langle A_i, G - G_0\rangle| \tag{D.10}$$

$$\leq \|G_0 - G\|_{\mathrm{F}} \max_{i=1,2,\ldots,m}\|A_i\|_{\mathrm{F}} \tag{D.11}$$

$$\leq \tau 2\sqrt{d(d+m)} \tag{D.12}$$

$$\leq \epsilon. \tag{D.13}$$

The first inequality follows from Lemma I.13. Combining with (D.6), we obtain that for all symmetric with rank at most $r$ and unit Frobenius norm,

$$\xi_p\left(\{|\langle A_i, G\rangle|\}_{i=1}^m\right) \in [\theta_p - 2\epsilon, \theta_p + 2\epsilon]. \tag{D.14}$$

The rest of the proof is to show that the above bound holds with probability at least $1 - c_1\exp(-c_2 m\epsilon^2)$ for some constants $c_1$ and $c_2$ which follows exactly the same argument as proof of Proposition 5.2 in [4].

## D.2 Proof of Proposition D.2

Let $\tilde{y}_i = \langle A_i, G\rangle$ be clean samples. By lemma I.14, we have

$$\xi_{\frac{1}{2}}\left(\{|y_i|\}_{i=1}^m\right) \in [\xi_{\frac{1}{2}-p}\left(\{|\tilde{y}_i|\}_{i=1}^m\right), \xi_{\frac{1}{2}+p}\left(\{|\tilde{y}_i|\}_{i=1}^m\right)]. \tag{D.15}$$

Moreover, applying Proposition D.1 to $(\xi_{\frac{1}{2}-p}\left(\{|\tilde{y}_i|\}_{i=1}^m\right), \frac{\epsilon}{2})$ and $(\xi_{\frac{1}{2}+p}\left(\{|\tilde{y}_i|\}_{i=1}^m\right), \frac{\epsilon}{2})$, we know that if $m \gtrsim (\epsilon^{-2}\log\epsilon^{-1})dr\log d$, the we can find constants $c_1, c_2$ that with probability at least $1 - c_1\exp(-c_2 m\epsilon^2)$,

$$\xi_{\frac{1}{2}-p}\left(\{|\tilde{y}_i|\}_{i=1}^m\right) \geq \theta_{\frac{1}{2}-p-\epsilon}\|G\|_{\mathrm{F}}, \qquad \xi_{\frac{1}{2}+p}\left(\{|\tilde{y}_i|\}_{i=1}^m\right) \leq \theta_{\frac{1}{2}+p-\epsilon}\|G\|_{\mathrm{F}} \tag{D.16}$$

holds for any symmetric matrix $G$ with rank at most $r$. Combining, we obtain

$$\xi_{\frac{1}{2}}\left(\{|y_i|\}_{i=1}^m\right) \in [\theta_{\frac{1}{2}-p-\epsilon}, \theta_{\frac{1}{2}+p+\epsilon}]\|G\|_{\mathrm{F}}. \tag{D.17}$$

In addition, we easily see that $p \to \theta_p$ is a Lipschitz function with some universal Lipschitz constant $L$ in interval $[0.3, 0.7]$. As a result,

$$[\theta_{\frac{1}{2}-p-\epsilon}, \theta_{\frac{1}{2}+p+\epsilon}]\|G\|_{\mathrm{F}} \subset [\theta_{\frac{1}{2}} - L(p+\epsilon), \theta_{\frac{1}{2}} + L(p+\epsilon)]\|G\|_{\mathrm{F}}. \tag{D.18}$$

We are done.

## D.3 Proof of Proposition D.3

Let $z_i$ be the indicator random variable that

$$z_i = \begin{cases} 1 & s_i \text{ is drawn from some corruption distribution } \mathbb{P}_i \\ 0 & s_i = 0 \end{cases}. \tag{D.19}$$

Under corruption model I, $z_i$'s are i.i.d. Bernoulli random variables with parameter $p$. By standard Chernoff inequality, we obtain

$$P\left(\sum_{i=1}^m z_i - pm \geq \frac{0.5 - p}{3}m\right) = P\left(\sum_{i=1}^m z_i - pm \geq \epsilon m\right) \tag{D.20}$$

$$\leq \exp(-m\epsilon^2/2). \tag{D.21}$$

Therefore, with probability at least $1 - \exp(-m\epsilon^2/2)$, the corruption fraction is less than $p + \frac{0.5-p}{3}$. Let $\tilde{y}_i = \langle A_i, G \rangle$ be clean samples. By Lemma I.14, we have

$$\xi_{\frac{1}{2}}\left(\{|y_i|\}_{i=1}^m\right) \in \left[\xi_{\frac{1}{2}-p-\frac{0.5-p}{3}}\left(\{|\tilde{y}_i|\}_{i=1}^m\right), \xi_{\frac{1}{2}+p+\frac{0.5-p}{3}}\left(\{|\tilde{y}_i|\}_{i=1}^m\right)\right]\|G\|_{\mathrm{F}}. \tag{D.22}$$

In addition, applying Proposition D.1 to $(\xi_{\frac{1}{2}-p-\frac{0.5-p}{3}}(\{|\tilde{y}_i|\}_{i=1}^m), \frac{\epsilon}{2})$ and $(\xi_{\frac{1}{2}+p+\frac{0.5-p}{3}}(\{|\tilde{y}_i|\}_{i=1}^m), \frac{\epsilon}{2})$ , we know that($\epsilon = \frac{0.5-p}{3}$) if $m \gtrsim (\epsilon^{-2}\log\epsilon^{-1})dr\log d \gtrsim dr\log d$, the we can find constants $c_1, c_2$ that with probability at least $1 - c_1\exp(-c_2 m\epsilon^2)$,

$$\xi_{\frac{1}{2}-p-\frac{0.5-p}{3}}\left(\{|\tilde{y}_i|\}_{i=1}^m\right) \geq \theta_{\frac{1}{2}-p-\frac{0.5-p}{3}-\epsilon}\|G\|_{\mathrm{F}}, \tag{D.23}$$

$$\xi_{\frac{1}{2}+p+\frac{0.5-p}{3}}\left(\{|\tilde{y}_i|\}_{i=1}^m\right) \leq \theta_{\frac{1}{2}+p+\frac{0.5-p}{3}+\epsilon}\|G\|_{\mathrm{F}} \tag{D.24}$$

holds for any symmetric matrix $G$ with rank at most $r$. Plug in $\epsilon = \frac{0.5-p}{3}$, we obtain

$$\xi_{\frac{1}{2}}\left(\{|y_i|\}_{i=1}^m\right) \in \left[\theta_{\frac{0.5-p}{3}}, \theta_{1-\frac{0.5-p}{3}}\right]\|G\|_{\mathrm{F}} \tag{D.25}$$

for any symmetric $G$ with rank at most $r$ with the desired probability.

# E   Proof of Initialization

Throughout this section, we denote

$$\mathbb{S} := \{X \in \mathcal{S}^{d\times d}: \|X\|_{\mathrm{F}} = 1\}, \quad \mathbb{S}_r := \{X \in \mathcal{S}^{d\times d}: \|X\|_{\mathrm{F}} = 1, \mathrm{rank}(X) \leq r\}.$$

Recall that, we construct the matrix

$$D = \frac{1}{m}\sum_{i=1}^m \mathrm{sign}(y_i)A_i.$$

Based on this, we consider its eigen decomposition

$$D = U\Sigma U^\top$$

Let $\Sigma_+^k$ be the top $k \times k$ submatrix of $\Sigma$, whose diagonal entries correspond to $k$ largest eigenvalues of $\Sigma$ with negative values replaced by 0. Accordingly, we let $U_k \in \mathbb{R}^{d\times k}$ be the submatrix of $U$, formed by its leftmost $k$ columns. Then we cook up a key ingredient of initialization:

$$B = U_k(\Sigma_+^k)^{1/2}.$$

In the following, we show that the initialization is close to the ground truth solution.

**Proposition E.1** (random corruption). *Let $F_0$ be the output of Algorithm 1. Fix constant $c_0 < 0.1$. For Model 2 with a fixed $p < 0.5$ and $m \geq c_1 dr\kappa^4(\log\kappa + \log r)\log d$. Then*

$$\|F_0 F_0^\top - c^* X_\natural\| \leq c_0\sigma_r/\kappa. \tag{E.1}$$

*holds for $c^* \in \left[\sqrt{\frac{1}{2\pi}} \cdot \frac{\theta_{\frac{0.5-p}{3}}}{\theta_{\frac{1}{2}}}, \sqrt{\frac{2}{\pi}} \cdot \frac{\theta_{1-\frac{0.5-p}{3}}}{\theta_{\frac{1}{2}}}\right]$ w.p. at least $1 - c_2\exp(-\frac{c_3 m}{\kappa^4 r})$, where the constants $c_1, c_2, c_3$ depend only on $p$ and $c_0$.*

*Proof.* By Lemma E.1 with $\delta = \frac{c_0}{3\theta_{1-\frac{0.5-p}{3}}}\frac{\sigma_r}{\sigma_1\kappa\sqrt{r}}$ and Proposition D.3, we know that there exists constants $c_1, c_2, c_3$ depending on $p$ and $c_0$ such that whenever $m \geq c_1 dr\kappa^4(\log\kappa + \log r)\log d$, then with probability at least $1 - c_2\exp(-\frac{c_3 m}{\kappa^4 r})$, we have

$$\|BB^\top - \psi(X_\natural)X_\natural/\|X_\natural\|_F\| \leq \frac{c_0}{\theta_{1-\frac{0.5-p}{3}}}\frac{\sigma_r}{\sigma_1\kappa\sqrt{r}} \tag{E.2}$$

and

$$\theta_{\frac{1}{2}}\left(\{|y_i|\}_{i=1}^m\right) \in \left[\theta_{\frac{0.5-p}{3}}, \theta_{1-\frac{0.5-p}{3}}\right]\|X_\natural\|_{\mathrm{F}}. \tag{E.3}$$

Combing with the fact that $\psi(X_\natural) \in [\sqrt{\frac{1}{2\pi}}, \sqrt{\frac{2}{\pi}}]$, we obtain

$$\left\| \frac{\xi_{\frac{1}{2}}(\{|y_i|\}_{i=1}^m)}{\theta_{\frac{1}{2}}} BB^\top - \frac{\xi_{\frac{1}{2}}(\{|y_i|\}_{i=1}^m)}{\theta_{\frac{1}{2}}} \psi(X_\natural) X_\natural / \|X_\natural\|_F \right\| \tag{E.4}$$

$$\leq \frac{c_0}{\theta_{1-\frac{0.5-p}{3}}(F)} \frac{\sigma_r}{\sigma_1 \kappa \sqrt{r}} \frac{\xi_{\frac{1}{2}}(\{|y_i|\}_{i=1}^m)}{\theta_{\frac{1}{2}}} \tag{E.5}$$

$$\leq \frac{c_0}{\theta_{1-\frac{0.5-p}{3}}(F)} \frac{\sigma_r}{\sigma_1 \kappa \sqrt{r}} \frac{\theta_{1-\frac{0.5-p}{3}}}{\theta_{\frac{1}{2}}} \|X_\natural\|_F \tag{E.6}$$

$$\leq c_0 \frac{\sigma_r}{\sigma_1 \kappa \sqrt{r}} \sqrt{r}\sigma_1 \tag{E.7}$$

$$\leq c_0 \sigma_r / \kappa. \tag{E.8}$$

Let $c_* = \frac{\xi_{\frac{1}{2}}(\{|y_i|\}_{i=1}^m) \psi(X_\natural)}{\theta_{\frac{1}{2}} \|X_\natural\|_F}$, clearly we have

$$c^* \in [(1-p)\theta_{\frac{0.5-p}{3}}(F), \theta_{1-\frac{0.5-p}{3}}(F)] \subset [\frac{1}{2}\theta_{\frac{0.5-p}{3}}(F), \theta_{1-\frac{0.5-p}{3}}(F)]. \tag{E.9}$$

The result follows. $\qquad\square$

**Lemma E.1** (random corrpution). *Suppose we are under model 2 with fixed $p < 0.5$, and we are given $\delta \leq \frac{1}{10\kappa\sqrt{r}}$. Then we have universal constants $c_1, c_2, c_3$ such that whenever $m \geq c_1 \frac{d(\log(\frac{1}{\delta}) \vee 1)}{\delta^2}$, with probability at least $1 - c_2 \exp(-c_3 m\delta^2)$, we have $\tilde{X}_0 = BB^\top$ satisfying the following*

$$\|\tilde{X}_0 - \bar{X}\| \leq 3\delta, \tag{E.10}$$

*where $\bar{X} = \psi(X_\natural)X_\natural/\|X_\natural\|_F$, and $\psi(X) = \frac{1}{m}\sum_{i=1}^m \sqrt{\frac{2}{\pi}}\left(1 - p + p\mathbb{E}_{s_i \sim \mathbb{P}_i}\left[\exp(-\frac{s_i^2}{2\|X\|_F^2})\right]\right)$.*

*Proof.* By Lemma E.2, we know that with probability at least $1 - C\exp(-c'm\delta^2)$,

$$\|D - \bar{X}\| \leq \delta. \tag{E.11}$$

Here $c'$ and $C$ are some universal constants. On the other hand, $\psi(X_\natural) \geq (1-p)\sqrt{\frac{2}{\pi}} \geq \sqrt{\frac{1}{2\pi}}$, so $\lambda_r(\bar{X}) \geq \sqrt{\frac{1}{2\pi}}\frac{\sigma_r}{\sigma_1\sqrt{r}} = \sqrt{\frac{1}{2\pi}}\frac{1}{\kappa\sqrt{r}}$. By Lemma I.3 and our assumption that $\delta \leq \frac{1}{10\kappa\sqrt{r}}$, we know that the top $r$ eigenvalues of $D$ are positive. Let $C$ be the best symmetric rank $r$ approximation of $D$ with $\lambda_r(C) > 0$ and

$$U_k = [U_r \quad U_{k-r}], \tag{E.12}$$

then we can write

$$BB^\top = C + U_{k-r}\Sigma_{k-r}U_{k-r}^\top, \tag{E.13}$$

where $\Sigma_{k-r} = \text{diag}((\lambda_{r+1}(D))_+, \cdots, (\lambda_k(D))_+)$. Then we have

$$\|BB^\top - \bar{X}\| \leq \|C - \bar{X}\| + \|\Sigma_{k-r}\|. \tag{E.14}$$

Finally, given that $C$ is the best symmetric rank-$r$ approximation of $D$, we have

$$\|C - D\| \leq \sigma_{r+1}(D) = |\sigma_{r+1}(D) - \sigma_{r+1}(\bar{X})| \leq \|D - \bar{X}\| \leq \delta, \tag{E.15}$$

where for the equality, we used the fact that $\sigma_{r+1}(\bar{X}) = 0$. Combining, we obtain

$$\|C - \bar{X}\| \leq \|C - D\| + \|D - \bar{X}\| \leq 2\delta, \tag{E.16}$$

and

$$\|\Sigma_{k-r}\| \leq \|D - \bar{X}\| \leq \delta. \tag{E.17}$$

Therefore, we have

$$\|BB^\top - \bar{X}\| \leq 3\delta \tag{E.18}$$

with probability at least $1 - C\exp(-c'm\delta^2)$, given $m \gtrsim \frac{d(\log(\frac{1}{\delta}) \vee 1)}{\delta^2}$. $\qquad\square$

**Lemma E.2** (perturbation bound under random corruption). *For any $\delta > 0$, whenever $m \gtrsim \frac{d(\log(\frac{1}{\delta}) \vee 1)}{\delta^2}$, we have*

$$\left\| D - \bar{X} \right\| \leq \delta \tag{E.19}$$

*holds with probability at least $1 - C \exp(-c'm\delta^2)$. Here, $\bar{X} = \psi(X_\natural)X_\natural / \|X_\natural\|_F$, and $c'$, and $C > 0$ are some positive numerical constants.*

*Proof.* Without loss of generality, we assume $\|X_\natural\|_F = 1$. First, we prove $\left\| D - \bar{X} \right\| \leq \delta$ by invoking Lemma C.4, then follow by a union bound. For each $Y \in \mathbb{S}_1$, let $B_1(\bar{Y}, \epsilon) = \{Z \in \mathbb{S}_1 : \|Z - \bar{Y}\|_F \leq \epsilon\}$. By Lemma I.8, we can always find an $\epsilon$-net $\mathbb{S}_{\epsilon,1} \subset \mathbb{S}_1$ with respect to Frobenius norm and satisfy $|\mathbb{S}_{\epsilon,1}| \leq \left(\frac{9}{\epsilon}\right)^{2d+1}$. Based on the $\epsilon$-net and triangle inequality, one has

$$\left\| D - \bar{X} \right\| = \sup_{Y \in \mathbb{S}_1} \left| \frac{1}{m} \sum_{i=1}^{m} \text{sign}(\langle A_i, X_\natural \rangle + s_i) \langle A_i, Y \rangle - \psi(X_\natural) \langle X_\natural, Y \rangle \right| \tag{E.20}$$

$$= \sup_{\bar{Y} \in \mathbb{S}_{\epsilon,1}} \sup_{Y \in B_1(\bar{Y}, \epsilon)} \left| \frac{1}{m} \sum_{i=1}^{m} \text{sign}(\langle A_i, X_\natural \rangle + s_i) \langle A_i, Y \rangle - \psi(X_\natural) \langle X_\natural, Y \rangle \right| \tag{E.21}$$

$$\leq \underbrace{\sup_{\bar{Y} \in \mathbb{S}_{\epsilon,1}} \left| \frac{1}{m} \sum_{i=1}^{m} \text{sign}(\langle A_i, X_\natural \rangle + s_i) \langle A_i, \bar{Y} \rangle - \psi(X_\natural) \langle X_\natural, \bar{Y} \rangle \right|}_{Z_1} \tag{E.22}$$

$$+ \underbrace{\sup_{\bar{Y} \in \mathbb{S}_{\epsilon,1}} \sup_{Y \in B_r(\bar{Y}, \epsilon)} \left| \frac{1}{m} \sum_{i=1}^{m} \text{sign}(\langle A_i, X_\natural \rangle + s_i) \langle A_i, Y \rangle - \text{sign}(\langle A_i, X_\natural \rangle + s_i) \langle A_i, \bar{Y} \rangle \right|}_{Z_2}$$

$$\tag{E.23}$$

$$+ \underbrace{\sup_{\bar{Y} \in \mathbb{S}_{\epsilon,1}} \sup_{Y \in B_1(\bar{Y}, \epsilon)} \left| \psi(X_\natural) \langle X_\natural, \bar{Y} \rangle - \psi(X_\natural) \langle X_\natural, Y \rangle \right|}_{Z_3} \tag{E.24}$$

Since $\psi(X) = \psi(\bar{X}) \leq 1$, we obtain

$$Z_3 \leq \|X_\natural\|_F \|\bar{Y} - Y\|_F \leq \epsilon. \tag{E.25}$$

Then we hope to bound $Z_1, Z_2$ separately. By union bound and Lemma C.4, we have $Z_1 \leq \tilde{\delta}$ with probability at least $1 - 2 |S_{\epsilon,1}| e^{-cm\tilde{\delta}^2}$. On the other hand, by $\ell_1/\ell_2$-rip (I.6),

$$Z_2 \leq \sup_{\bar{Y} \in \mathbb{S}_{\epsilon,1}, Y \in B_1(\bar{Y}, \epsilon)} \frac{1}{m} \sum_{i=1}^{m} |\langle A_i, Y - \bar{Y} \rangle| \tag{E.26}$$

$$\leq \epsilon \sup_{Z \in \mathbb{S}_2} \frac{1}{m} \sum_{i=1}^{m} |\langle A_i, Z \rangle| \tag{E.27}$$

$$\leq \epsilon \left( \sqrt{\frac{2}{\pi}} + 1 \right) \tag{E.28}$$

with probability at least $1 - e^{-cm}$, given $m \gtrsim d$.
Combining, we have

$$\left\| D - \bar{X} \right\| \leq \delta_1 + \epsilon \left( \sqrt{\frac{2}{\pi}} + 1 \right) + \epsilon \tag{E.29}$$

with probability as least $1 - 2 |S_{\epsilon,1}| e^{-cm\tilde{\delta}^2} - e^{-cm}$, given $m \gtrsim d$. Take $\tilde{\delta} = \delta/3, \epsilon = \delta/10$, we have

$$\left\| D - \bar{X} \right\| \leq \delta \tag{E.30}$$

with probability at least(given $m \gtrsim d$)

$$1 - 2\left(\frac{90}{\delta}\right)^{(2d+1)} e^{-cm\delta^2} - e^{-cm} \tag{E.31}$$

Given $m \gtrsim \frac{d\left(\log(\frac{1}{\delta})\vee 1\right)}{\delta^2}$, we have

$$2\left(\frac{90}{\delta}\right)^{(2d+1)} e^{-cm\delta^2} + e^{-cm} \tag{E.32}$$

$$\lesssim \exp\left((2d+1)\log\left(\frac{90}{\delta}\right) - cm\delta^2\right) + \exp(-cm) \tag{E.33}$$

$$\lesssim \exp(-c'm\delta^2) \tag{E.34}$$

So if $m \gtrsim \frac{d\log\left(\frac{1}{\delta}\right)}{\delta^2}$,

$$\left\|D - \bar{X}\right\| \leq \delta \tag{E.35}$$

with probability at least $1 - C\exp(-c'm\delta^2)$. $\qquad\square$

**Proposition E.2** (arbitrary corruption). *Let $F_0$ be the output of Algorithm 1. Fix constant $c_0 < 0.1$. For model 1 with $p \leq \frac{\tilde{c}_0}{\kappa^2\sqrt{r}}$ where $\tilde{c}_0$ depends only on $c_0$, there exist constants $c_1, c_2, c_3$ depending only on $c_0$ such that whenever $m \geq c_1 dr\kappa^2 \log d(\log\kappa + \log r)$, we have*

$$\|F_0 F_0^\top - c^* X_\natural\| \leq c_0\sigma_r/\kappa. \tag{E.36}$$

*with probability at least $1 - c_2\exp(-c_3\frac{m}{\kappa^4 r}) - \exp(-(pm + d))$. Here $c^* = 1$.*

*Proof.* Taking $\epsilon = \frac{c_0\theta_{\frac{1}{2}}}{4L\kappa^2}$ in Proposition D.2, where $L$ is a universal constant doesn't depend on anything from Proposition D.2, we know that with probability at least $1 - c_5\exp(-c_6\frac{m}{\kappa^4})$

$$\xi_{\frac{1}{2}}\left(\{|y_i|\}_{i=1}^m\right) \in [\theta_{\frac{1}{2}} - L(p+\epsilon), \theta_{\frac{1}{2}} + L(p+\epsilon)]\|X_\natural\|_{\mathrm{F}}, \tag{E.37}$$

given $m \geq c_7 dr\kappa^4 \log d \log\kappa$. Here $c_5, c_6, c_7$ are constants depending only on $c_0$. Given $\tilde{c}_0 = \frac{c_0}{4L}$, the above inclusion implies that

$$\left|1 - \frac{\xi_{\frac{1}{2}}\left(\{|y_i|\}_{i=1}^m\right)}{\|X_\natural\|_{\mathrm{F}}\theta_{\frac{1}{2}}}\right| \leq \frac{L(p+\epsilon)}{\theta_{\frac{1}{2}}} \leq \frac{c_0}{2\kappa^2}. \tag{E.38}$$

Take $\delta = \frac{c_0\sqrt{\frac{2}{\pi}}}{12(1+\frac{L}{\theta_{\frac{1}{2}}})}\frac{\sigma_r}{\sigma_1\kappa\sqrt{r}}$ in lemma E.3, we know that with probability at least $1 - c_8\exp(-c_9\frac{m}{\kappa^4 r}) - \exp(-(pm + d))$ for constants $c_8, c_9$ depending only on $c_0$,

$$\left\|BB^\top - \sqrt{\frac{2}{\pi}}X_\natural/\|X_\natural\|_F\right\| \leq \frac{c_0\sqrt{\frac{2}{\pi}}}{2(1+\frac{L}{\theta_{\frac{1}{2}}})}\frac{\sigma_r}{\sigma_1\kappa\sqrt{r}}, \tag{E.39}$$

given $m \gtrsim dr\kappa^4(\log\kappa + \log r)$. The above inequality implies that

$$\left\|\frac{\theta_{\frac{1}{2}}(\{|y_i|\}_{i=1}^m)}{\sqrt{\frac{2}{\pi}}\theta_{\frac{1}{2}}}BB^\top - \frac{\theta_{\frac{1}{2}}(\{|y_i|\}_{i=1}^m)X_\natural}{\|X_\natural\|_F\theta_{\frac{1}{2}}}\right\| \leq \frac{1+\frac{L}{\theta_{\frac{1}{2}}}}{\sqrt{\frac{2}{\pi}}}\frac{c_0\sqrt{\frac{2}{\pi}}}{2(1+\frac{L}{\theta_{\frac{1}{2}}})}\frac{\sigma_r}{\sigma_1\sqrt{r}}\|X_\natural\|_F \tag{E.40}$$

$$\leq \frac{c_0\sigma_r}{2}. \tag{E.41}$$

$\qquad\square$

Combining, we can find some constants $c_1, c_2, c_3$ depending only on $c_0$ such that whenever $m \geq c_1 dr\kappa^4 \log d(\log \kappa + \log r)$, then with probability at least $1 - c_2 \exp(-c_3 \frac{m}{\kappa^4 r}) - \exp(-(pm + d))$,

$$\left\| \frac{\theta_{\frac{1}{2}}(\{|y_i|\}_{i=1}^m)}{\sqrt{\frac{2}{\pi}}\theta_{\frac{1}{2}}} BB^\top - X_\natural \right\| \tag{E.42}$$

$$\leq \left\| \frac{\theta_{\frac{1}{2}}(\{|y_i|\}_{i=1}^m)}{\sqrt{\frac{2}{\pi}}\theta_{\frac{1}{2}}} BB^\top - \frac{\theta_{\frac{1}{2}}(\{|y_i|\}_{i=1}^m)X_\natural}{\|X_\natural\|_{\mathrm{F}}\theta_{\frac{1}{2}}} \right\| + \left\| \left( 1 - \frac{\theta_{\frac{1}{2}}(\{|y_i|\}_{i=1}^m)}{\|X_\natural\|_{\mathrm{F}}\theta_{\frac{1}{2}}} \right) X_\natural \right\| \tag{E.43}$$

$$\leq \frac{c_0 \sigma_r}{2\kappa} + \frac{L(p+\epsilon)}{\theta_{\frac{1}{2}}}\sigma_1 \tag{E.44}$$

$$\leq \frac{c_0 \sigma_r}{\kappa}. \tag{E.45}$$

**Lemma E.3** (arbitrary corrption). *Suppose we are given $\delta \leq \frac{1}{10\kappa\sqrt{r}}$. Suppose we are under model I with fixed $p < \delta/10$. Then we have universal constants $c_1, c_2, c_3$ such that whenever $m \geq c_1 \frac{d\left(\log(\frac{1}{\delta})\vee 1\right)}{\delta^2}$, with probability at least $1 - c_2 \exp(-c_3 m\delta^2) - \exp(-(pm + d))$, we have $\tilde{X}_0 = BB^\top$ satisfying the following*

$$\|\tilde{X}_0 - \bar{X}\| \leq 6\delta, \tag{E.46}$$

*where $\bar{X} = \psi(X_\natural)X_\natural/\|X_\natural\|_F$, and $\psi(X) = \frac{1}{m}\sum_{i=1}^m \sqrt{\frac{2}{\pi}}\left( 1 - p + p\mathbb{E}_{s_i \sim \mathbb{P}_i}\left[\exp(-\frac{s_i^2}{2\|X\|_{\mathrm{F}}^2})\right]\right)$.*

*Proof.* By Lemma E.4, given $m \geq c_1 \frac{d\left(\log(\frac{1}{\delta})\vee 1\right)}{\delta^2}$, we know that with probability at least $1 - \exp(-(pm + d)) - \exp(-c_2 m\delta^2)$,

$$\|D - \bar{X}\| \leq 2\delta. \tag{E.47}$$

On the other hand, $\psi(X_\natural) \geq (1-p)\sqrt{\frac{2}{\pi}} \geq \sqrt{\frac{1}{2\pi}}$, so $\lambda_r(\bar{X}) \geq \sqrt{\frac{1}{2\pi}}\frac{\sigma_r}{\sigma_1\sqrt{r}} = \sqrt{\frac{1}{2\pi}}\frac{1}{\kappa\sqrt{r}}$. By Lemma I.3 and our assumption that $\delta \leq \frac{1}{10\kappa\sqrt{r}}$, we know that the top $r$ eigenvalues of $D$ are positive. Let $C$ be the best symmetric rank $r$ approximation of $D$ with $\lambda_r(C) > 0$ and

$$U_k = [U_r \quad U_{k-r}], \tag{E.48}$$

then we can write

$$BB^\top = C + U_{k-r}\Sigma_{k-r}U_{k-r}^\top, \tag{E.49}$$

where $\Sigma_{k-r} = \mathrm{diag}((\lambda_{r+1}(D))_+, \cdots, (\lambda_k(D))_+)$. Then we have

$$\|BB^\top - \bar{X}\| \leq \|C - \bar{X}\| + \|\Sigma_{k-r}\|. \tag{E.50}$$

Finally, given that $C$ is the best symmetric rank-$r$ approximation of $D$, we have

$$\|C - D\| \leq \sigma_{r+1}(D) = |\sigma_{r+1}(D) - \sigma_{r+1}(\bar{X})| \leq \|D - \bar{X}\| \leq 2\delta, \tag{E.51}$$

where for the equality, we used the fact that $\sigma_{r+1}(\bar{X}) = 0$. Combining, we obtain

$$\|C - \bar{X}\| \leq \|C - D\| + \|D - \bar{X}\| \leq 4\delta, \tag{E.52}$$

and

$$\|\Sigma_{k-r}\| \leq \|D - \bar{X}\| \leq 2\delta. \tag{E.53}$$

Therefore, we have

$$\|BB^\top - \bar{X}\| \leq 6\delta \tag{E.54}$$

with probability at least $1 - \exp(-(pm + d)) - \exp(-c_2 m\delta^2)$, given $m \geq c_1 \frac{d\left(\log(\frac{1}{\delta})\vee 1\right)}{\delta^2}$. $\qquad\square$

**Lemma E.4** (perturbation bound under arbitrary corruption). *Given a fixed constant $\delta > 0$. Suppose the measurements $A_i$'s are i.i.d. GOE, $s_i$'s are from model I with fixed $p \leq \delta/10$. There exist universal constants $c_1$ and $c_2$ such that whenever $m \geq c_1 \frac{d\left(\log(\frac{1}{\delta}) \vee 1\right)}{\delta^2}$, with probability with probability at least $1 - \exp(-(pm + d)) - \exp(-c_2 m\delta^2)$, we have $D = \frac{1}{m}\sum_{i=1}^{m} \mathrm{sign}(y_i)A_i$. satisfying the following*

$$\|D - \bar{X}\| \leq 2\delta, \tag{E.55}$$

*where $\bar{X} = \sqrt{\frac{2}{\pi}}X_\natural / \|X_\natural\|_F$.*

*Proof.* Let $S$ be the set of indices that the corresponding observations are corrupted. We assume for simplicity that $pm$ and $(1-p)m$ are integers. Note that

$$D - \bar{X} = \frac{1}{m}\sum_{i=1}^{m}\mathrm{sign}(\langle A_i, X_\natural\rangle + s_i)A_i - \sqrt{\frac{2}{\pi}}\frac{X_\natural}{\|X_\natural\|_F} \tag{E.56}$$

$$= \frac{1}{m}\sum_{i \in S}\mathrm{sign}(\langle A_i, X_\natural\rangle + s_i)A_i + \frac{1}{m}\sum_{i \notin S}\mathrm{sign}(\langle A_i, X_\natural\rangle)A_i - \sqrt{\frac{2}{\pi}}\frac{X_\natural}{\|X_\natural\|_F} \tag{E.57}$$

$$= \underbrace{\frac{1}{m}\sum_{i \in S}\mathrm{sign}(\langle A_i, X_\natural\rangle + s_i)A_i}_{Z_1} + \underbrace{\frac{1}{m}\sum_{i \notin S}\mathrm{sign}(\langle A_i, X_\natural\rangle)A_i - (1 - p)\sqrt{\frac{2}{\pi}}\frac{X_\natural}{\|X_\natural\|_F}}_{Z_2} \tag{E.58}$$

$$\underbrace{- p\sqrt{\frac{2}{\pi}}\frac{X_\natural}{\|X_\natural\|_F}}_{Z_3} \tag{E.59}$$

We bound $Z_1, Z_2, Z_3$ separately.

- For $Z_1$, we observe the following fact: let $e_i \in \{-1, 1\}$ be sign variables. For any fixed $\{e_i\}_{i \in S}$, $\sum_{i \in S}e_i A_i$ is a GOE matrix with $N(0, pm)$ diagonal elements and $N(0, \frac{pm}{2})$ off-diagonal elements. By lemma I.10, we have

$$P\left(\left\|\sum_{i \in S}e_i A_i\right\| \geq \sqrt{pm}(\sqrt{d} + t)\right) \leq e^{-\frac{t^2}{2}}. \tag{E.60}$$

Take $t = 2\sqrt{pm + d}$, we obtain

$$P\left(\left\|\sum_{i \in S}e_i A_i\right\| \geq \sqrt{pm}(\sqrt{d} + 2\sqrt{pm + d})\right) \leq e^{-2(pm+d)}. \tag{E.61}$$

As a result, by union bound(the union of all the possible signs), with probability at least $1 - 2^{pm}e^{-2(pm+d)} \geq 1 - e^{-(pm+d)}$,

$$\left\|\sum_{i \in S}\mathrm{sign}(\langle A_i, X_\natural\rangle - s_i)A_i\right\| \leq \sqrt{pm}(\sqrt{d} + 2\sqrt{pm + d}). \tag{E.62}$$

Note also that $\sqrt{d} + 2\sqrt{pm + d} \leq 3\sqrt{d} + 2\sqrt{pm}$, so with probability at least $1 - \exp(-(pm + d))$,

$$\|Z_1\| \leq 3\sqrt{\frac{dp}{m}} + 2p \tag{E.63}$$

for any $X$.

- For $Z_2$, by the proof of Lemma E.2 with zero corruption and the assumption that $p < \frac{1}{2}$, we obtain that with probability exceeding $1 - \exp(-cm(1-p)\delta^2) \geq 1 - \exp(-c'm\delta^2)$, the following holds,

$$\left\| \frac{1}{(1-p)m} \sum_{i \notin S} \text{sign}(\langle A_i, X_\natural \rangle) A_i - \sqrt{\frac{2}{\pi}} \frac{X_\natural}{\|X_\natural\|_F} \right\| \leq \delta, \qquad (\text{E.64})$$

given $m \gtrsim \frac{d(\log(\frac{1}{\delta}) \vee 1)}{\delta^2}$. Consequently, given $m \gtrsim \frac{d(\log(\frac{1}{\delta}) \vee 1)}{\delta^2}$, with probability exceeding $1 - \exp(-cm(1-p)\delta^2) \geq 1 - \exp(-c'm\delta^2)$,

$$\|Z_2\| \leq \delta \qquad (\text{E.65})$$

for any $X$ with rank at most $r$.

- For $Z_3$, we have a deterministic bound

$$\|Z_3\| \leq \sqrt{\frac{2}{\pi}} p. \qquad (\text{E.66})$$

Combining, we obtain that given $m \gtrsim \frac{d(\log(\frac{1}{\delta}) \vee 1)}{\delta^2}$, then with probability exceeding $1 - \exp(-(pm+d)) - \exp(-c'm\delta^2)$,

$$\left\| \frac{1}{m} \sum_{i=1}^{m} \text{sign}(\langle A_i, X_\natural \rangle - s_i) A_i - \psi(X_\natural) \frac{X_\natural}{\|X_\natural\|_F} \right\| \leq 3\sqrt{\frac{dp}{m}} + 3p + \delta. \qquad (\text{E.67})$$

Take $\delta = \frac{c_0}{3\kappa\sqrt{r}}$ and let $m \gtrsim \frac{d(\log(\frac{1}{\delta}) \vee 1)}{\delta^2}$, we know that if $p \leq \delta/10$, we have

$$\left\| \frac{1}{m} \sum_{i=1}^{m} \text{sign}(\langle A_i, X_\natural \rangle - s_i) A_i - \psi(X_\natural) \frac{X_\natural}{\|X_\natural\|_F} \right\| \leq 2\delta \qquad (\text{E.68})$$

with probability at least $1 - \exp(-(pm+d)) - \exp(-c'm\delta^2)$.

$\square$

# F  Proof of Theorem 2.3

Here we prove the identifiability result in Section 2.

*Proof.* Using Lemma I.6, we know that the $\ell_1/\ell_2$-RIP conditions holds for $\mathcal{A}$: for some universal $c > 0$, with probability at least $1 - \exp(-cm\delta^2)$, there holds.

$$\left| \frac{1}{m} \|\mathcal{A}(X)\|_1 - \sqrt{\frac{2}{\pi}} \|X\|_F \right| \leq \delta \|X\|_F, \quad \forall X \in \mathbb{R}^{d \times d} : \text{rank}(X) \leq k + r.$$

Now for any subset $L \subset \{1, \ldots, m\}$, we can define $\mathcal{A}_L$ as $[\mathcal{A}(X)]_i = \langle A_i, X \rangle$ if $i \in L$ and $0$ otherwise. Then if the size of $L$ satisfies that $|L| \geq Cd(r+k)\log d$ for some universal constant, using Lemma I.6 again, we have with probability at least $1 - \exp(-c|L|\delta^2)$, there holds

$$\left| \frac{1}{|L|} \|\mathcal{A}_L(X)\|_1 - \sqrt{\frac{2}{\pi}} \|X\|_F \right| \leq \delta \|X\|_F, \quad \forall X \in \mathbb{R}^{d \times d} : \text{rank}(X) \leq k + r,$$

Note that the above holds for each fixed $L$. If we choose $S$ to be the set of indices of nonzero $s_i$. Using Bernstein's inequality, we know with probability at least $1 - \exp(-c\epsilon^2 m(1-p))$, $|L| \geq (1-\epsilon)(1-p)m$. Due to our model assumptions, $S$ is independent of $\mathcal{A}$. Hence, the above displayed inequality does hold for $L = S^c$ with probability at least $1 - \exp(-c_1(\epsilon^2 + \delta^2)m(1-p))$.

Let us assume the above two displayed inequalities, the second one with $L = S^c$ in the following derivation. Let $F$ is optimal for (1.2). Starting from the optimality of $F$ and $X_\natural$ has rank $r \leq k$, we have

$$0 \geq \frac{1}{m} \left\| \mathcal{A}(FF^\top) - y \right\|_1 - \frac{1}{m} \left\| \mathcal{A}(X_\natural) - y \right\|_1$$

$$= \frac{1}{m} \left\| \mathcal{A}(FF^\top - X_\natural) - s \right\|_1 - \frac{1}{m} \|s\|_1$$

$$= \frac{1}{m} \left\| \left[ \mathcal{A}_{S^c}(FF^\top - X_\natural) \right] \right\|_1 + \frac{1}{m} \left\| \left[ \mathcal{A}_S(FF^\top - X_\natural) \right] - s \right\|_1 - \frac{1}{m} \|s\|_1$$

$$\geq \frac{1}{m} \left\| \left[ \mathcal{A}_{S^c}(FF^\top - X_\natural) \right] \right\|_1 - \frac{1}{m} \left\| \left[ \mathcal{A}_S(FF^\top - X_\natural) \right] \right\|_1$$

$$= \frac{2}{m} \left\| \left[ \mathcal{A}_{S^c}(FF^\top - X_\natural) \right] \right\|_1 - \frac{1}{m} \left\| \mathcal{A}(FF^\top - X_\natural) \right\|_1$$

$$\geq \left( 2(1-p)(1-\epsilon) \left( \sqrt{\frac{2}{\pi}} - \delta \right) - \left( \sqrt{\frac{2}{\pi}} + \delta \right) \right) \left\| FF^\top - X_\natural \right\|_F .$$

Hence so long as $2(1-p)(1-\epsilon)\left( \sqrt{\frac{2}{\pi}} - \delta \right) - \left( \sqrt{\frac{2}{\pi}} + \delta \right) > 0$, we know $FF^\top = X_\natural$. The condition $2(1-p)(1-\epsilon)\left( \sqrt{\frac{2}{\pi}} - \delta \right) - \left( \sqrt{\frac{2}{\pi}} + \delta \right) > 0$ is satisfied with probability at least $1 - \exp(-c'm)$ and $m \geq C'(r+k)d \log d$ for some $c'$ and $C'$ depending on $p$.

$\square$

# G  Results under better initialization

As indicated in remarks under Theorem 3.2, we can show that the sample complexity for provable convergence is indeed $O(dk^3\kappa^4(\log \kappa + \log k) \log d)$, given $p \lesssim \frac{1}{\kappa\sqrt{r}}$ in either model. The proof consists of two theorems stated below.

**Theorem G.1.** *Suppose the following conditions hold:*

(i) *Suppose $F_0$ satisfies*

$$\|F_0 F_0^\top - X_\natural\| \leq c_0 \sigma_r \qquad (G.1)$$

*for small sufficiently small universal constant $c_0$.*

(ii) *The stepsize satisfies $0 < \frac{c_1}{\sigma_1} \leq \frac{\eta_t}{\|F_t F_t^\top - X_\natural\|_F} \leq \frac{c_2}{\sigma_1}$ for some small numerical constants $c_1 < c_2 \leq 0.01$ and all $t \geq 0$.*

(iii) *$(r+k, \delta)$-RDPP holds for $\{A_i, s_i\}_{i=1}^m$ with $\delta \leq \frac{c_3}{\kappa\sqrt{k+r}}$ and a scaling function $\psi \in \left[ \sqrt{\frac{1}{2\pi}}, \sqrt{\frac{2}{\pi}} \right]$. Here $c_3$ is some sufficiently small universal constant.*

*Then, we have a sublinear convergence in the sense that for any $t \geq 0$,*

$$\left\| F_t F_t^\top - X_\natural \right\| \leq c_5 \sigma_1 \frac{1}{\kappa + t}.$$

*Moreover, if $k = r$, then under the same set of condition, we have convergence at a linear rate*

$$\left\| F_t F_t^\top - X_\natural \right\| \leq c_6 \sigma_r \left( 1 - \frac{c_7}{\kappa} \right)^t, \qquad \forall t \geq 0.$$

*Here $c_5, c_6$ and $c_7$ are universal constants.*

*Proof.* Take $c_0 = 0.01$ and $c_3 = 0.001$. Next, by definition, $\gamma_t = \frac{\eta_t \psi(F_t F_t^\top - X_\natural)}{\|F_t F_t^\top - X_\natural\|_F}$. By the second assumption and the assumption on range of $\psi$, we know

$$\gamma_t \in \left[ \sqrt{\frac{1}{2\pi}} \frac{c_1}{\sigma_1}, \sqrt{\frac{2}{\pi}} \frac{c_2}{\sigma_1} \right]. \qquad (G.2)$$

Since we assumed $c_2 \leq 0.01$, so the stepsize condition $\gamma_t \leq \frac{0.01}{\sigma_1}$ is satisfied. Hence, both Proposition A.9 and Proposition A.10 hold for $t = 0$. We consider two cases separately.

- $k > r$, By Proposition A.9 and induction, we know

$$E_{t+1} \le E_t(1 - \gamma_t E_t) \le E_t(1 - \frac{c_\gamma}{\sigma_1} E_t), \qquad \forall t \ge 0. \tag{G.3}$$

where $c_1\sqrt{\frac{2}{\pi}} \le c_\gamma \le 0.01$. Define $G_t = \frac{c_\gamma}{\sigma_1} E_t$, then we have $G_0 < 1$ and

$$G_{t+1} \le G_t(1 - G_t), \qquad \forall t \ge 0. \tag{G.4}$$

Taking reciprocal, we obtain

$$\frac{1}{G_{t+1}} \ge \frac{1}{G_t} + \frac{1}{1 - G_t} \ge \frac{1}{G_t} + 1, \qquad \forall t \ge 0. \tag{G.5}$$

So we obtain

$$G_t \le \frac{1}{\frac{1}{G_0} + t}, \qquad \forall t \ge 0. \tag{G.6}$$

Plugging in the definition of $G_t$, we obtain

$$E_{\mathcal{T}_2+t} \le \frac{\sigma_1}{c_\gamma} \frac{1}{\frac{\sigma_1}{c_\gamma E_0} + t} \le \frac{\sigma_1}{c_\gamma} \frac{1}{\frac{100\sigma_1}{c_\gamma \sigma_r} + t} = \frac{\sigma_1}{c_\gamma} \frac{1}{\frac{100}{c_\gamma}\kappa + t} \le \frac{\sigma_1}{c_\gamma} \frac{1}{\kappa + t}. \tag{G.7}$$

Since $c_\gamma \ge c_1\sqrt{\frac{2}{\pi}}$, we can simply take $c_5 = \frac{1}{4c_1}\sqrt{\frac{\pi}{2}}$, apply Lemma I.5, and get

$$\left\| F_t F_t^\top - X_\natural \right\| \le c_5 \sigma_1 \frac{1}{\kappa + t}, \qquad \forall t \ge 0. \tag{G.8}$$

So the proof is complete in overspecified case.

- $k = r$. By Proposition A.10 and induction, we obtain

$$E_{t+1} \le (1 - \frac{\gamma_t \sigma_r}{3})E_t \le (1 - \frac{c_\gamma \sigma_r}{\sigma_1})E_t, \forall t \ge 0. \tag{G.9}$$

Applying this inequality recursively and noting $c_\gamma \ge c_1\sqrt{\frac{2}{\pi}}$, we obtain

$$E_{\mathcal{T}_2+t} \le (1 - \frac{c_\gamma \sigma_r}{\sigma_1})^t E_{\mathcal{T}_2} \le \left(1 - \frac{c_1\sqrt{\frac{2}{\pi}}}{\kappa}\right)^t 0.01\sigma_r, \forall t \ge 0. \tag{G.10}$$

Thus, we can take $c_6 = 0.01/4$, $c_7 = c_1\sqrt{\frac{2}{\pi}}$, apply Lemma I.5 and get

$$\left\| F_t F_t^\top - X_\natural \right\| \le c_6 \sigma_r \left(1 - \frac{c_7}{\kappa}\right)^t, \qquad \forall t \ge 0. \tag{G.11}$$

The proof is complete. $\qquad\square$

**Theorem G.2.** *Suppose under either Model 1 or 2, we have $m \ge c_1' dk^2 \kappa^4 (\log \kappa + \log k) \log d$ and $p \le \frac{c_2'}{\kappa\sqrt{k}}$ for some constants $c_1', c_2'$ depending only on $c_0$ and $c_3$. Then under both models, with probability at least $1 - c_4' \exp(-c_5' \frac{m}{k^2\kappa^4}) - \exp(-(pm+d))$ for some constants $c_4', c_5'$ depending only on $c_0$ and $c_3$, our subgradient method (3.1) with the initialization in Algorithm 1 and the adaptive stepsize choice (3.2) with $C_\eta \in [\frac{c_6'}{\theta_{\frac{1}{2}}\sigma_1}, \frac{c_7'}{\theta_{\frac{1}{2}}\sigma_1}]$ with some universal $c_6', c_7' \le 0.001$, converges as stated in Theorem G.1.*

*Proof.* We can WLOG only prove this for model 1 because model 2 can be reduced to model 1 by adding a small failure probability. Taking $\epsilon = \frac{c_0\theta_{\frac{1}{2}}}{4L\kappa}$ in Proposition D.2, where $L$ is a universal constant doesn't depend on anything from Proposition D.2, we know that with probability at least $1 - c_8' \exp(-c_9' \frac{m}{\kappa^2})$

$$\xi_{\frac{1}{2}}\left(\{|y_i|\}_{i=1}^m\right) \in [\theta_{\frac{1}{2}} - L(p+\epsilon), \theta_{\frac{1}{2}} + L(p+\epsilon)]\|X_\natural\|_{\mathrm{F}}, \tag{G.12}$$

given $m \geq c'_{10} dr\kappa^2 \log d \log \kappa$. Here $c'_8, c'_9, c'_{10}$ are constants depending only on $c_0$. Given $c'_2 \leq \frac{c_0 \theta_{\frac{1}{2}}}{4L}$, the above inclusion implies that

$$\left| 1 - \frac{\xi_{\frac{1}{2}}(\{|y_i|\}_{i=1}^m)}{\|X_\natural\|_{\mathrm{F}} \theta_{\frac{1}{2}}} \right| \leq \frac{L(p+\epsilon)}{\theta_{\frac{1}{2}}} \leq \frac{c_0}{2\kappa}. \tag{G.13}$$

Take $\delta = \frac{c_0\sqrt{\frac{2}{\pi}}}{12(1+\frac{L}{\theta_{\frac{1}{2}}})} \frac{\sigma_r}{\sigma_1\sqrt{r}}$ in lemma E.3, we know that with probability at least $1 - c'_{11}\exp(-c'_{12}\frac{m}{\kappa^2 r}) - \exp(-(pm+d))$ for constants $c'_{11}, c'_{12}$ depending only on $c_0$,

$$\left\| BB^\top - \sqrt{\frac{2}{\pi}} X_\natural / \|X_\natural\|_F \right\| \leq \frac{c_0\sqrt{\frac{2}{\pi}}}{2(1+\frac{L}{\theta_{\frac{1}{2}}})} \frac{\sigma_r}{\sigma_1\sqrt{r}}, \tag{G.14}$$

given $m \geq c'_{13} dr\kappa(\log\kappa + \log r)$ with $c'_{13}$ depending only on $c_0$. The above inequality implies that

$$\left\| \frac{\theta_{\frac{1}{2}}(\{|y_i|\}_{i=1}^m)}{\sqrt{\frac{2}{\pi}}\theta_{\frac{1}{2}}} BB^\top - \frac{\theta_{\frac{1}{2}}(\{|y_i|\}_{i=1}^m)X_\natural}{\|X_\natural\|_{\mathrm{F}}\theta_{\frac{1}{2}}} \right\| \leq \frac{1+\frac{L}{\theta_{\frac{1}{2}}}}{\sqrt{\frac{2}{\pi}}} \frac{c_0\sqrt{\frac{2}{\pi}}}{2(1+\frac{L}{\theta_{\frac{1}{2}}})} \frac{\sigma_r}{\sigma_1\sqrt{r}}\|X_\natural\|_{\mathrm{F}} \tag{G.15}$$

$$\leq \frac{c_0\sigma_r}{2}. \tag{G.16}$$

Combining, we can find some constants $c'_{14}, c'_{15}, c'_{16}$ depending only on $c_0$ such that whenever $m \geq c'_{14}dr\kappa^2\log d(\log\kappa + \log r)$, then with probability at least $1 - c'_{15}\exp(-c'_{16}\frac{m}{\kappa^4 r}) - \exp(-(pm+d))$,

$$\left\| \frac{\theta_{\frac{1}{2}}(\{|y_i|\}_{i=1}^m)}{\sqrt{\frac{2}{\pi}}\theta_{\frac{1}{2}}} BB^\top - X_\natural \right\| \tag{G.17}$$

$$\leq \left\| \frac{\theta_{\frac{1}{2}}(\{|y_i|\}_{i=1}^m)}{\sqrt{\frac{2}{\pi}}\theta_{\frac{1}{2}}} BB^\top - \frac{\theta_{\frac{1}{2}}(\{|y_i|\}_{i=1}^m)X_\natural}{\|X_\natural\|_{\mathrm{F}}\theta_{\frac{1}{2}}} \right\| + \left\| \left(1 - \frac{\theta_{\frac{1}{2}}(\{|y_i|\}_{i=1}^m)}{\|X_\natural\|_{\mathrm{F}}\theta_{\frac{1}{2}}}\right)X_\natural \right\| \tag{G.18}$$

$$\leq \frac{c_0\sigma_r}{2} + \frac{L(p+\epsilon)}{\theta_{\frac{1}{2}}}\sigma_1 \tag{G.19}$$

$$\leq c_0\sigma_r. \tag{G.20}$$

Thus, $\|F_0 F_0^\top - X_\natural\| \leq c_0\sigma_r$, which is the first condition.
Recall stepsize rule (3.2),

$$\tau_{\mathcal{A},y}(F) = \xi_{\frac{1}{2}}\left(\{|\langle A_i, FF^\top\rangle - y_i|\}_{i=1}^m\right), \quad \text{and} \quad \eta_t = C_\eta \tau_{\mathcal{A},y}(F_t). \tag{G.21}$$

By Proposition D.2, with same choice of $\epsilon$, we know that with probability at least least $1 - c'_{17}\exp(-c'_{18}\frac{m}{\kappa^2})$

$$\tau_{\mathcal{A},y}(F_t) \in [\theta_{\frac{1}{2}} - L(p+\epsilon), \theta_{\frac{1}{2}} + L(p+\epsilon)]\|F_t F_t^\top - X_\natural\|_{\mathrm{F}}, \quad \forall t \geq 0, \tag{G.22}$$

given $m \geq c'_{19}dr\kappa^2\log d\log\kappa$. Here $c'_{17}, c'_{18}, c'_{19}$ are constants depending only on $c_0$. By our condition on $C_\eta$, we know

$$\frac{\eta_t}{\|F_t F_t^\top - X_\natural\|_{\mathrm{F}}} \in \left[ \frac{c'_6(1 - \frac{c_0}{2\kappa})}{\sigma_1}, \frac{c'_7(1 + \frac{c_0}{2\kappa})}{\sigma_1} \right] \tag{G.23}$$

Hence, the second condition in Theorem G.1 is satisfied.
By Proposition 2.2, we know that whenever $m \gtrsim c'_{20}dk^2\kappa^4(\log k + \log\kappa)$ for some constant depending on $c_3$ and $c'_2 \leq c_3/10$, $(r+k, \delta)$-RDPP holds with $\delta \leq \frac{c_3}{\kappa\sqrt{k+r}}$ and $\psi(X) = \sqrt{\frac{2}{\pi}}$ with probability at least $1 - \exp(-(pm+d)) - \exp(-\frac{c_{21}m}{k^2\kappa^4})$ for some constant $c_{21}$ depending only on $c_3$. Since all the constants we introduced in this proof depend only on $c_0$ and $c_3$, so we can combing them and find desired $c'_i, i \geq 1$. $\qquad\square$

## H  RDPP and $\ell_1/\ell_2$-RIP

Recall our definition of $(k', \delta)$ RDPP states that for all rank at most $k'$ matrix $X$, the following holds:

$$D(X) := \frac{1}{m}\sum_{i=1}^{m}\text{sign}(\langle A_i, X\rangle - s_i)A_i, \quad \text{and} \quad \left\|D(X) - \psi(X)\frac{X}{\|X\|_{\text{F}}}\right\| \leq \delta. \tag{H.1}$$

The $(k', \delta)$ $\ell_1/\ell_2$-RIP states that for all rank $k'$ matrix $X$, the following holds.

$$\left(\sqrt{\frac{2}{\pi}} - \delta\right)\|X\|_{\text{F}} \leq \frac{1}{m}\sum_{i=1}^{m}|\langle A_i, X\rangle| \leq \left(\sqrt{\frac{2}{\pi}} + \delta\right)\|X\|_{\text{F}}. \tag{H.2}$$

We shall utilize the following top $k'$ Frobenius norm: for an matrix $Y \in \mathbb{R}^{d\times d}$

$$\|Y\|_{\text{F},\text{k}'} := \sqrt{\sum_{i=1}^{k'}\sigma_i^2(Y)} = \sup_{\text{rank}(Z)\leq k', \|Z\|_{\text{F}}=1}\langle Y, Z\rangle.$$

Here $\sigma_i(Y)$ is the $i$-th largest singular value of $Y$. The second variational characterization can be proved by considering the orthogonal projection of the rank $k'$ singular vector space of $Y$ and its complement.

Now suppose there holds the $(k', \frac{\delta}{\sqrt{k'}})$ RDPP with corruption always 0 and scale function being $\sqrt{\frac{2}{\pi}}$. Then we have

$$\begin{aligned}
\frac{\delta}{\sqrt{k'}} &\overset{(a)}{\geq} \left\|D(X) - \psi(X)\frac{X}{\|X\|_{\text{F}}}\right\| \\
&\overset{(b)}{=} \left\|\frac{1}{m}\sum_{i=1}^{m}\text{sign}(\langle A_i, X\rangle)A_i - \sqrt{\frac{2}{\pi}}\frac{X}{\|X\|_{\text{F}}}\right\| \\
&\overset{(c)}{\geq} \frac{1}{\sqrt{k'}}\|\frac{1}{m}\sum_{i=1}^{m}\text{sign}(\langle A_i, X\rangle)A_i - \sqrt{\frac{2}{\pi}}\frac{X}{\|X\|_{\text{F}}}\|_{\text{F},\text{k}'} \\
&\overset{(d)}{=} \frac{1}{\sqrt{k'}}\sup_{\text{rank}(Y)\leq k', \|Y\|_{\text{F}}\leq 1}\left\langle\frac{1}{m}\sum_{i=1}^{m}\text{sign}(\langle A_i, X\rangle)A_i - \sqrt{\frac{2}{\pi}}\frac{X}{\|X\|_{\text{F}}}, Y\right\rangle \\
&\overset{(e)}{\geq} \frac{1}{\sqrt{k'}}\left\langle\frac{1}{m}\sum_{i=1}^{m}\text{sign}(\langle A_i, X\rangle)A_i - \sqrt{\frac{2}{\pi}}\frac{X}{\|X\|_{\text{F}}}, \frac{X}{\|X\|_{\text{F}}}\right\rangle \\
&= \frac{1}{\sqrt{k'}}\left(\frac{1}{m}\sum_{i=1}^{m}\frac{|\langle A_i, X\rangle|}{\|X\|_{\text{F}}} - \sqrt{\frac{2}{\pi}}\right).
\end{aligned} \tag{H.3}$$

Here in the step $(a)$, we use the definition of RDPP. In the step $(b)$, we use the assumption on $s_i = 0$ always and $\psi = \sqrt{\frac{2}{\pi}}$. In $(c)$, we use the relationship between operator norm and top $k'$ Frobenius norm. In step $(d)$, we use the variational characterization of top $k'$ Frobenius norm. In step $(e)$, we use the fact that $X$ is rank at most $k'$. The above derivation completes one side of the $\ell_1/\ell_2$-RIP. The other side can be proved by taking $Y = -\frac{X}{\|X\|_{\text{F}}}$ in the above step $(e)$.

## I  Auxiliary Lemmas

This section contains lemmas that will be useful in the proof.

**Lemma I.1.** *Let $A$ be an $n \times n$ symmetric matrix. Suppose that $\|A\| \leq \frac{1}{2\eta}$, the largest singular value and the smallest singular value of $A(I - \eta A)$ are $\sigma_1(A) - \eta\sigma_1^2(A)$ and $\sigma_m(A) - \eta\sigma_m^2(A)$.*

*Proof.* Let $U_A\Sigma_A U_A^\top$ be the SVD of $A$. Simple algebra shows that

$$A(I - \eta A) = U_A\left(\Sigma_A - \eta\Sigma_A^2\right)U_A^\top. \tag{I.1}$$

This is exactly the SVD of $A\,(I - \eta A)$. Let $g(x) = x - \eta x^2$. By taking derivative, $g$ is monotone increasing in interval $[-\infty, \frac{1}{2\eta}]$. Since the singular values of $A(I - \eta A)$ are exactly the singular values of $A$ mapped by $g$, the result follows. $\qquad\square$

**Lemma I.2.** *Let $A$ be an $m \times n$ matrix. Suppose that $\|A\| \leq \sqrt{\frac{1}{3\eta}}$, the largest singular value and the smallest singular value of $A(I - \eta A^\top A)$ are $\sigma_1(A) - \eta\sigma_1^3(A)$ and $\sigma_m(A) - \eta\sigma_m^3(A)$.*

*Proof.* Let $U_A \Sigma_A V_A^\top$ be the SVD of $A$. Simple algebra shows that

$$A\left(I - \eta A^\top A\right) = U_A \left(\Sigma_A - \eta\Sigma_A^3\right) V_A^\top. \tag{I.2}$$

This is exactly the SVD of $A\left(I - \eta A^\top A\right)$. Let $g(x) = x - \eta x^3$. By taking derivative, $g$ is monotone increasing in interval $[-\sqrt{\frac{1}{3\eta}}, \sqrt{\frac{1}{3\eta}}]$. Since the singular values of $A(I - \eta A^\top A)$ are exactly the singular values of $A$ mapped by $g$, the result follows. $\qquad\square$

**Lemma I.1.** *Let $A$ be an $n \times n$ matrix such that $\|A\| < 1$. Then $I + A$ is invertible and*

$$\left\|(I + A)^{-1}\right\| \leq \frac{1}{1 - \|A\|}. \tag{I.3}$$

*Proof.* Since $\|A\| < 1$, the matrix $B = \sum_{i=0}^{\infty}(-1)^i A^i$ is well defined and indeed $B$ is the inverse of $I + A$. By continuity, subaddivity and submultiplicativity of operator norm,

$$\left\|(I + A)^{-1}\right\| = \|B\| \leq \sum_{i=0}^{\infty}\left\|A^i\right\| \leq \sum_{i=0}^{\infty}\|A\|^i = \frac{1}{1 - \|A\|}. \tag{I.4}$$

$\square$

**Lemma I.2.** *Let $A$ be an $r \times r$ matrix and $B$ be an $r \times k$ matrix. Then*

$$\sigma_r(AB) \leq \|A\|\,\sigma_r(B). \tag{I.5}$$

*Proof.* For any $r \times k$ matrix $C$, the variational expression of $r$-th singular value is

$$\sigma_r(C) = \sup_{\substack{\text{subspace } S \subset \mathbb{R}^k \\ \dim(S) = r}} \inf_{\substack{x \in S \\ x \neq 0}} \frac{\|Cx\|}{\|x\|}. \tag{I.6}$$

Applying this variational result twice, we obtain

$$\sigma_r(AB) = \sup_{\substack{\text{subspace } S \subset \mathbb{R}^k \\ \dim(S) = r}} \inf_{\substack{x \in S \\ x \neq 0}} \frac{\|ABx\|}{\|x\|} \tag{I.7}$$

$$\leq \sup_{\substack{\text{subspace } S \subset \mathbb{R}^k \\ \dim(S) = r}} \inf_{\substack{x \in S \\ x \neq 0}} \frac{\|A\|\,\|Bx\|}{\|x\|} \tag{I.8}$$

$$= \|A\|\,\sigma_r(B). \tag{I.9}$$

$\square$

**Lemma I.3** (Weyl's Inequality). *Let $A$ and $B$ be any $m \times n$ matrices. Then*

$$\sigma_i(A - B) \leq \|A - B\|, \qquad \forall 1 \leq i \leq \min\{m, n\}. \tag{I.10}$$

*When both $A$ and $B$ are symmetric matrices, the singular value can be replaced by eigenvalue.*

**Lemma I.4.** *Let $A$ be any $m \times n$ matrix with rank $r$. Then*

$$\|A\| \leq \|A\|_{\mathrm{F}} \leq \sqrt{r}\,\|A\|. \tag{I.11}$$

*Proof.* Let $\sigma_1 \geq \sigma_2 \geq \ldots \geq \sigma_r > 0$ be singular values of $A$. Then we know $\|A\| = \sigma_1$ and $\|A\|_{\mathrm{F}} = \sqrt{\sum_{i=1}^{r}\sigma_i^2}$. The result follows from Cauchy's inequality. $\qquad\square$

**Lemma I.5.** *Let $F_t$ be the iterates defined by algorithm 3.1. Then we have*

$$\left\| F_t F_t^\top - X_\natural \right\| \leq \left\| S_t S_t^\top - D_S^* \right\| + 2 \left\| S_t T_t^\top \right\| + \left\| T_t T_t^\top \right\|. \tag{I.12}$$

*Moreover,*

$$\max\{ \left\| S_t S_t^\top - D_S^* \right\|, \left\| S_t T_t^\top \right\|, \left\| T_t T_t^\top \right\| \} \leq \left\| F_t F_t^\top - X_\natural \right\|. \tag{I.13}$$

*Proof.* Recall that $F_t = U S_t + V T_t$ and $X_\natural = U D_S^* U^\top$, so we have

$$F_t F_t^\top - X_\natural = (U S_t + V T_t)(U S_t + V T_t)^\top - U D_S^* \tag{I.14}$$

$$= U(S_t S_t^\top - D_S^*) U^\top + U S_t V_t^\top V^\top + V T_t S_t^\top U^\top + V T_t T_t^\top V^\top \tag{I.15}$$

By triangle inequality and the fact that $\|U\| = \|V\| = 1$, we obtain

$$\left\| F_t F_t^\top - X_\natural \right\| \leq \left\| S_t S_t^\top - D_S^* \right\| + 2 \left\| S_t T_t^\top \right\| + \left\| T_t T_t \right\|. \tag{I.16}$$

For the second statement, we observe

$$\left\| S_t S_t^\top - D_S^* \right\| = \sup_{x \in \mathbb{R}^r, \|x\|=1} x^T \left( S_t S_t^\top - D_S^* \right) x \tag{I.17}$$

$$\leq \sup_{y \in \mathbb{R}^d, \|y\|=1} y^\top U \left( S_t S_t^\top - D_S^* \right) U^\top y \tag{I.18}$$

The last inequality follows from the fact that for any $x \in \mathbb{R}^r$, we can find a $y \in \mathbb{R}^d$ such that $U^\top y = x$ and $\|y\| = \|x\|$. Indeed, we can simply take $y = Ux$. On the other hand,

$$\sup_{y \in \mathbb{R}^d, \|x\|=1} y^\top U \left( S_t S_t^\top - D_S^* \right) U^\top y = \left\| U(S_t S_t^\top - D_S^*) Y^\top \right\| \tag{I.19}$$

$$\leq \left\| S_t S_t^\top - D_S^* \right\|, \tag{I.20}$$

so actually we have equality

$$\sup_{x \in \mathbb{R}^r, \|x\|=1} x^T \left( S_t S_t^\top - D_S^* \right) x = \sup_{y \in \mathbb{R}^d, \|y\|=1} y^\top U \left( S_t S_t^\top - D_S^* \right) U^\top y. \tag{I.21}$$

Clearly, the sup can be attained, let $y_* = \operatorname{argmax}_{y \in \mathbb{R}^d, \|y\|=1} y^\top U \left( S_t S_t^\top - D_S^* \right) U^\top y$. Then we claim that $y_*$ must lie in the column space of $U$. If not so, we can always take the projection of $y_*$ onto the column space of $U$ and normalize it, which will give a larger objective value, contradiction. As a result, $V^\top y^* = 0$ and we obtain

$$\left\| S_t S_t^\top - D_S^* \right\| = y_*^\top U(S_t S_t^\top - D_S^*) U^\top y_* \tag{I.22}$$

$$= y_*^\top \left( F_t F_t^\top - X_\natural \right) y_* \tag{I.23}$$

$$\leq \left\| F_t F_t^\top - X_\natural \right\|. \tag{I.24}$$

We can apply the same argument to get $\left\| S_t T_t^\top \right\| \leq \left\| F_t F_t^\top - X_\natural \right\|$ and $\left\| T_t T_t^\top \right\| \leq \left\| F_t F_t^\top - X_\natural \right\|$. $\square$

**Lemma I.6** ($\ell_1/\ell_2$-RIP, [5, Proposition 1 ]). *Let $r \geq 1$ be given, suppose sensing matrices $\{A_i\}_{i=1}^m$ have i.i.d. standard Gaussian entries with $m \gtrsim dr$. Then for any $0 < \delta < \sqrt{\frac{2}{\pi}}$, there exists a universal constant $c > 0$, such that with probability exceeding $1 - \exp(-cm\delta^2)$, we have*

$$\left( \sqrt{\frac{2}{\pi}} - \delta \right) \|X\|_{\mathrm{F}} \leq \frac{1}{m} \sum_{i=1}^m |\langle A_i, X \rangle| \leq \left( \sqrt{\frac{2}{\pi}} + \delta \right) \|X\|_{\mathrm{F}} \tag{I.25}$$

*for any rank $2r$-matrix $X$.*

**Lemma I.7** ($\ell_2$-RIP, [39, Theorem 2.3]). *Fix $0 < \delta < 1$, suppose that sensing matrices $\{A_i\}_{i=1}^m$ have i.i.d. standard Gaussian entries with $m \gtrsim \frac{1}{\delta^2} \log\left(\frac{1}{\delta}\right) dr$. Then with probability exceeding $1 - C \exp(-Dm)$, we have*

$$(1-\delta)\|X\|_{\mathrm{F}}^2 \leq \frac{1}{m} \sum_{i=1}^m \langle A_i, X \rangle^2 \leq (1+\delta)\|X\|_{\mathrm{F}}^2 \tag{I.26}$$

*for any rank-$r$ matrix $X$. Here $C, D$ are universal constants.*

*Proof.* This lemma is not exactly as Theorem 2.3 in [39] stated, but it's straight forward from the proof of this theorem. All we need to note in that paper is that the sample complexity we need is $m \gtrsim \frac{\log(\frac{1}{\delta})}{c}$, where $c$ is the constant defined in Theorem 2.3 [39] for $t$ chosen to be $\delta$. By standard concentration, $c \lesssim \frac{1}{\delta^2}$ and the result follows. $\square$

**Lemma I.8** (Covering number for symmetric low rank matrices). *Let* $\mathbb{S}_r = \{X \in \mathbb{S}^{d \times d} \colon \operatorname{rank}(X) \leq r, \|X\|_{\mathrm{F}} = 1\}$. *Then, there exists an $\epsilon$-net $\mathbb{S}_{\epsilon,r}$ with respect to the Frobenius norm satisfying* $|\mathbb{S}_{\epsilon,r}| \leq \left(\frac{9}{\epsilon}\right)^{(2d+1)r}$.

*Proof.* The proof is the same as the proof of lemma 3.1 in [39], except that we will do eigenvalue decomposition, instead of SVD. $\square$

**Lemma I.9** ( [4, Lemma A.1]). *Suppose $F(\cdot)$ is cumulative distribution function with continuous density function $f(\cdot)$. Assume the samples $\{x_i\}_{i=1}^{m}$ are i.i.d. drawn from $f$. Let $0 < p < 1$. If $l < f(\theta) < L$ for all $\theta$ in $\{\theta \colon |\theta - \theta_p| \leq \epsilon\}$, then*

$$|\theta_p(\{x_i\}_{i=1}^{m}) - \theta_p(F)| < \epsilon \tag{I.27}$$

*holds with probability at least $1 - 2\exp(-2m\epsilon^2 l^2)$. Here $\theta_p(\{x_i\}_{i=1}^{m})$ and $\theta_p(F)$ are $p$-quantiles of samples and distribution $F$ (see Definition 5.1 in [4])*

**Lemma I.10** (Concentration of operator norm). *Let $A$ be a $d$-by-$d$ GOE matrix having $N(0,1)$ diagonal elements and $N(0, \frac{1}{2})$ off-diagonal elements. Then we have*

$$\mathbb{E}\left[\|A\|\right] \leq \sqrt{d} \tag{I.28}$$

*and*

$$P\left(\|A\| - \mathbb{E}\left[\|A\|\right] \geq t\right) \leq e^{-\frac{t^2}{2}}. \tag{I.29}$$

*Proof.* We will use the following two facts [11]:

1. For a $d$-by-$d$ matrix $B$ with i.i.d. $N(0,1)$ entries,

$$\mathbb{E}\left[\|B\|\right] \leq 2\sqrt{d}. \tag{I.30}$$

2. Suppose $f$ is $L$-Lipschitz(with respect to the Euclidean norm) function and $a$ is a standard normal vector, then

$$P\left(f(a) - \mathbb{E}\left[f(a)\right] \geq t\right) \leq e^{-\frac{t^2}{2L}}. \tag{I.31}$$

Now we can prove the lemma. Firstly, we note that $A$ has the same distribution as $\frac{B+B^\top}{4}$ where $B$ has i.i.d. standard normal entries. By the first fact, we obtain

$$\mathbb{E}\left[\|A\|\right] \leq \mathbb{E}\left[\frac{\|B\| + \|B^\top\|}{4}\right] \leq \sqrt{d}. \tag{I.32}$$

On the other hand, $\|A\|$ can be written as a function of $\{A_{ii}\}$ and $\{\sqrt{2}A_{ij}\}_{i<j}$, which are i.i.d. standard normal random variables. Simple algebra yields that this function is 1-Lipschitz. By the second fact,

$$P\left(\|A\| \geq \sqrt{d} + t\right) \leq P\left(\|A\| \geq \mathbb{E}\left[\|A\|\right] + t\right) \tag{I.33}$$

$$\leq e^{-\frac{t^2}{2}}. \tag{I.34}$$

$\square$

**Lemma I.11** (Concentration for $\chi^2$ distribution). *Let $Y \sim \chi^2(n)$ be a $\chi^2$ random variable. Then we have*

$$P\left(Y \geq (1 + 2\sqrt{\lambda} + 2\lambda)n\right) \leq \exp(-\lambda n/2). \tag{I.35}$$

*Proof.* It follows from standard sub-exponential concentration inequality and the fact that the square of a standard normal random variable is sub-exponential [11]. $\square$

**Lemma I.12** ( [4, Lemma A.8]). *Suppose $A_i \in \mathbb{R}^{d \times d}$'s are independnet GOE sensing matrices having $N(0,1)$ diagonal elements and $N(0, \frac{1}{2})$ off-diagonal elements, for $i = 1, 2, \ldots, m$ and $m \geq d$. Then*

$$\max_{i=1,2,\ldots,m} \|A_i\|_{\mathrm{F}} \leq 2\sqrt{d(d+m)} \tag{I.36}$$

*holds with probability exceeding $1 - m \exp(-d(d+m)/2)$.*

*Proof.* Let $A$ be a GOE sensing matrix described in this lemma, and $A_{ij}$ be the $ij$-th entry of $A$. Since

$$\|A\|_{\mathrm{F}}^2 = \sum_{i=1}^{d} A_{ii}^2 + 2\sum_{i<j} A_{ij}^2 = \sum_{i=1}^{d} A_{ii}^2 + \sum_{i<j}(\sqrt{2}A_{ij})^2, \tag{I.37}$$

we see that $\|A\|_{\mathrm{F}}^2$ is a $\chi^2(d(d+1)/2)$ random variable. By Lemma I.11, we have

$$P\left(\|A\|_{\mathrm{F}}^2 \geq \left(1 + 2\sqrt{\lambda} + 2\lambda\right)d^2\right) \leq \exp(-\lambda d^2/2) \tag{I.38}$$

for any $\lambda > 0$. Take $\lambda = \frac{d+m}{d} \geq 2$. Simple calculus shows $2\lambda \geq 2\sqrt{\lambda} + 1$. Thus, we obtain

$$P\left(\|A\|_{\mathrm{F}}^2 \geq 4d(d+m)\right) \leq \exp(-d(d+m)/2). \tag{I.39}$$

Therefore, the proof is completed by applying the union bound. $\qquad \square$

**Lemma I.13** ( [4, Lemma A.2]). *Given vectors $x = [x_1, x_2, \ldots, x_n]$ and $y = [y_1, y_2, \ldots, y_n]$. We reorder them so that*

$$x_{(1)} \leq x_{(2)} \leq \ldots \leq x_{(n)}, \qquad and \qquad y_{(1)} \leq y_{(2)} \leq \ldots \leq y_{(n)}. \tag{I.40}$$

*Then*

$$\left|x_{(k)} - y_{(k)}\right| \leq \|x - y\|_{\infty}, \qquad \forall k = 1, 2, \ldots, n. \tag{I.41}$$

**Lemma I.14** ( [4, Lemma A.3]). *Consider corrupted samples $y_i = \langle A_i, X_\natural \rangle + s_i$ and clean samples $\tilde{y}_i = \langle A_i, X_\natural \rangle$, $i = 1, 2, \ldots, m$. If $\mu < \frac{1}{2}$ is the fraction of samples that are corrupted by outliers, for $\mu < p < 1 - \mu$, we have*

$$\theta_{p-\mu}(\{|\tilde{y}_i|\}_{i=1}^m) \leq \theta_p(\{|y_i|\}_{i=1}^m) \leq \theta_{p+\mu}(\{|\tilde{y}_i|\}_{i=1}^m) \tag{I.42}$$