# OpenReview forum: "Rank Overspecified Robust Matrix Recovery: Subgradient Method and Exact Recovery"
_NeurIPS.cc/2021/Conference — NeurIPS 2021 Poster_

### Official Review · Reviewer_skcZ · 2021-07-07

**Rating:** 7
**Confidence:** 3

**Summary:**

This paper studies the problem of recovering a positive semi-definite matrix from a set of corrupted measurements, under the setup that the rank of the target matrix is unknown. Based on some Gaussian assumptions on the sensing matrices and the corruption model, the authors prove that the nonconvex program (1.2) equipped with a subgradient algorithm can achieve exact recovery. The key to accomplish the proof is a new regularity condition called Restricted Direction Preserving Property (RDPP).

**Ethics Review Area:**

["I don’t know"]

**Limitations And Societal Impact:**

The assumption on the sensing matries is very restrictive, and the authors have adequately addressed this in Section 5.

**Main Review:**

1. Recently, tremendous attention has been attracted by the theoretical side of non-convex optimization based robust low-rank matrix recovery. Though I don't understand why that happens, this submission truly makes some progress in the direction: they resolve the issues raised by over-specified rank on the target matrix.

2. The paper is well-written. In fact, this paper is an improved version of another submission entitled "Sign-RIP: A Robust Restricted Isometry Property for Low-rank Matrix Recovery", with the same problem space and similar techniques. For the Sign-RIP paper, I have concerns about the rationality of its claims and justifications. The statements and claims in this submission are adequate, thereby I vote for acceptance.

3. The importance of low-rank matrix recovery could be made more convincing. Besides the well-known applications such as image analysis, I have seen some recent papers that apply recovery methodology to the filed of time series forecasting, e.g., "Recovery of Future Data via Convolution Nuclear Norm Minimization", "Time Series Forecasting via Learning Convolutionally Low-Rank Models".

**Time Spent Reviewing:**

about 1 hours

---

> ### Author Response · Authors · 2021-08-10
> **Response to Reviewer skcZ**
>
> Thanks for the suggestion of the references! We shall add them to make the importance of low-rank matrix recovery more convincing.

---

### Official Review · Reviewer_z2Cg · 2021-07-16

**Rating:** 7
**Confidence:** 3

**Summary:**

The authors study the problem of low-rank P.S.D matrix recovery in the presence of outliers. In particular, the paper studies a subgradient method to solve the above problem. The authors firstly define the notion of restricted direction preserving property (and show that this holds for the usual Gaussian measurements) and then propose a different rule for diminishing step size. With these conditions, the paper shows that even in the case of rank-overspecification, the method converges (linearly in certain settings) to the ground truth. The authors also perform some numerical experiments that corroborate the claims.

**Limitations And Societal Impact:**

Please see above.

**Main Review:**

Overall I think this is a fantastic paper with very impressive results. To the best of my knowledge, the results claimed in the paper are very novel in the area, and are a great addition to the literature on low-rank matrix recovery. The problem studied is an important one that has several applications, and a rich literature. I have a few minor comments/questions to the authors that I list below:

- The particular problem is for the low-rank matrix recovery problem, if one were interested in low-rank matrix completion, can the authors comment something about the RDPP for say random bernoulli/uniform random sampling sort of matrices?

- The wording of the two settings on $s_i$'s (Def.  Model1, 2) is a bit confusing and can be improved to highlight the differences better

- In Def 2.1, if $\psi(X)$ is just a scalar, why use a function notation at all?

- There are some discrepancies in $D(X)$ and $\mathcal{D}(X)$ throughout the paper.

- Are the entries in Table 2 based on a Monte-Carlo averaging? Also table 2 seems to contradict the results of [13] which claim (in an order notation at least) that Gaussians satisfy sign-RIP? I believe including a few settings where sign-RIP holds would improve the exposition of the paper.

- I also think that given the close connection with sign-RIP a more detailed discussion (the paragraph following Table 2) regarding these properties is warranted.

- In the expectation discussion below (3.2), what is the expectation taken over? because there are potentially several sources of randomness.

- this is just a question out of curiosity: so since there is some implicit regularization that is happening, would the authors be able to comment on how the stable rank of $F_t F_t^\top$ evolves over iterations?


**Time Spent Reviewing:**

2

---

> ### Author Response · Authors · 2021-08-10
> **Response to Reviewer z2Cg**
>
> Thank you for your comments and time invested in reviewing our work. Here we reply reviewer's concerns point by point. Reviewer's comment is after the letter capital **Q** and our response is after the letter capital **A**.
>
> **Q**: The particular problem is for the low-rank matrix recovery problem, if one were interested in low-rank matrix completion, can the authors comment something about the RDPP for say random bernoulli/uniform random sampling sort of matrices?
>
> **A**: Empirically, we find that our algorithm works for matrix completion problems. Theoretically, we conjecture that Bernoulli/uniform random sampling does not satisfy the current version of RDPP, but a relaxed version thereof holds. Another major difficulty is showing the iterates having certain incoherence properties for the matrix completion problem. We leave it as an interesting future direction to analyze nonsmooth, robust formulation of matrix completion (potentially with rank overspecification).
>
>
> **Q**: If $\psi(X)$ is just a scalar, why use a function notation at all?
>
> **A**: $\psi(X)$ is a scalar and is constant for some cases (e.g., Model 1 in proposition 2.2), but varies for other cases (e.g., Model 2 in Proposition 2.2). We will clarify this in the paper.
>
>
> **Q**: Are the entries in Table 2 based on a Monte-Carlo averaging? Also table 2 seems to contradict the results of [13] which claim (in an order notation at least) that Gaussians satisfy sign-RIP? I believe including a few settings where sign-RIP holds would improve the exposition of the paper.
>
> **A**: The reviewer is correct that Table 2 is based on a Monte-Carlo averaging. We tried different settings, but we observed similar results indicating that the sign-RIP (based on the Frobenius norm) is hard to satisfy. Although this seems to contradict [13], as far as we could see, this is because there is a discrepancy between its proof and theorem statement in [13]: the proof establishes the operator norm, but the theorem (i.e., sign-RIP) is about Frobenius norm.
>
> **Q**: There are some discrepancies in $D(X)$ and $\mathcal{D}(X)$ throughout the paper.
>
> **A**: As stated in Section 2, $D(X)$ (eq. 2.2) is a particular member of $\mathcal{D}(X)$ (eq. 2.1). We will resolve the discrepancies in the revision.
>
>
> **Q**: "I also think that given the close connection with sign-RIP a more detailed discussion (the paragraph following Table 2) regarding these properties is warranted."
>
> **A**: We thank the reviewer for the suggestion. We will expand the current discussions on the relationship between RDPP and sign RIP, which is among line 130 - 136.
>
> **Q**: In the expectation discussion below (3.2), what is the expectation taken over? because there are potentially several sources of randomness.
>
> **A**: Here the matrix $F$ is considered fixed and constant,  the only randomness is in the measurement $A_1$.
>
> **Q**: this is just a question out of curiosity: so since there is some implicit regularization that is happening, would the authors be able to comment on how the stable rank of
>  evolves over iterations?
>
> **A**: We check the rank by the evolution of the singular values. Interestingly, we found that the $r+1$-th singular value is the one that decays slowly (which is still small compared to $r$-th singular value), while the $r+2,r+3,\dots,d$-th singular values decay very fast to $10^{-10}$.
>
> We will also address all the other minor comments in the revision.

---

### Official Review · Reviewer_aMMW · 2021-07-16

**Rating:** 7
**Confidence:** 4

**Summary:**

This work studies the problem of recovering a low-rank matrix from grossly corrupted linear measurements. Specifically, the authors analyze an $\ell_1$ empirical risk minimization problem by parameterizing the unknown matrix via the factorization $FF^T$, where the inner dimension of $FF^T$ can be over-specified (i.e., larger than the ground truth rank $r$). The authors show that rigorous recovery guarantees are possible in both the exact specification and over-specified case with a sufficiently small amount of corruption. Subgradient descent with proper initialization and an adaptive step-size can exactly recover the low-rank matrix at a sublinear rate and linear rate in the over-specified and exactly specified case, respectively. To prove this, the authors introduce a regularity condition, called the Restricted Direction Preserving Property (RDPP), and show that satisfaction of this condition is sufficient for subgradient descent to succeed, as this condition essentially states that subgradients behave like their expectations. Experiments show that the proposed algorithm works comparably well to other descent methods with decaying step-size schedules and also that decaying step-sizes could potentially help the Deep Image Prior (DIP) from overfitting to sparse corruptions.

**Limitations And Societal Impact:**

As the work is theoretical in nature, there are no foreseeable societal impacts. Some limitations on the theory are discussed near the end, and one I would add here is the sub-optimal sample complexity scaling.

**Main Review:**

# Originality

The proposed work provides a new sufficient condition on the measurement matrix and corruption statistics that is a variant of previous work on robust matrix sensing. While the RDPP is similar cosmetically to previous work, the novelty lies in the result that it is shown to be sufficient for local convergence of a subgradient descent algorithm with adaptive step sizes. Moreover, this convergence holds for general rank $r$ ground truth matrices and rank over-specification $k \geqslant r$.

# Quality

- Strengths: - The work is well-written and easy to follow. - The results hold for general rank $r$ and any amount of over-specification $k \geqslant r$. - The proposed algorithm can exactly recover the underlying low-rank matrix in the presence of sparse corruptions, and convergence rates on the algorithm are shown.

- Weaknesses: - The results require spectral initialization, although the initial error bound in the present work is less stringent in previous works. Experimentally, was the spectral initialization scheme necessary for subGD with the proposed step sizes to converge? The case $k = d$ used random initialization with small norm, but the over-specified and exact rank case all used spectral initialization. - The overall sample complexity of the convergence results appears off by a large factor of $k^2$. Could the authors comment on where this bottleneck arises and where the difficulty lies? This may be helpful for the reader to know where potential improvements could be made. It appears this factor is due to the sample complexity required for the RDPP, as it scales like $\delta^{-4}$, and the theory requires $\delta \leqslant 1/\sqrt{k}$.

# Clarity

I found the paper easy to follow, and gained a good amount of intuition from the proof sketches and discussion of results.

- Typos: pg 1 line 13 “adversary sparse” -> “adversarial sparse”, pg 3 line 112 “facterized” -> “factorized”, pg 3 line 117 “dynamic” -> “dynamics”, pg 5 line 164 is missing a period

Other comments:

- I believe that the proof of Proposition 2.2 is in Appendix C instead of D.
- Eq (2.1) should have $y_i$ instead of $s_i$.
- The results shown for DIP are certainly interesting, but the proposed method wasn’t used in this setting (namely the adaptive step sizes). Thus the sentence on pg 2 line 57-58 may be misleading (instead one could simply note robustness to overfitting for DIP is shown with decaying step sizes). Along these lines, is there an analogous adaptive step-size rule that could be used for denoising with sparse corruptions for the DIP (say, simply taking the median of the measurements)?
- The authors commented on how analyzing other measurement matrices would be an interesting avenue for future work. Along similar lines, could the RDPP (or a variant thereof) potentially have applications to other inverse problems as well?

# Significance

The present work adds an interesting contribution to providing rigorous guarantees for the over-specified matrix sensing setting. This setting is important as, in practice, one would rarely know the exact rank of the ground-truth matrix.

**Time Spent Reviewing:**

5

---

> ### Author Response · Authors · 2021-08-10
> **Response to Reviewer aMMW**
>
> Thank you for your comments and time invested in reviewing our work. Here we reply reviewer's concerns point by point. Reviewer's comment is after the letter capital **Q** and our response is after the letter capital **A**.
>
> **Q**: The results require spectral initialization, although the initial error bound in the present work is less stringent in previous works. Experimentally, was the spectral initialization scheme necessary for subGD with the proposed step sizes to converge?
>
> **A**: In our experiments, we found that small random initialization is enough for rapid convergence for both over-specified rank $(k=2r)$ and exact rank $(k=r)$ setting. However, for random initialization, establishing theoretical guarantees is challenging even when the rank is known for the nonsmooth case.
>
> We note that for the smooth case, a recent work [1’] makes progress on analyzing small random initialization. It is an interesting future direction to investigate whether such a result could be extended to the nonsmooth, corrupted settings as considered in our paper.
>
> [1’] "Small random initialization is akin to spectral learning: Optimization and generalization guarantees for overparameterized low-rank matrix reconstruction"
>
> **Q**: The case $k=d$ used random initialization with small norms, but the over-specified and exact rank case all used spectral initialization. - The overall sample complexity of the convergence results appears off by a large factor of $k^2$. Could the authors comment on where this bottleneck arises and where the difficulty lies? This may be helpful for the reader to know where potential improvements could be made. It appears this factor is due to the sample complexity required for the RDPP, as it scales like $\delta^{-4}$, and the theory requires $\delta < 1/\sqrt{k}$
>
> **A**: The reviewer is correct that the bottleneck of sample complexity of robust recovery mainly comes from the sample complexity of RDPP, and the reviewer has made a very good point. We can improve the sample complexity of robust recovery to $O(dk^2)$ if the sample complexity of RDPP can be improved. However, the current $\delta^{-4}$ rate is the best we can get （which is in contrast to $\delta^{-2}$ in the smooth case).
> More specifically, the Cauchy-Schwartz inequality we applied in (C.31) might make the dependence on delta not tight.  Note that $O(dk^2)$ matches the best result for nonconvex approaches when the rank is known exactly, see the reply to point 6 of Reviewer x1C7.
>
> **Q**: ​​The results shown for DIP are certainly interesting, but the proposed method wasn’t used in this setting (namely the adaptive step sizes). Thus the sentence on pg 2 line 57-58 may be misleading (instead one could simply note robustness to overfitting for DIP is shown with decaying step sizes). Along these lines, is there an analogous adaptive step-size rule that could be used for denoising with sparse corruptions for the DIP (say, simply taking the median of the measurements)?
>
> **A**:  Thanks for the good suggestion. We will rephrase the sentence in line 57-58 to avoid the confusion. We implemented the adaptive step-size rule (taking the median of the measurements) for the DIP but we found it doesn’t work as expected. We leave the thorough study as future work.
>
> **Q**: The authors commented on how analyzing other measurement matrices would be an interesting avenue for future work. Along similar lines, could the RDPP (or a variant thereof) potentially have applications to other inverse problems as well?
>
> **A**: We believe that variants of RDPP can be used to analyze the nonsmooth formulation of other inverse problems (such as Robust PCA). Empirically, we find that our algorithm is applicable to a broader setting including Robust PCA.
>
> We will also address all the other minor comments in the revision.

---

> > ### Comment · Reviewer_aMMW · 2021-08-24
> > **Post-rebuttal response**
> >
> > Dear authors,
> >
> > Thank you for your detailed response and discussion! After reading the other reviews and author responses, I have decided to raise my score and recommend acceptance. I think that the paper provides a nice contribution in rigorously analyzing this non-smooth and overparameterized formulation of robust matrix recovery. It sets out what it aims to do fairly well and I think that there is potential to build off of this work.

---

### Official Review · Reviewer_x1C7 · 2021-07-23

**Rating:** 6
**Confidence:** 3

**Summary:**

The paper considers the problem of recovering a rank $r$ matrix $X$ in $\mathbb{R}^{d\times d}$ under possibly corrupted linear measurements. The paper studies the non-convex formulation, first considered by [Jianhao Ma, 2021], that uses a overspecified factorization of the unknown matrix and employs an $\ell_1$ minimization. The goal is to recover $X$ by recovering a matrix $F\in \mathbb{R}^{d\times k}$ such that $X = FF^\intercal$, with $k$ possibly bigger than $r$. The authors introduce a Restricted Direction Preserving Property (RDPP), which describes the alignment of the sub-differential of the $\ell_1$ loss function with gradient of the expectation of the loss function. The authors show that RDPP holds for random measurements with certain corruption models and use this to show that the low-rank matrix can be exactly recovered with high probability using a sub-gradient descent method that using diminishing step-size. The result requires number of measurements on the order of $O(dk^3)$, up to log factors and convergence is linear in the case $k = r$. Bulk of the paper rests on the theoretical exposition.  I did not go through the theoretical proofs from the supplementary material in detail, however the proof sketch from the main paper provides sufficient intuition.


**Limitations And Societal Impact:**

Limitations of the paper is discussed in the concluding section.

**Main Review:**

1. It is not immediately clear, by reading the introduction section, that the corruptions considered in the paper are sparse. In line 32, the authors state "in many cases corruptions $s_i$ are sparse" which falsely implies to the readers that corruptions considered in the paper could be dense and the result presented applies to this general case. Sparsity of corruptions should be made clear before stating the contributions and to motivate the $\ell_1$ formulation presented in the paper.

2. In contribution (exact recovery with rank overspecification) and in abstract, randomness (Gaussian) of the measurement is not stated. This falsely leads the readers to believe that the result applies to general measurements (for example, subsampled Fourier measurements). To my knowledge, the measurements considered in the paper is random.

3. The measurement matrices considered in the paper are generated from Gaussian Orthogonal Ensemble. How practical is this assumption? This is more restrictive than standard Gaussian measurement. Does a similar recovery result hold of standard Gaussian measurement (with entries i.i.d. $\mathcal{N}(0,1)$?

4. In line 135, it is stated that "generic sensing matrices satisfy RDPP". What are the class of matrices that satisfy RDPP? Can the result be extended to more realistic scenarios where measurements don't follow this property?

5. The subgradient algorithm studied in the paper is initialized using a spectral initialization, which generals involves computing the SVD of a matrix. This problem was reduced to computing eigenvalues/eigenvectors of a matrix because the measurement matrices were symmetric. In Line 60-61, the authors state that their algorithm "does not require SVD" which seems to be not entirely true (at least the initialization requires SVD). This should be made clear.

6. The sample complexity required for successful recovery is on the order of $O(dk^3)$, up to log factors, which is not optimal. This should be discussed in the paper, providing comparison to other related works.

7. There are some words missing or unclear phrasing in Line 63 and Line 147.




**Time Spent Reviewing:**

10

---

> ### Author Response · Authors · 2021-08-10
> **Response to Reviewer x1C7**
>
> Thank you for your comments and time invested in reviewing our work. Here we reply reviewer's concerns point by point.
>
> 1. Thanks for pointing this out. We will change the wording and make it clear that the corruption is sparse, and thus this sparsity motivates us to consider the $\ell_1$ formulation.
>
>
> 2. We will change the wording of "generic", and be more explicit on the claim that our  measurements are random Gaussian.
>
>
> 3. The same results also hold for standard Gaussian measurements with iid entries, and we will make this clear in the revision.
>
>
> 4. So far, we are able to prove RDPP for standard Gaussian measurements and GOE measurements. In the literature "generic measurement" is sometimes used to denote unstructured, random measurement matrices (such as Gaussian and GOE matrices). Again, we will remove this phrase to avoid confusion and be more explicit.
>
>       We believe RDPP also holds for measurements that are distributionally rotationally invariant. For other measurements such as matrices with iid sub-Gaussian entries, we believe that RDPP has to be modified. Empirically, we find that the algorithm is applicable to a broader setting (e.g., Robust PCA) even if the RDPP does not hold.
>
>
> 5. Thanks for the good catch. What we mean here is that SVD is only needed in initialization for once, and it is not needed during the iterations of the algorithm; we will make this more precise in the paper. Experimentally, we found we can even avoid this initialization SVD by using a small random initialization.
>
>
> 6.  Thanks for this suggestion. We did not compare with related works because the current work is for the rank overspecified setting, while most of the existing work on nonconvex approaches assumes the rank is known in advance. Nonetheless, with this caveat in mind, we will follow the reviewer’s suggestion and add comparison to the existing works. In particular, for nonconvex approaches with corrupted measurements, we find that the following papers provide the best sample complexity result, which requires $O(dr^2)$ samples to initialize and run their algorithms.
>
> Yuanxin Li, Yuejie Chi, Huishuai Zhang, and Yingbin Liang. Non-convex low-rank matrix recovery with arbitrary outliers via median-truncated gradient descent. Information and Inference: A Journal of the IMA, 9(2):289–325, 2020.
>
> Xiao Li, Zhihui Zhu, Anthony Man-Cho So, and Rene Vidal. Nonconvex robust low-rank matrix recovery. 373 SIAM Journal on Optimization, 30(1):660–686, 2020.
>
>
> 7. Thanks for the good catch. We will edit the paper, clarify the confusion and fill out the missing words accordingly.

---

### Decision · Program_Chairs · 2021-09-27

**Decision:**

Accept (Poster)

**Comment:**

There is a consensus among the reviewers that this paper presents significant results of interest to the conference.